# Single-cell and spatial multi-omics highlight effects of anti-integrin therapy across cellular compartments in ulcerative colitis

Elvira Mennillo [1], Yang Joon Kim[2], Gyehyun Lee[1], Iulia Rusu[1], Ravi K. Patel [1,3], Leah C. Dorman[2], Emily Flynn[3], Stephanie Li[1], Jared L. Bain[1], Christopher Andersen[1,3], Arjun Rao[1,3], Stanley Tamaki[3], Jessica Tsui[1,3,4], Alan Shen[1,3,4], Madison L. Lotstein[1,3], Maha Rahim[5], Mohammad Naser [6], Faviola Bernard-Vazquez[1], Walter Eckalbar[3], Soo-jin Cho [4], Kendall Beck[1], Najwa El-Nachef[1], Sara Lewin[1], Daniel R. Selvig[1], Jonathan P. Terdiman[1], Uma Mahadevan[1], David Y. Oh[1,7], Gabriela K. Fragiadakis [1,3], Angela Pisco [2], Alexis J. Combes [1,3,4] & Michael G. Kattah [1] ✉

Ulcerative colitis (UC) is driven by immune and stromal subsets, culminating in epithelial injury. Vedolizumab (VDZ) is an anti-integrin antibody that is effective for treating UC. VDZ is known to inhibit lymphocyte trafficking to the intestine, but its broader effects on other cell subsets are less defined. To identify the inflammatory cells that contribute to colitis and are affected by VDZ, we perform single-cell transcriptomic and proteomic analyses of peripheral blood and colonic biopsies in healthy controls and patients with UC on VDZ or other therapies. Here we show that VDZ treatment is associated with alterations in circulating and tissue mononuclear phagocyte (MNP) subsets, along with modest shifts in lymphocytes. Spatial multi-omics of formalin-fixed biopsies demonstrates trends towards increased abundance and proximity of MNP and fibroblast subsets in active colitis. Spatial transcriptomics of archived specimens pre-treatment identifies epithelial-, MNP-, and fibroblast-enriched genes related to VDZ responsiveness, highlighting important roles for these subsets in UC.

Ulcerative colitis (UC) is a chronic inflammatory disorder of the intestine characterized by abnormal immune, stromal, and epithelial responses to microbial stimuli in genetically susceptible individuals, culminating in mucosal inflammation and epithelial injury. Vedolizumab (VDZ) is a monoclonal antibody against $\alpha_4\beta_7$ integrin that prevents binding to gut endothelial mucosal addressin-cell adhesion molecule 1 (MAdCAM-1), blocking the trafficking of leukocytes to the intestine. The cellular and genetic factors that mediate VDZ response and non-response are incompletely characterized[1–3]. There are conflicting data regarding which inflammatory intestinal cell subsets are the most affected[4]. Many studies emphasize the effects of VDZ on B and T lymphocytes and regulatory T cells (Tregs)[3–9], while others suggest an impact on innate immune myeloid populations[10,11]. Understanding the peripheral and tissue effects of VDZ is critical to stratify

[1]Department of Medicine, University of California San Francisco, San Francisco, CA, USA. [2]Chan Zuckerberg Biohub, San Francisco, CA, USA. [3]CoLabs, University of California San Francisco, San Francisco, CA, USA. [4]Department of Pathology, University of California San Francisco, San Francisco, CA 94143, USA. [5]Helen Diller Family Comprehensive Cancer Center, University of California San Francisco, San Francisco, CA 94143, USA. [6]Biological Imaging Development CoLab, University of California San Francisco, San Francisco, CA, USA. [7]Division of Hematology/Oncology, Department of Medicine, University of California San Francisco, San Francisco, CA, USA. ✉e-mail: michael.kattah@ucsf.edu

patients as potential responders or non-responders prior to treatment, and for identifying alternative treatment strategies for non-responders.

Tissue multi-omics studies in inflammatory bowel disease (IBD) have revealed important roles for stromal, epithelial, and immune compartments in driving disease and mediating treatment response and non-response[12–17]. To generate an unbiased, global assessment of the effects of colitis within the context of VDZ treatment, we performed a comprehensive single-cell multi-omics analysis that included single-cell RNA sequencing (scRNA-seq), cellular indexing of transcriptomes and epitopes by sequencing (CITE-seq, a modality that includes single-cell RNA-seq in addition to a DNA-barcoded antibody-based proteomic panel), and mass cytometry (CyTOF) of peripheral leukocytes and colonic biopsies in healthy controls (HC) and patients with UC on aminosalicylates (UC) or VDZ (UC-VDZ). To further investigate the spatial heterogeneity and proximity of intestinal tissue subsets, we performed multiplex ion beam imaging (MIBI), co-detection by indexing (CODEX), and highly multiplexed RNA in situ *hybridization* (RNA-ISH, CosMx) on formalin-fixed, paraffin-embedded (FFPE) colonic biopsies, using both unsupervised and supervised analytic methods. In separate groups of patients, we validated our findings with additional CyTOF analyses, and applied 1000-plex spatial transcriptomics to archived clinical FFPE specimens before and after VDZ treatment.

## Results

### scRNA-seq of cryopreserved intestinal biopsies has high fidelity to fresh processing

To characterize the effects of VDZ in the peripheral blood and colon of UC patients, we first employed a cross-sectional, case-control study design consisting of healthy controls without IBD (HC, $n = 4$), patients with UC on 5-aminosalicylates (5-ASA) (UC, $n = 4$), and patients with UC on VDZ (UC-VDZ, $n = 4$) (Fig. 1a, Supplementary Table 1). To improve internal validity and cross-validate multi-omics signatures in a small number of patients, we analyzed the same patient samples using a variety of single-cell and spatial transcriptomic and proteomic methods, including scRNA-seq, CITE-seq[18], cytometry by time of flight (CyTOF), multiplex RNA-ISH, MIBI[19], and CODEX[20] (Fig. 1a,

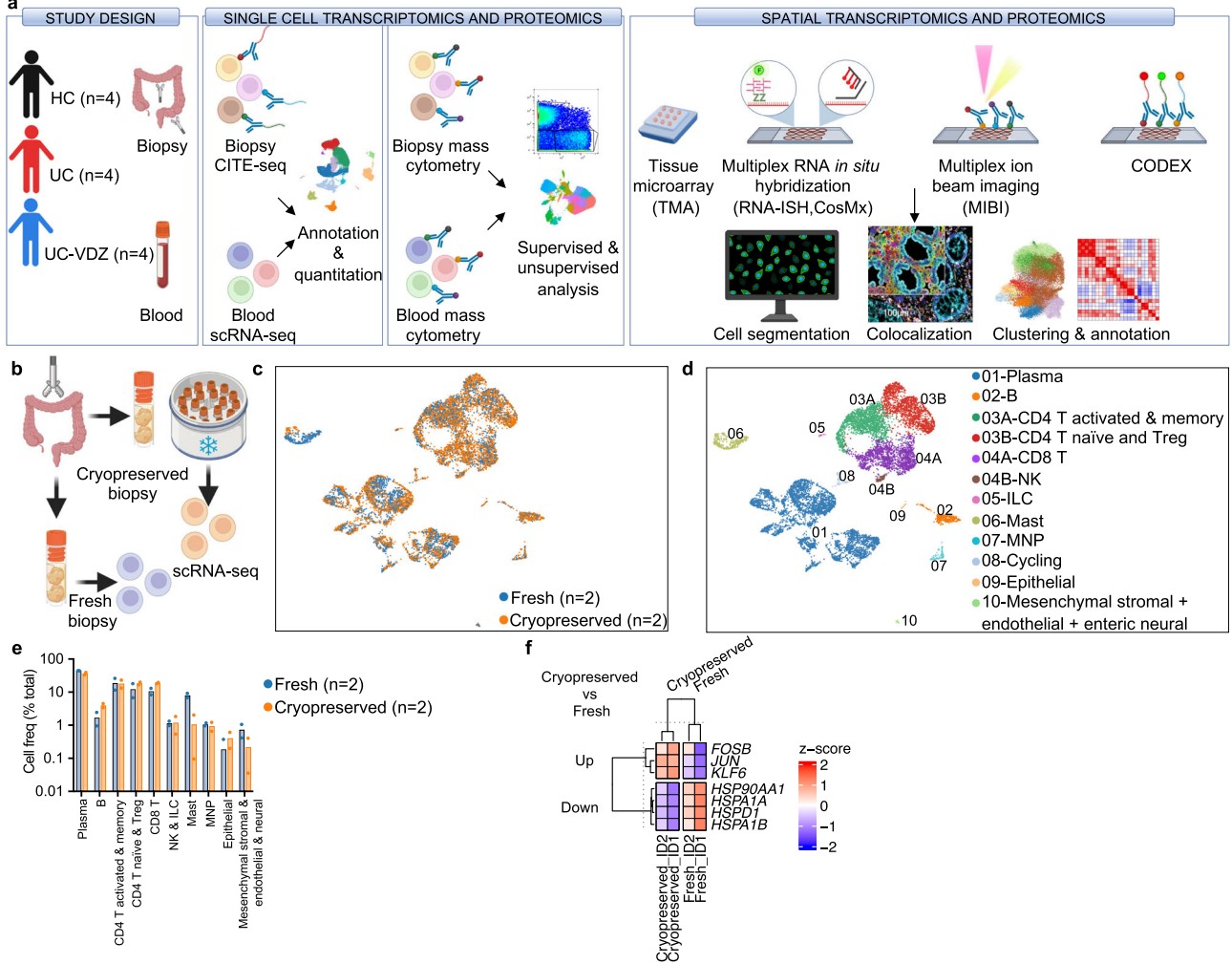

**Fig. 1 | Schematic of study design and fidelity of cryopreserved compared to fresh biopsy processing for scRNA-seq. a** Schematic of study design. Created with BioRender.com. **b** Schematic of Cryopreserved versus Fresh biopsy processing comparison. Created with BioRender.com. Representative UMAP visualization of 10,648 cells for two donors comparing (**c**) Cryopreserved versus Fresh biopsies and (**d**) coarse cell subset annotations. **e** Cell frequency as a percent of total for coarse cell subsets for two donors comparing Cryopreserved versus Fresh biopsies (mean; *n*=number of patients; each dot represents one patient sample; multiple Mann−Whitney tests with FDR correction; q < 0.1 threshold for discovery, only significant differences are indicated). **f** Heatmap of expression z-scores for differentially expressed (DE) genes with log₂ fold-change (log₂fc) > 0.4 or <−0.4 and Bonferroni *p* value < 0.1 comparing Cryopreserved (Up/Down) relative to Fresh biopsies for two study subjects ID1 and ID2 identified by MAST analysis. Only significant differences are shown. MNP mononuclear phagocyte: Treg-regulatory T cell, NK natural killer, ILC innate lymphoid cell.

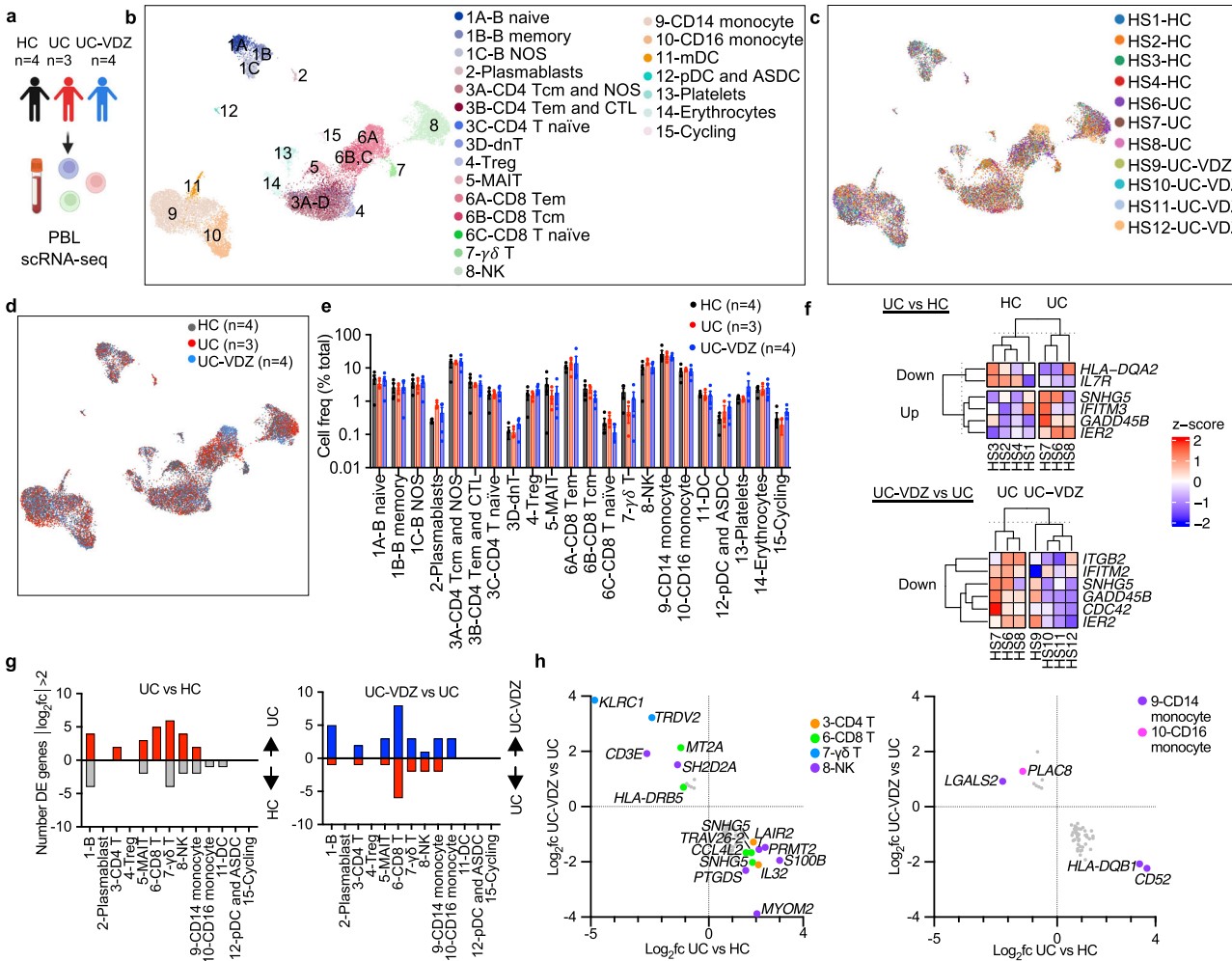

**Fig. 2 | scRNA-seq of peripheral blood leukocytes (PBLs) reveals correlation of VDZ with DE genes, but not circulating leukocyte subset frequency.**
**a** Schematic of PBL scRNA-seq. Created with BioRender.com. UMAP visualization of pooled multiplex scRNA-seq for 20,130 PBLs from HC (*n* = 4), UC (*n* = 3), and UC-VDZ (*n* = 4) patients highlighting (**b**) fine cell annotations, (**c**) patient identity, and (**d**) disease and treatment status. **e**, Cell frequency as a percent of total cells per study subject stratified by disease and treatment status (mean ± SEM; n = number of patients; each dot represents one patient sample; multiple one-way ANOVA Kruskal-Wallis test with FDR correction; q < 0.1 threshold for discovery, only significant differences are indicated). **f** Heatmap of expression z-scores for scRNA-seq

DE genes for all PBLs with log₂fc > 1 or <-1 and Bonferroni-corrected *p* value < 0.1 comparing UC (Up/Down) relative to HC, and UC-VDZ (Up/Down) relative to UC identified by MAST analysis. **g** Number of scRNA-seq DE genes in the indicated PBL subsets with log₂fc > 2 or <-2 in UC relative to HC and UC-VDZ relative to UC identified by MAST analysis. **h** scRNA-seq DE genes in the indicated PBL subsets with log₂fc > 1.5 or <-1 in UC versus HC, and an inverse log₂fc for UC-VDZ versus UC log₂fc <-0.5 or >0.5, respectively identified by MAST analysis. For (**f, h**), ribosomal and mitochondrial genes are not displayed. NOS-not otherwise specified; mDC-myeloid dendritic cell; ASDC-AXL⁺ SIGLEC6⁺ myeloid DC pDC-plasmacytoid DC, MAIT mucosal-associated invariant T.

Supplementary Table 2). Recent methods for cryopreserving undigested, intact intestinal mucosal biopsies have revolutionized tissue multi-omics pipelines by facilitating batch processing[16]. Batch processing of samples for scRNA-seq reduces cost and improves inter-sample comparison by minimizing batch effects[21]. To compare the performance of cryopreserved and non-frozen intestinal biopsies in scRNA-seq, we compared scRNA-seq of biopsies from two donors that were frozen briefly in cryopreservation media (Cryopreserved) to samples that were stored briefly on ice (Fresh) (Fig. 1b). Fresh and Cryopreserved colon biopsies yielded similar coarse cell clusters (Fig. 1c, d), with no statistically significant differences in immune, epithelial, and stromal cell subset frequencies (Fig. 1e). Cryopreservation reduces or eliminates granulocytes, and mast cells were decreased with cryopreservation (Fig. 1e), though not statistically significant. Differentially expressed (DE) gene analysis identified up-regulation of heat-shock proteins (*HSPA1A, HSPA1B, HSP90AA1*, and *HSPD1*) in Fresh biopsies as well as up-regulation of *JUN* and *FOSB* in Cryopreserved biopsies (Fig. 1f). Reasonable concordance between freshly processed

and cryopreserved intestinal biopsies, combined with the logistic, financial, and batch processing benefits of cryopreserved biopsies, favors cryopreservation.

## VDZ is associated with modest transcriptional changes in peripheral leukocyte subsets
We hypothesized that patients on VDZ would exhibit increased circulating inflammatory lymphocytes due to inhibition of intestinal trafficking. To test this, we performed scRNA-seq on peripheral blood leukocytes (PBLs) from HC, UC, and UC-VDZ patients (Fig. 2a). Cryopreserved PBLs were thawed, pooled, run in a single batch, and deconvoluted with freemuxlet/demuxlet[21], minimizing batch effects. The anticipated coarse (Supplementary Fig. 1a) and fine leukocyte subsets (Fig. 2b) were identified based on landmark genes expressed for each cluster (Supplementary Fig. 1b–e), in agreement with previously described gene sets[22,23]. Granulocytes were not observed, likely due to poor viability after freezing and thawing. Cells from each patient were mostly distributed across coarse and fine clusters

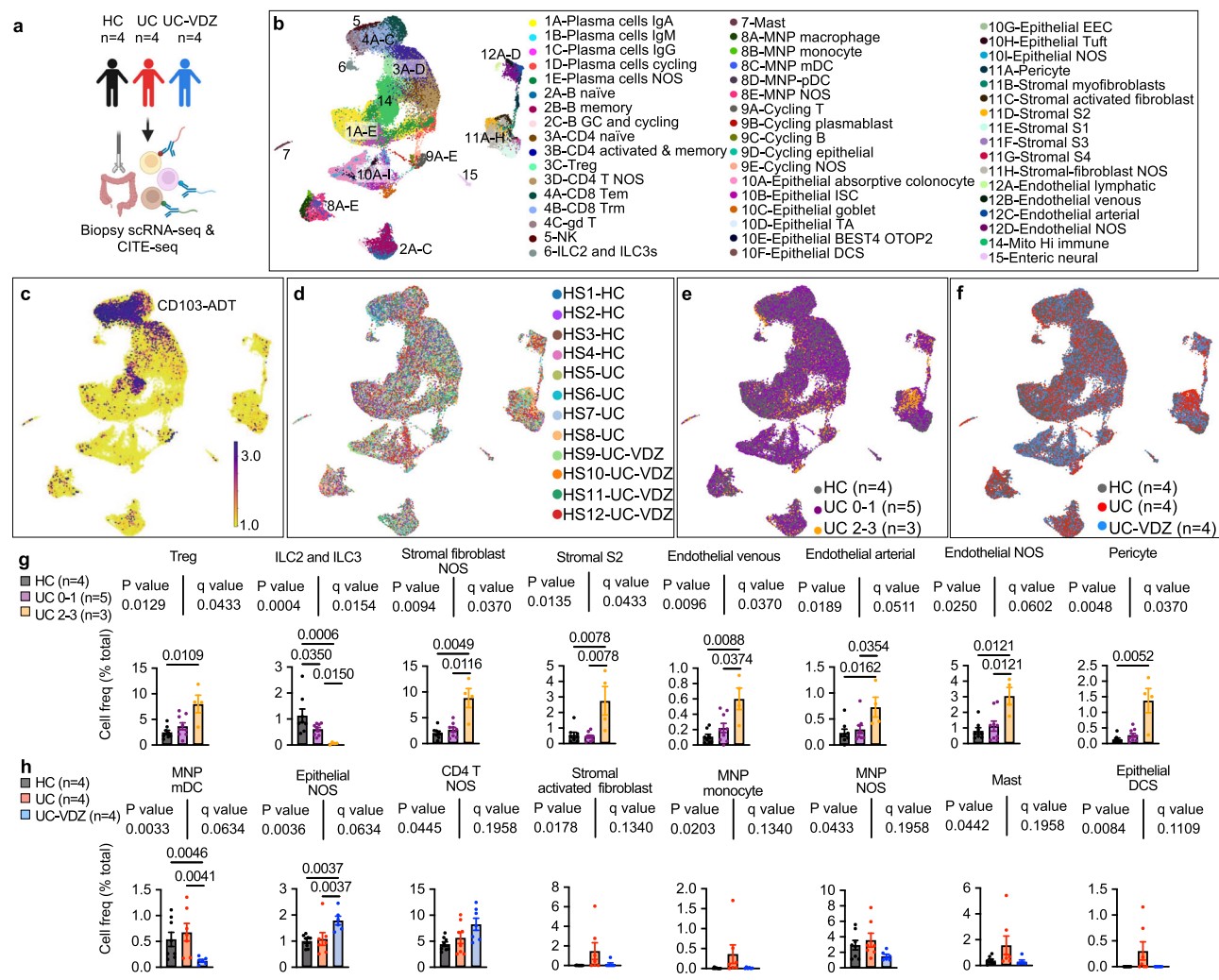

**Fig. 3 | scRNA-seq and CITE-seq of mucosal biopsies highlighted multiple immune and non-immune subsets correlating with inflammatory severity, disease status, and VDZ treatment. a**, Schematic of scRNA-seq and CITE-seq of mucosal biopsies. Created with BioRender.com. **b-f**, UMAP visualization of 93,900 cells from HC ($n = 4$), UC ($n = 4$), and UC-VDZ ($n = 4$) patients highlighting (**b**) fine cell subset annotations (**c**) representative CITE-seq CD103 antibody-derived tag (ADT), (**d**) patient identity, (**e**) endoscopic severity scores, and (**f**) disease and treatment status. Cell frequency for the indicated fine cell subset, expressed as a percent of total cells per study subject, stratified by (**g**) endoscopic severity and (**h**)

disease and treatment status (mean ± SEM; n = number of patients; each dot represents one biopsy location, up to two locations were biopsied per patient; multiple one-way ANOVA Kruskal-Wallis tests with FDR correction; q < 0.1 threshold for discovery; select subsets are shown with exact *p*-value and q-value; individual inter-column q-values are displayed only for cell subsets with overall q < 0.1, an additional nested one-way ANOVA test was performed treating biopsies as replicates, with unadjusted *p* < 0.05 as an additional threshold for discovery). NOS not otherwise specified.

(Fig. 2c), as well as disease and treatment status (Fig. 2d). HS12 had an expanded circulating cytotoxic lymphocyte population, but this was not observed in other UC-VDZ patients (Fig. 2b, c). When we compared the cell frequency for each fine leukocyte cell subset across conditions, we did not observe any statistically significant differences among patients (Fig. 2d,e). scRNA-seq DE gene analysis revealed up-regulation of *IFITM3* in UC patients compared to HC, and down-regulation of *IFITM2* in UC-VDZ patients compared to UC patients (Fig. 2f). DE gene analysis of individual leukocyte subsets identified a small number of differences in lymphocyte and monocyte subsets (Fig. 2g, Supplementary Fig. 2a, b). VDZ was associated with an increase in *ITGB1* in CD8 T cells (Supplementary Fig. 2b). VDZ was also associated with increases in *TMEM176A* and *TMEM176B* in circulating CD14⁺ monocytes (Supplementary Fig. 2b). *TMEM176B* has been suggested to inhibit the inflammasome and dendritic cell maturation[24–26]. To further identify pathways that were potentially pathogenic in UC, and targeted or reversed by VDZ, we then filtered for DE genes that were up-regulated

in UC vs HC, and reciprocally down-regulated in UC-VDZ vs UC, and vice versa (Fig. 2h). This analysis highlighted up-regulation of *CCL4L2* in CD8 T cells, as well as up-regulation of *CD52 and HLA-DQB1* in CD14⁺ monocytes in UC as compared to HC, all with reciprocal down-regulation in UC-VDZ patients. Although scRNA-seq identified some global and cell subset specific transcriptomic associations with UC and VDZ therapy, in general circulating leukocyte subset frequencies were relatively stable with modest transcriptional differences.

**Tissue CITE-seq identifies shifts in multiple mucosal cell subsets associated with VDZ**

To examine the alterations in cell subset abundance and expression programs associated with UC and VDZ therapy, we performed scRNA-seq and CITE-seq on colonic mucosal biopsies from the same patients (Fig. 3a, Supplementary Table 2). Cryopreserved biopsies were thawed, digested, pooled, run in a single batch, and deconvoluted with free-muxlet/demuxlet[21], minimizing batch effects. The anticipated coarse

(Supplementary Fig. 3a) and fine colonic mucosal subsets (Fig. 3b) were identified using landmark genes associated with each cluster (Supplementary Fig. 3b and Supplementary Figs. 4, 5). The subsets identified using cryopreserved biopsies were similar to those previously described for fresh biopsies[12,13,15,27]. The CITE-seq panel provided a human intestinal cell surface proteome in HC, UC, and UC-VDZ patients, and for example, highlighted CD103 (αE integrin) expression by CD8 tissue resident memory T cells (Trm) (Fig. 3c, Supplementary Fig. 4h). CITE-seq identified sets of surface proteins associated with each intestinal cell subset (Supplementary Figs. 3–5). Cells from each patient were distributed across intestinal cell clusters (Fig. 3d). Biopsies were categorized as HC (gray), mild endoscopic disease activity (UC 0–1, purple), or moderate-to-severe endoscopic disease activity (UC 2–3, orange), or as coming from HC (gray), UC (blue), or UC-VDZ (red) patients (Fig. 3e–h, Supplementary Fig. 6). After correcting for multiple comparisons, there were significant increases in Tregs, S2 fibroblasts, pericytes, and endothelial cell subsets, with reductions in innate lymphoid cells (ILCs) and activated and memory CD4+ T cells in more severely inflamed segments (Fig. 3g, Supplementary Fig. 6a, b). The relative frequencies of activated and memory CD4+ and CD8+ T lymphocyte subsets vary among studies depending on inflammatory status and disease activity[15,28], but increases in Tregs have been consistently reported[29–33]. Alterations in ILC3s have also been previously described in IBD scRNA-seq[34]. The relative increase in goblet cells we observed was likely due to a relative reduction in absorptive colonocytes and intestinal stem cells (ISCs) (Supplementary Fig. 6a, b), and variations in goblet cell abundances and maturation have been reported in other IBD transcriptomic studies[35,36]. We observed a global expansion of both venous and arterial endothelial cells in more severely inflamed UC biopsies (Fig. 3g, Supplementary Fig. 6a, b). Previous studies have characterized stromal cell subsets in IBD[12], and here CITE-seq highlighted a general shift from S1/3/4 stromal fibroblasts to increased S2 and activated fibroblasts in more severely inflamed UC biopsies. The activated fibroblasts identified in our dataset expressed high levels of *TIMP1, MMP1, MMP3, AREG, TMEM158, TNFRSF11B*, and surface CD10, while the S2 fibroblasts expressed high levels of *F3, POSTN, CXCL14, ENHO, PDGRFA, SOX6*, and surface CD10, CD146, and CD49a (Supplementary Fig. 5c, d), sharing some features with the *IL13RA2+IL11+TNFRSF11B+* inflammatory fibroblasts associated with anti-TNF resistance[15]. Interestingly, VDZ was not associated with significant reductions in lymphocyte subsets compared to UC biopsies. However, VDZ was associated with a statistically significant reduction in myeloid dendritic cells (mDCs), which were high in *CD1C* gene expression, as well as surface CD1c, CD11c, and FcεR1α (Fig. 3h, Supplementary Fig. 4j). VDZ correlated with trends toward fewer activated fibroblasts, monocytes, macrophages, and mast cells (Fig. 3h, Supplementary Fig. 6c). In the epithelial compartment, VDZ was associated with a statistically significant increase in epithelial cells generally, with a trend toward fewer deep crypt secretory (DCS) cells (Fig. 3h), which could be consistent with a mucosal healing effect. Although VDZ did not appear to significantly alter lymphocyte populations in colonic biopsies by CITE-seq, subsets of inflammatory MNPs and stromal cells trended lower in UC-VDZ patients, with a concomitant expansion of some epithelial subsets.

Analysis of gene expression in tissue CITE-seq revealed significant transcriptomic changes associated with HC, UC, and UC-VDZ patients. scRNA-seq DE gene analysis for all cells revealed an increase of *TIMP1* and *CD74* in UC as compared to UC-VDZ (Supplementary Fig. 7a). Both *TIMP1* and *CD74* were increased in a meta-analysis of UC colonic biopsy bulk gene-expression[37], and these genes were expressed at high levels in endothelial, stromal, and MNP subsets in our dataset. More DE genes were observed in biopsy cell subsets compared to peripheral blood. Stromal fibroblasts and endothelial cells exhibited the highest number of DE genes among the study subjects, as well as MNPs, mast cells, and cycling cells (Supplementary Fig. 7b). Although VDZ is frequently

discussed in terms of its effect on lymphocyte trafficking, non-lymphoid subsets generally exhibited more dynamic DE genes than lymphocyte subsets (Supplementary Figs. 7, 8). MNPs from UC patients expressed higher levels of *TIMP1, SOD2, TYMP, C15orf48*, and *CD63* compared to HC, all of which were reciprocally decreased in UC-VDZ patients (Supplementary Fig. 7c, 8a, b). Mast cells expressed multiple inflammatory genes at higher levels in UC-derived as compared to HC-derived biopsies, but VDZ did not appear to antagonize these signatures (Supplementary Fig. 7c and 8a, b). Stromal fibroblasts in UC patients exhibited elevated levels of *MMP3, TIMP1, TMEM158, COL6A3*, all of which were previously reported to be increased in bulk UC biopsies[37] (Supplementary Fig. 5c; Supplementary Fig. 7c). This activated signature was reciprocally downregulated by VDZ (Supplementary Fig. 8b). Endothelial cells expressed higher levels of *TIMP1, MGP, S100A6, TPM4, TM4SF1, CD59, PRKCDBP*, and lower levels of *TXNIP* and *FABP5* in UC vs HC samples (Supplementary Fig. 7c and 8b). These endothelial cell DE genes were consistent with trends observed in bulk UC biopsies[37], and all were reversed by VDZ (Supplementary Figs. 8a, b). Epithelial cells from UC patients expressed increased levels of *LCN2* and lower *FABP1*, correlating with a relative reduction of absorptive colonocytes in UC compared to HC and reversed by VDZ, consistent with recovery of absorptive colonocytes (Supplementary Fig. 5a and 7c, 8b). Taken together, DE gene analysis identified relatively few transcriptional differences in lymphocyte subsets among HC, UC, and UC-VDZ patients, with more dynamic transcriptional changes in MNPs, stromal fibroblasts, and endothelial cells. The scRNA-seq and CITE-seq data from colonic mucosal biopsies indicates that VDZ may play a role in attenuating MNP trafficking and activation, diminishing activation of inflammatory fibroblasts and endothelial cells, thus facilitating the recovery of intestinal epithelial cells.

## Unsupervised and supervised CyTOF analysis confirms VDZ association with changes in innate immune and epithelial compartments

To further explore the effects of VDZ on various cell populations, and to validate scRNA-seq and CITE-seq data with an orthogonal multi-omics technique, we performed CyTOF on paired PBLs and colon biopsies collected and processed in parallel from HC, UC, and UC-VDZ patients (Fig. 4a, Supplementary Table 2). This complementary multi-omic assay was carried out on the same initial cell preparations as CITE-seq, with the exception that CyTOF biopsy samples did not undergo Annexin-V-based dead cell depletion, and right (R) and left (L) colon biopsies were combined for each patient to increase cell yield. We performed an initial unsupervised analysis using UMAP visualization and clustering[38]. Based on marker intensity, we annotated 21 unsupervised clusters in PBLs (Supplementary Fig. 9a–e) and colon biopsies (Fig. 4b, Supplementary Fig. 9f–i), and clusters with highly similar phenotypes were combined. For unsupervised analysis of PBL CyTOF data, all patients were represented in each cell cluster (Supplementary Fig. 9c, d). CyTOF identified granulocytes in PBLs, representing less than 2% of the total cells after freezing and thawing. After adjusting for multiple comparisons, there were no statistically significant alterations in circulating PBL subsets among the groups (Supplementary Fig. 9e). In the unsupervised analysis of colon biopsy CyTOF data, patient samples were distributed across cell clusters (Supplementary Fig. 9g), with some clusters enriched for HC, UC, or UC-VDZ derived cells (Fig. 4c). The fraction of epithelial cells was higher in CyTOF than CITE-seq, likely because dead cell depletion was not needed prior to the CyTOF analysis. Multiple epithelial subsets emerged from the unsupervised clustering, including epithelial clusters 2/3_EpCAM+CD15hi, 5/6/8_EpCAM+ICOShi, and 9/12_EpCAM+HLA-DR+ (Supplementary Fig. 9h, i), in addition to the expected non-epithelial clusters. Cross-referencing the complementary CITE-seq data suggested that CD15hi epithelial cells are enriched in secretory and transit amplifying epithelial subsets (Supplementary Fig. 5b). CD15hi epithelial cells were

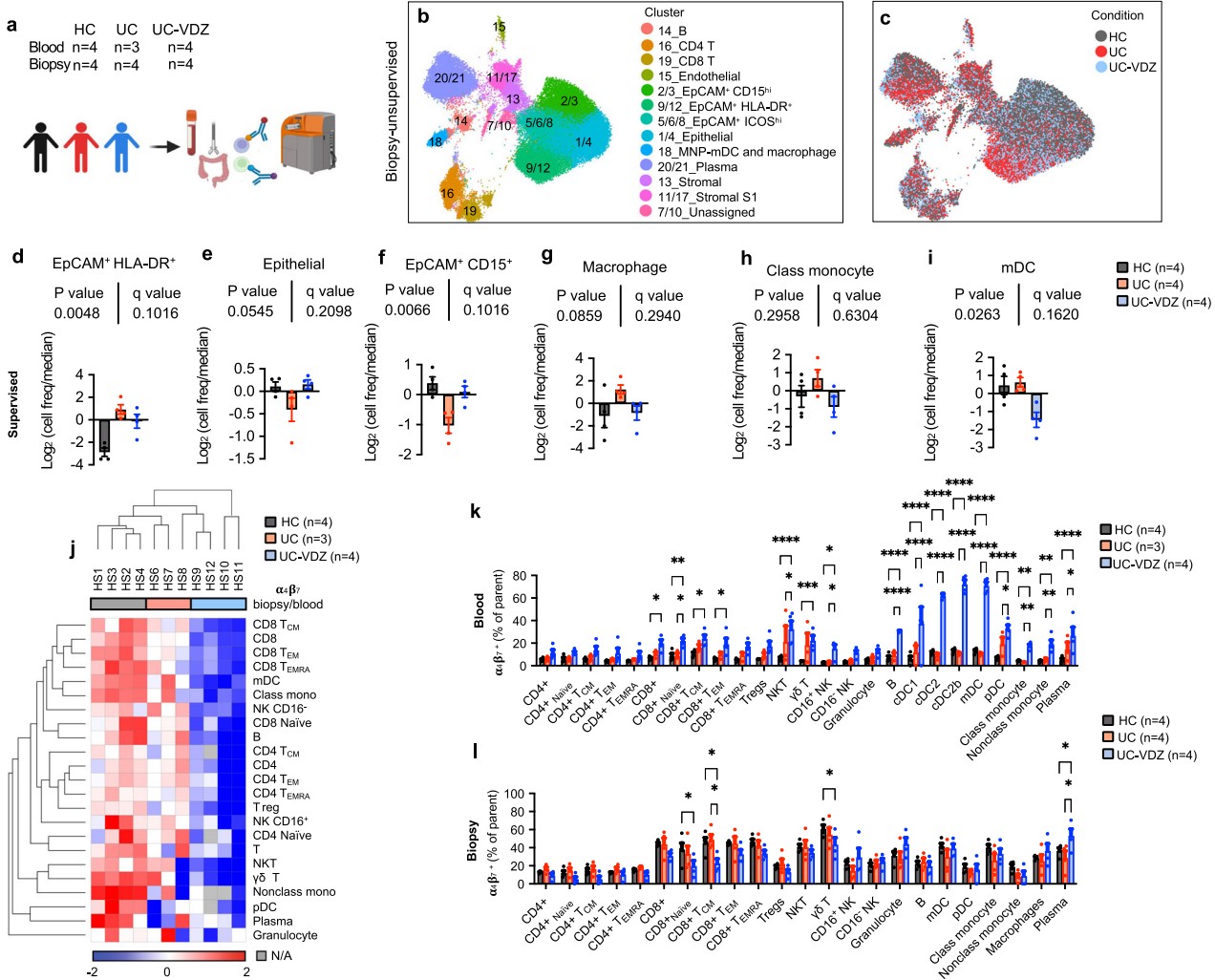

**Fig. 4 | Unsupervised and supervised CyTOF analysis identifies significant increase in circulating α4β7⁺ DCs in UC-VDZ patients. a** Schematic of CyTOF on blood and biopsy samples. Created with BioRender.com. UMAP visualization of the indicated samples (60,000 out of 684,249 live cell events displayed for biopsies) highlighting (**b**) annotated clusters, and (**c**) disease and treatment status. **d–i** Cell frequency of the indicated supervised subset analysis among conditions in biopsies expressed as log₂ (cell freq/median) (mean ± SEM; *n*=number of patients; each dot represents one patient sample; multiple one-way ANOVA Kruskal-Wallis test with FDR correction; q < 0.1 threshold for discovery; individual inter-column q-values are displayed only for cell subsets with overall FDR corrected q < 0.1); the legend for

(**d–i**) is shown in (**i**). **j** Heatmap of biopsy/blood ratio of α4β7⁺ cells for each cell subset by patient (hierarchically clustered by Euclidian distance, average linkage). **k, l** Percentage of α4β7⁺ cells in each defined cell subset per condition for blood and biopsy samples, respectively (mean ± SEM; *n*=number of patients; each dot represents one patient sample; two-way ANOVA comparing HC vs UC-VDZ and UC vs UC-VDZ with FDR correction; q < 0.1 threshold for discovery; *q < 0.05; **q < 0.01; ***q < 0.001; and ****q < 0.0001 (exact q-value are reported in Source Data); only significant differences are indicated. Class mono-classical monocyte; Nonclass mono-nonclassical monocyte; mDC-cDC1,cDC2,cDC2b; N/A-Not Applicable.

significantly reduced in UC compared to HC and increased in UC-VDZ biopsies in the unsupervised analysis (Supplementary Fig. 9i). Unsupervised CyTOF analysis was informative in guiding the supervised analysis and identifying distinct epithelial subsets.

Following unsupervised clustering and analysis, we then performed a supervised gating strategy including the unique epithelial subsets (Supplementary Figs. 10, 11). Supervised analysis did not reveal any statistically significant changes in circulating PBLs by CyTOF (Supplementary Fig. 11a). In contrast, supervised analysis of biopsy CyTOF data demonstrated a borderline significant increase in HLA-DR⁺ IECs in UC and UC-VDZ samples compared to HC, and a trend toward reduced CD15ʰⁱ IECs in UC, with expansion in VDZ-treated patients (Fig. 4d–f). IECs expansion in UC-VDZ compared to UC approached statistical significance, consistent with CITE-seq data (Fig. 3f). In agreement with the unsupervised CyTOF analysis and CITE-seq data, MNP subsets exhibited a trend toward reduction in UC-VDZ patients, including mDCs (cDC1, cDC2, cDC2b), classical monocytes, and

macrophages (Fig. 4g–i, Supplementary Fig. 11b). Since VDZ selectively binds to the integrin α4β7, blocking its interaction with MAdCAM-1, we then looked at the ratio of α4β7⁺ cells in the biopsies relative to the blood for each subset, and found that VDZ significantly decreased α4β7⁺ cells in colon biopsies in the majority (21 of 23) cell subsets to varying extents (Fig. 4j–l, Supplementary Fig. 11c). This pattern was prominent in a hierarchically clustered heatmap, showing unsupervised clustering of UC-VDZ patients HS9-12 based on a lower abundance for all the defined α4β7⁺ cell populations in biopsies relative to blood (Fig. 4j). Therefore, VDZ does broadly interfere to some extent with intestinal trafficking for many cell types, even if frequencies of MNP subsets were more significantly affected than CD4⁺ T lymphocyte subsets. To understand this process further, we evaluated the percent of each α4β7⁺ subset in the peripheral blood and tissue samples (Fig. 4k, l). mDCs, including cDC1, cDC2, and cDC2b, exhibited the largest percent increase in circulating α4β7⁺ cells in VDZ-treated patients (Fig. 4k). α4β7⁺ naïve and central memory CD8 + T cells and γδ

T cells were reduced in UC-VDZ biopsies as compared to HC and UC patient biopsies (Fig. 4l), and reciprocally increased in the circulation (Fig. 4k), but there was no overall reduction in the frequency of these T cell subsets in UC-VDZ patient biopsies (Supplementary Fig. 11b).

To further validate these findings, we performed a replicate CyTOF experiment on a second group of patients in endoscopic remission, or near endoscopic remission, on stable maintenance therapy with VDZ, anti-TNF, or 5-ASA therapy (Supplementary Table 1). In VDZ-treated patients, mDCs again exhibited the most pronounced increase in the circulating $\alpha_4\beta_7^+$ fraction (Supplementary Fig. 12a). There were no differences in the overall PBL cell subset frequencies (Supplementary Fig. 12b). These results confirmed that the higher levels of circulating $\alpha_4\beta_7^+$ mDCs observed in VDZ-treated patients persisted during stable maintenance therapy, and when compared against anti-TNF agents. CyTOF of mucosal biopsies did not reveal a significant reduction in mucosal mDCs of VDZ-treated patients compared to HC or UC patients on other therapies, which could be explained by the later timepoint and endoscopic remission of this group (Supplementary Fig. 12c, d). There was an apparent reduction in total naïve CD4$^+$ and CD8$^+$ T cells in VDZ-treated UC patients compared to other therapies, although the results did not reach statistical significance (Supplementary Fig. 12d). Lastly, when we looked at the ratio of $\alpha_4\beta_7^+$ cells in the biopsies relative to the blood for each subset, we confirmed that VDZ significantly reduced $\alpha_4\beta_7^+$ cells in colon biopsies for 18 of 23 cell subsets (Supplementary Fig. 12e, f). Overall, CyTOF experiments confirmed that VDZ is associated with a shift of $\alpha_4\beta_7^+$ cells from tissue to peripheral blood across most subsets in either active or inactive UC. Notably impacted were the circulating $\alpha_4\beta_7^+$ mDCs, followed by CD8 + T, NK, γδ T, B, and plasma cells.

### Spatial analysis of MNP and fibroblast subsets in FFPE samples from VDZ-treated patients

We used a variety of spatial transcriptomic and proteomic methods with single-cell resolution to evaluate the colonic tissue microenvironment across disease and treatment status in FFPE samples. 12-plex RNA-ISH exhibited sufficient sensitivity and specificity for some high-expressing cell lineage markers, but struggled to adequately measure genes expressed at lower levels, including *ITGA4*, *ITGB7*, *MADCAM1*, and *FLT3* (Supplementary Fig 13a, b, Supplementary Table 3). 39-plex MIBI on the same tissue microarray (TMA) identified 20 phenotypes with high signal-to-noise ratio (SNR), but the need to balance resolution and field-of-view (FOV) dimensions with longer acquisition times ultimately yielded fewer cells than RNA-ISH (Fig. 5a–c, Supplementary Table 3,4). Despite limited cell counts with MIBI, there was a trend indicating a reduction in fibroblasts in UC-VDZ compared to UC biopsies (Fig. 5d; Supplementary Fig. 13c). We then performed 28-plex CODEX and unsupervised clustering, visualization, and annotation, mirroring the CyTOF analysis (Fig. 5e–j; Supplementary Fig. 13d; Supplementary Tables 3, 4). Unsupervised analysis generated 24 clusters, and highly similar clusters were subsequently grouped to yield 11 clusters (Fig. 5f). The UMAP, similarity matrix heatmap, along with visual inspection of each channel, identified specific and nonspecific markers for the final supervised cell phenotype analysis of 18 subsets (Fig. 5g, h; Supplementary Fig. 13d, Supplementary Table 4). Memory CD8$^+$ T cells exhibited a statistically significant increase in UC compared to HC, but this was not significantly reduced in VDZ-treated patients (Supplementary Fig. 13d). While the CODEX panel lacked the specificity to accurately quantify mDCs, it could reliably detect and quantify both macrophages and CD44$^+$PDPN$^+$ activated/inflammatory fibroblasts. While not statistically significant, both subsets showed a trend toward increase in UC and reduction with VDZ (Fig. 5i, j), similar to the pattern observed with CITE-seq. In both MIBI and CODEX datasets, MNP and fibroblast subsets exhibited trends toward increased proximity by nearest neighbor (NN) analysis in UC compared to HC, with inhibition VDZ (Fig. 6a, b).

To further evaluate the colonic tissue microenvironment in this setting, we then performed 960-plex RNA-ISH (CosMx from Nanostring) on our FFPE TMA at subcellular resolution, as recently described[39,40] (Fig. 6c). After cell segmentation and mapping of transcript location (Fig. 6d; Supplementary Fig 13e–g), UMAP and corresponding spatial scatter plots (Fig. 6e, f) were used to manually annotate cell types. Landmark genes for each subset correlated well with biopsy scRNA-seq (Supplementary Fig. 13g). Activated fibroblasts (*TIMP1, IL1R1, CXCL14, CD44*) and activated inflammatory MNPs (*S100A4, TIMP1, S100A9, CD80, ITGAX, LYZ, IL1B*) containing a mixture of inflammatory monocytes, macrophages, and mDCs, trended toward increased spatial proximity in UC biopsies as compared to HC, but not in UC-VDZ patients (Fig. 6g, h).

### Spatial transcriptomics of archived FFPE biopsies identifies pre-treatment signatures associated with VDZ response and non-response

To examine tissue level differences between VDZ responders (VDZ-R) and non-responders (VDZ-NR), we retrieved longitudinal archived FFPE biopsies before and after therapy and performed 1000-plex CosMx spatial transcriptomics (Fig. 7a, Supplementary Tables 1, 3). The data quality using retrospectively identified, clinical archived FFPE samples underperformed prospectively collected FFPE samples, likely reflecting sample age and storage. Approximately 20% of cells were not annotated after filtering, and additional stromal and lymphocyte subsets could not be confidently assigned due to lower levels of landmark gene expression and ambiguous cell identity. Importantly, the myeloid, stromal, and epithelial subsets of interest expressed landmark genes at higher levels. Unsupervised hierarchical clustering based on the abundance of these subsets distinguished HC from UC patient samples (Supplementary Fig 14a). We observed an increase in activated MNPs in UC compared to controls before VDZ treatment, with a decrease in responders and an increase in non-responders post-treatment (Fig. 7b). IECs expressing high levels of MHCII were similarly elevated in active colitis compared to controls before VDZ treatment, with an apparent reduction in responders post-treatment (Fig. 7c). Again, neighborhood enrichment analysis revealed trends toward increased proximity of activated fibroblast and activated MNP subsets in active colitis, and reduction after treatment, although this was not statistically significant and not clearly associated with response or non-response to VDZ (Fig. 7d). Pre-treatment differences are the most relevant for developing precision medicine algorithms. Therefore, we performed pseudobulk DE gene analysis of pre-treatment FFPE biopsies from non-responders versus responders, to identify distinguishing baseline features (Supplementary Table 7, Supplementary Fig 14b). Genes specific to the IEC crypt base including *REG1A, OLFM4, AGR2, SPINK1*, and *LYZ* were associated with response to VDZ, while fibroblast and MNP-enriched genes including *MMP1, MMP2*, and *THBS1* were relatively higher in VDZ non-responders (Fig. 7e). IgA plasma cell-associated genes were also higher in responders (Fig. 7e). Spatial scatter plots of cell subsets and transcripts suggested that a robust IEC crypt base was associated with response to VDZ, while the abundance and activation of fibroblasts and MNPs were more linked to non-response prior to VDZ treatment (Fig. 7f–i).

To further validate the association of MNP, stromal, and IEC genes with VDZ response and non-response, we performed gene set enrichment analysis (GSEA)[41,42] of a longitudinal, publicly available[1], bulk transcriptomic dataset of UC patients using landmark genes from our multi-omics analysis (Supplementary Table 5). Neutrophils were not present in our biopsy CITE-seq data, but a gene set signature was generated from canonical landmark genes. Not surprisingly, VDZ responders ($n = 9$) exhibited broad reductions in immune and activated stromal Normalized Enrichment Scores (NES), with epithelial gene set enrichment post-treatment, consistent with reduced inflammation and mucosal healing (Fig. 8a). In contrast, VDZ non-responders

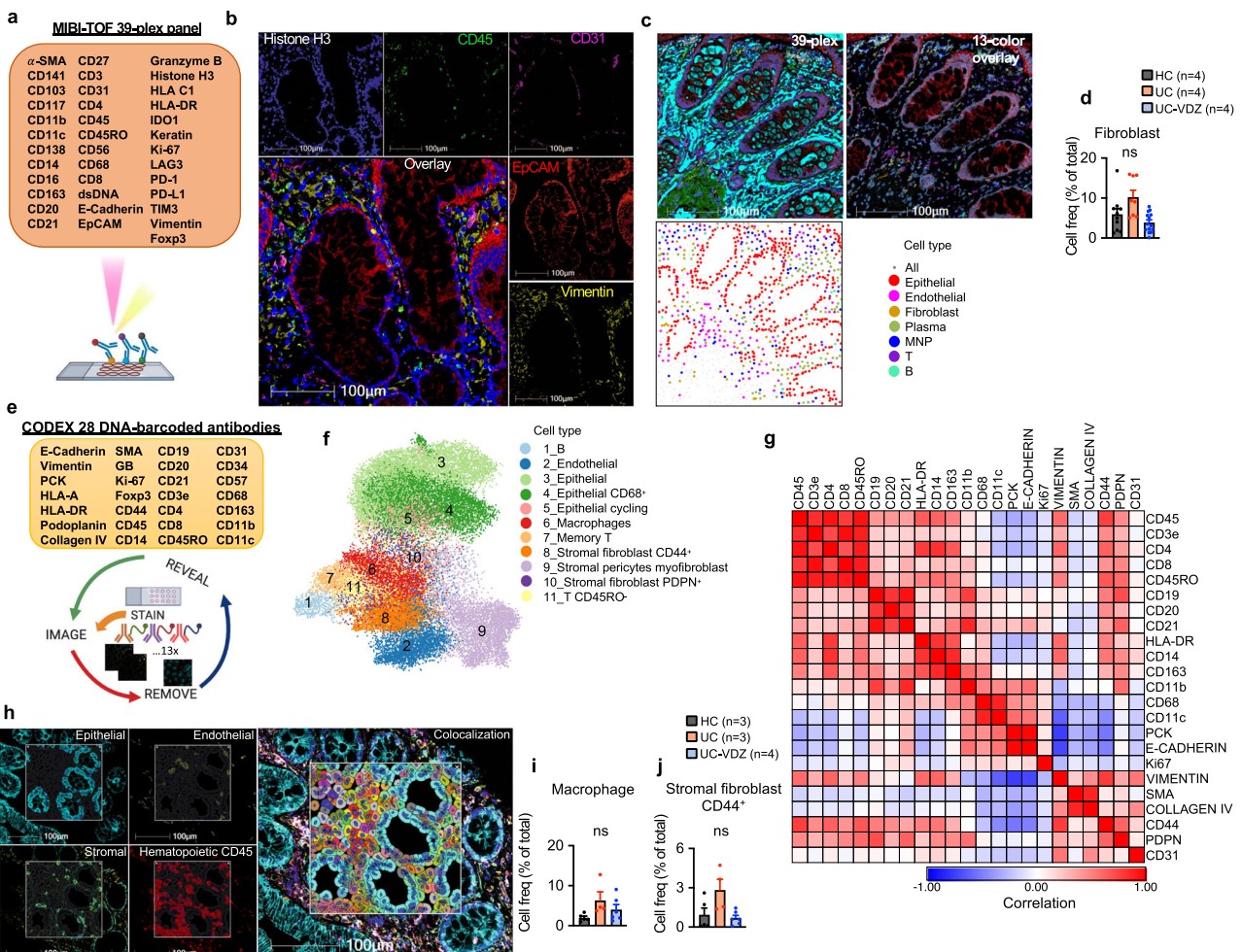

**Fig. 5 | MIBI and CODEX spatial proteomics using FFPE tissue identifies distinct phenotypes in mucosal biopsies of UC-VDZ patients. a** Schematic of MIBI workflow and customized antibody panel. Created with BioRender.com. MIBI images representative of 32 FOVs for (**b**) nuclear DNA, indicated major cell lineage markers, and 5 color overlay, and (**c**) 39-plex overlay, selected channels, and related spatial scatter plots for coarse annotation of the indicated cell subsets. **d** Cell frequency as a percent of total cells detected by MIBI for the indicated cell subset. **e** Schematic of CODEX workflow and antibody panel. Created with BioRender.com. **f** UMAP visualization of 68,804 captured cells (50,000 cells displayed) highlighting annotated clusters. **g** Marker similarity matrix among 23 selected markers (Pearson correlation). **h** CODEX images representative of 15 cores, phenotype identification highlighting indicated markers and major phenotype colocalization. **i, j** Cell frequency as a percent of total cells detected by CODEX for the indicated cell subsets. For panels (**d**),(**i**),(**j**), mean ± SEM; *n*=number of patients; each dot represents one FOV for MIBI or one core for CODEX; multiple one-way ANOVA Kruskal-Wallis test with FDR correction; q < 0.1 threshold for discovery; ns not significant.

(*n* = 5) exhibited a high initial pre-treatment cytotoxic lymphocyte signature, and persistent activated and S2 fibroblast gene signatures post-treatment, without significant epithelial enrichment post-treatment, consistent with a model where high initial cytotoxic lymphocyte injury and persistent stromal tissue inflammation prevents mucosal healing (Fig. 8a). Interestingly, the reduction in immune subsets in VDZ non-responders was smaller and not statistically significant compared to VDZ responders. VDZ non-responders (*n* = 9) were differentiated from responders (*n* = 11) by pre-treatment enrichment for endothelial, activated fibroblast, neutrophil, macrophage, and monocyte signatures (Fig. 8b). Gene signatures investigated were clearly distinguished by leading edge analysis (Fig. 8c). Additionally, GSEA using VDZ response and non-response signatures identified from our longitudinal spatial transcriptomics analysis of FFPE biopsies were validated in this external bulk transcriptomic dataset (Fig. 8d, e; Supplementary Table 5). Interestingly, the VDZ non-response signature was also significantly enriched in Infliximab non-responders prior to treatment (pre-IFX), categorizing these genes as markers of non-response to both treatments (Fig. 8f, g). This is expected as all of the VDZ-treated patients in that study had previously been exposed to anti-TNF therapy[1], similar to our VDZ non-responders (Supplementary Table 1). In contrast, the pre-VDZ-response signature was specific to VDZ, and not associated with response to IFX (Fig. 8f, g). These data suggest that VDZ non-responders have higher pre-treatment tissue innate immune and activated stromal subset inflammation, and that these cell subsets likely drive inflammatory cell trafficking via $\alpha_4\beta_7$-independent pathways. Conversely, a robust IEC crypt base signature pre-treatment is linked to response to VDZ and mucosal healing.

## Discussion

To identify and validate the transcriptomic, proteomic, and cellular signatures associated with UC and VDZ treatment, we optimized a multi-omics pipeline that included batch processing of cryopreserved biopsies for simultaneous multiplexed CITE-seq and CyTOF, coupled with spatial analysis of FFPE biopsies. VDZ was associated with small shifts in peripheral leukocytes by scRNA-seq, while colonic tissue profiling demonstrated a significant reduction in MNPs, expansion of some epithelial subsets, and a trend toward fewer activated fibroblasts by CITE-seq. Among immune subsets analyzed by CyTOF, mDCs exhibited the largest increase in circulating $\alpha_4\beta_7^+$ cells in VDZ-treated

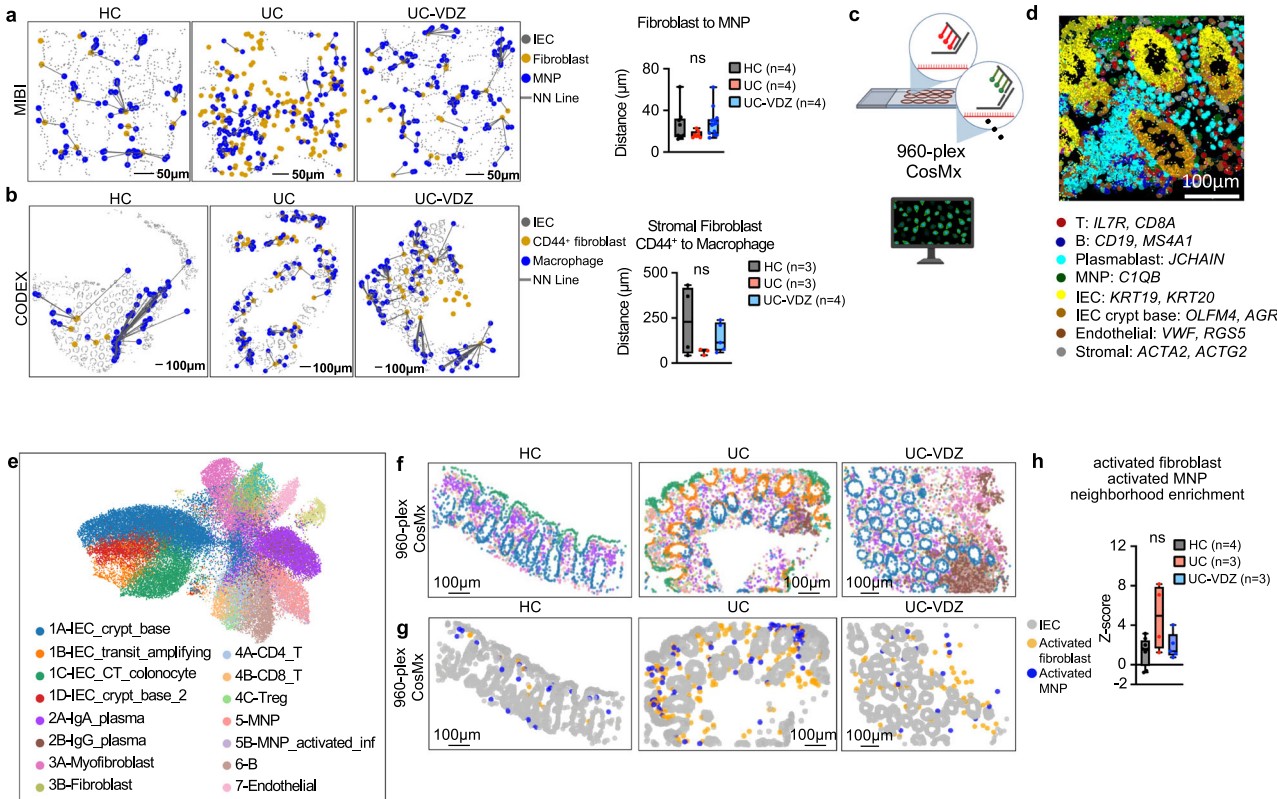

**Fig. 6 | MNP and fibroblast subsets trend toward spatial proximity in UC patients compared to HC.** Spatial scatter plots of the indicated cell subsets and nearest-neighbor (NN) analysis for (**a**) MIBI (representative of 29 FOVs) and (**b**) CODEX (representative of 12 cores); $n$=number of patients; each dot represents one FOV for MIBI or one core for CODEX; one-way ANOVA Kruskal-Wallis tests with FDR correction; q < 0.1 threshold for discovery. **c** Schematic of 960-plex RNA-ISH of FFPE TMA. Created with BioRender.com. **d** CosMx images representative of 17 FOVs, cell segmentation and probe signal for the indicated cells and genes. **e** UMAP visualization of 960-plex CosMx for 48,783 cells from HC ($n$ = 4), UC ($n$ = 3), and UC-

VDZ ($n$ = 3) patients highlighting the indicated cell subsets. **f, g** Representative spatial scatter plots highlighting the indicated cell subsets (spatial scatter plots were representative of 17 FOVs); the legend for (**f**) is shown in (**e**). **h** Z-score of activated fibroblast and activated MNP neighborhood enrichment ($n$=number of patients; each dot represents one FOV; one-way ANOVA Kruskal-Wallis tests with FDR correction; q < 0.1 threshold for discovery; ns-not significant). For panels (**a, b**) and (**h**) box and whisker plots, the band indicates the median, the box indicates the first and third quartiles, and the whiskers indicate minimum and maximum, all points are shown.

patients. Spatial proteomics and transcriptomics of FFPE colonic biopsies using MIBI, CODEX, and highly multiplexed RNA-ISH demonstrated trends toward increased density and proximity of MNP and fibroblast subsets in UC as compared to HC. Spatial transcriptomics of archived clinical FFPE samples before treatment identified MNP, fibroblast, and epithelial gene signatures linked to VDZ response and non-response, and these signatures were confirmed using an external, publicly available bulk transcriptomic dataset.

There are conflicting data regarding the primary cell subsets targeted by VDZ. VDZ has been reported to bind to CD4 T, CD8 T, B, NK, and granulocytes in the peripheral blood[43,44]. Here we confirm that VDZ shifts $\alpha_4\beta_7^+$ cells from the colon to the circulation for most cell types, but this results in a small net change in tissue cell frequency for most cell subsets. Multiple studies have investigated the effect of VDZ on T cell subsets, in general supporting models where pathogenic effector T cell subsets are excluded more efficiently than regulatory T cell subsets[6–8,45]. Our data do not exclude any of these observations, but rather add that VDZ is correlated with alterations in the abundance and expression of circulating and intestinal MNP subsets, which likely contributes to a reduction in tissue inflammation. One recent study demonstrated that VDZ did not consistently alter the phenotype, activation, or repertoire of lamina propria T cells by flow cytometry and TCR sequencing, but bulk transcriptomic data was consistent with a shift in MNP gene signatures[11], aligning with our analysis. Nonetheless, the impact of VDZ on intestinal MNP populations likely affects

the recruitment and activation of both adaptive and innate immune subsets.

This study has important therapeutic implications. Our CyTOF analysis demonstrated that mDCs are highly dependent on $\alpha_4\beta_7$ for intestinal trafficking, consistent with MAdCAM-1-deficient and $\beta7$ integrin-deficient mice[10]. Our data suggest that T cells may be better able to exploit $\alpha_4\beta_7$-independent intestinal trafficking pathways than mDCs. These pathways could include $\alpha_4\beta_1$:VCAM1, GPR15:C10ORF99, CXCR4:CXCL12, or CCR6:CCL20[46]. Interestingly, *GPR15* and *CCR6* are expressed by lymphocytes, but not mDCs or monocytes, which could partially explain the impact of VDZ on tissue MNPs. Refractory IBD patients may benefit from additional combination therapies that target lymphocytes or activated stromal cell subsets, as is done sometimes in clinical practice[47]. Recently, inflammatory fibroblasts have been reported to secrete neutrophil-tropic CXCR1 and CXCR2 ligands in response to IL-1β[48]. Granulocytes are greatly diminished by cryopreservation, so our study was not optimized to analyze neutrophils, but we do show that VDZ non-responders exhibit increased activated fibroblast signatures at baseline and after therapy. The observation that inflammatory MNPs and fibroblasts share similar cellular neighborhoods further supports a rationale for neutralization of IL-1 family members in refractory colitis.

A secondary objective of this study was to evaluate various multiomics platforms on cryopreserved biopsies and FFPE tissue. CyTOF quantitated the most cells per patient, MIBI the fewest, while CITE-seq,

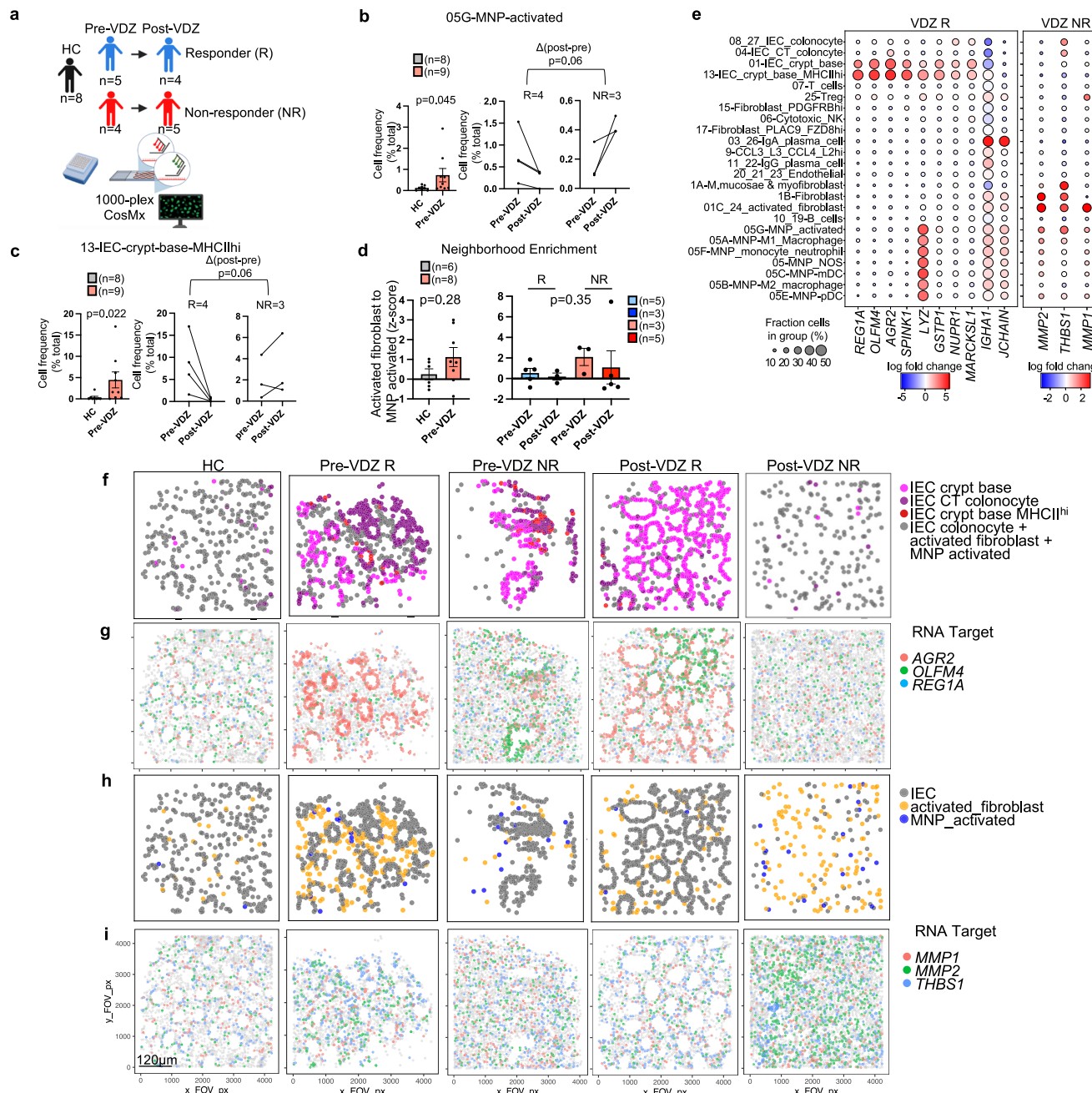

**Fig. 7 | CosMx spatial transcriptomics of archived FFPE specimens with single-cell resolution identified tissue signatures of VDZ response and non-response prior to therapy in activated MNP, fibroblast, and IEC crypt base subsets.**
**a** Schematic of retrospective, longitudinal analysis of archived FFPE specimens using 1000-plex CosMx spatial transcriptomics of 126,368 cells from 73 FOVs; n of schematic applies to left panels in (**b**–**d**). Created with BioRender.com. **b**, **c** Cell frequencies of indicated subsets comparing (left) HC and pre-treatment samples (pre-VDZ) (Mann–Whitney, two-tailed), as well as (middle, right) pre-VDZ and post-VDZ treatment for the indicated subsets for both responders (R) and non-responders (NR), only patients with matching biopsies pre- and post-VDZ are shown (Mann–Whitney, two-tailed of Δ post VDZ - pre VDZ for R and NR). **d** Z-score of activated fibroblast and activated MNP neighborhood enrichment comparing (left) HC and pre-VDZ (Mann–Whitney, two-tailed) and (right) one-way ANOVA Kruskal-Wallis test with Dunn's multiple comparison test. **e** Dot plot representation of a subset of genes from pseudobulk DE gene analysis for the indicated subsets. Representative spatial cell scatter plots highlighting the relevant cell subsets relatively increased in (**f**) VDZ R or (**h**) VDZ NR. Representative spatial transcript scatter plots highlighting a subset of genes relatively increased in (**g**) VDZ R and (**i**) VDZ NR. **f**–**i** Spatial scatter plots were representative of 73 FOVs. For panels (**b**–**d**), mean ± SEM; *n*=number of patients; each dot represents averaged FOV per patient. R-responder; NR non-responder.

CODEX, and CosMx methods yielded comparable numbers of cells per patient (Supplementary Fig 14c). Comparing across tissue compartments, CITE-seq over-sampled the immune compartment and under-sampled the epithelial compartment, when compared to other multi-omics methods (Supplementary Fig 14d, Supplementary Table 6). Given the ubiquity of FFPE tissue, preservation of spatial relationships, and more accurate representation of in situ cell frequencies and gene

expression, FFPE-compatible spatial multi-omic technologies provide a powerful complementary method for analyzing patient-derived biospecimens.

There are several important limitations of this study. No functional investigations were performed, therefore alterations in cell abundance or gene expression could be directly or indirectly related to VDZ. Additional limitations include a small sample size and case-

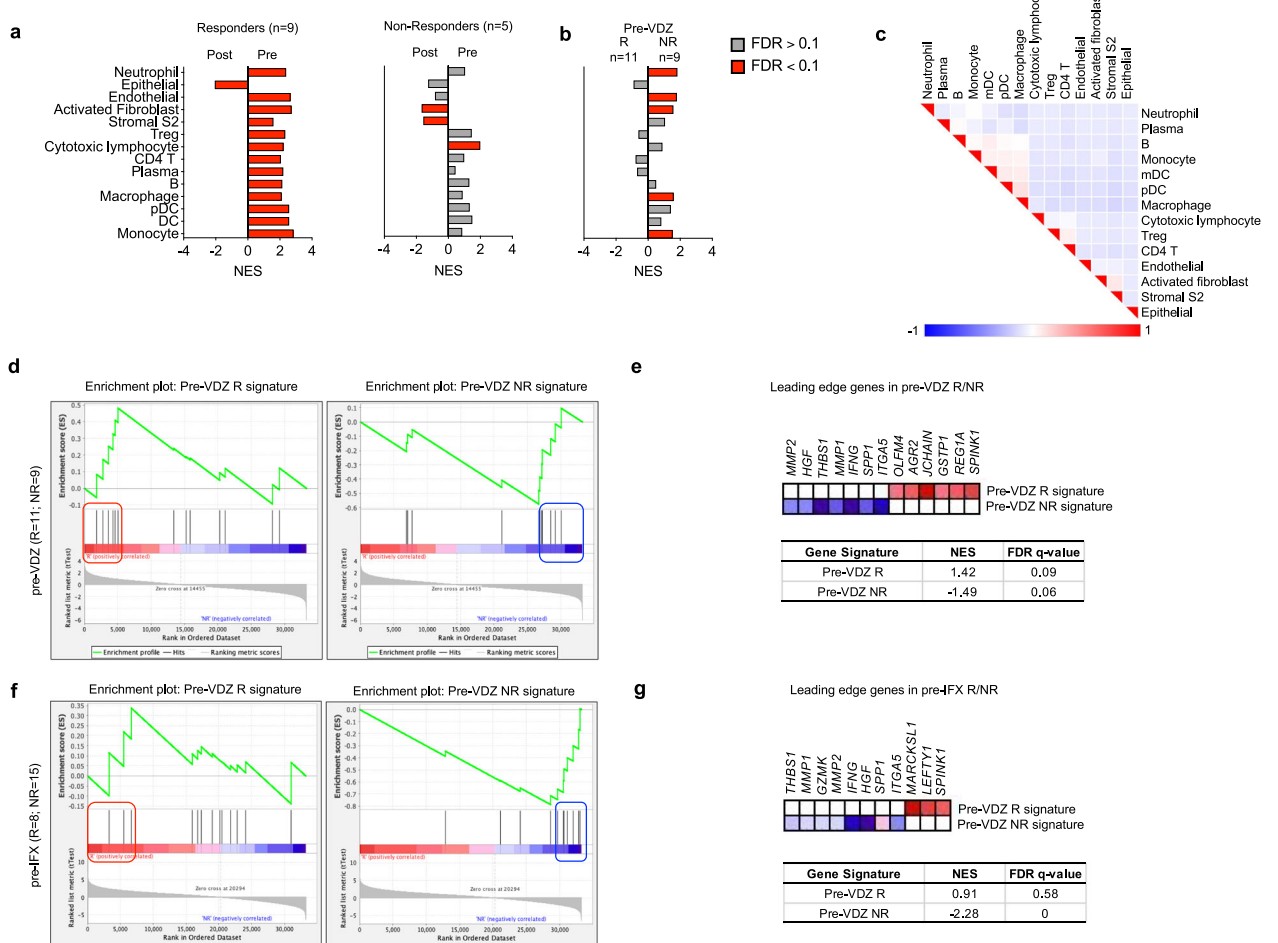

**Fig. 8 | Gene set enrichment analysis (GSEA) of an external, publicly-available, bulk transcriptomic dataset (GSE73661) using cell subset and spatial transcriptomic signatures associated with response and non-response to VDZ.** Normalized Enrichment Scores (NES) in bulk tissue transcriptomic data comparing (**a**) pre- and post-treatment samples for VDZ responders (R) and non-responders (NR), (**b**) pre-VDZ R vs NR (red bars FDR < 0.1, gray bars FDR > 0.1). **c** Leading edge analysis of significantly enriched gene sets. **d**–**g**, GSEA of VDZ response and non-response spatial signatures in external cohort of patients (**d**) pre-VDZ and (**f**) pre-IFX. Subset of genes comprising the leading edge of the NES and FDR q-values in (**e**) pre-VDZ and (**g**) pre-IFX patients, respectively. IFX infliximab.

control approach for the primary analysis. We chose to perform in-depth multi-omic analysis in the same patients to maximize paired phenotypic data from each patient. In the future, prospective longitudinal single-cell and spatial multi-omic data from more patients on diverse therapies could permit detection of more subtle changes in differential gene and protein expression across various medication classes. Of note, the VDZ-non-responders in our study and the publicly available dataset were both previously exposed to anti-TNF, likely explaining why our VDZ non-response signature was associated with both VDZ and IFX non-response. While we found that archived clinical FFPE samples are suitable for spatial transcriptomics, the data are noisier, with smaller fold changes than sequencing methods, and some variability in probe performance and specimen integrity. Finally, this study focused exclusively on host factors, but microbial determinants and metabolomics have also been shown to contribute to therapeutic response in IBD[49,50]. Incorporating additional multi-omic modalities in future studies could enhance our understanding of treatment response and non-response in UC.

In summary, we performed comprehensive single-cell and spatial transcriptomic and proteomic phenotyping to establish MNPs as an important cell type impacted by anti-integrin therapy in UC, with associated changes in stromal and epithelial populations. This study highlights important cellular networks involving MNPs and fibroblasts

in colitis. The combination of CITE-seq and CyTOF on identical sets of biopsies establish a surface protein cell atlas for the colon in health, disease, and during treatment. We also describe a spatial atlas for colitis with single-cell resolution using MIBI, CODEX, and CosMx on FFPE samples, allowing comprehensive analysis of cell subset frequency, differential gene expression, and cellular proximity. Ultimately, precision medicine implies approaching each patient as an n-of-1, and here we show that multiple orthogonal multi-omics analyses enhance internal validity for immunophenotyping, even with small sample sizes. Implementing single-cell and spatial multi-omics methods simultaneously in individual patients will provide deep immunophenotyping and lead to more precise treatment algorithms.

# Methods
## Study approval
The study was conducted according to Declaration of Helsinki principles and was approved by the Institutional Review Board of the University of California, San Francisco (19-27302). Written informed consent was received from participants prior to inclusion in the study.

## Study participants and biospecimen collection
For prospective sample collection, patients undergoing colonoscopy or sigmoidoscopy for standard of care indications were screened for

study eligibility. Eligible patients were recruited consecutively to minimize self-selection bias. Patients were compensated $50 for each sample collection event. All participating patients gave written informed consent and approval. Peripheral blood and cold forceps biopsy samples were obtained from patients with UC, and individuals without IBD, referred to as healthy controls (HC). For retrospective archived FFPE sample retrieval, study subjects were identified by querying the electronic medical records of patients previously seen by UCSF Gastroenterology, followed by written informed consent and approval. Baseline demographic and clinical data for the study participants are provided in Supplementary Table 1. We have consent to publish de-identified patient demographics including age at the time of sample collection, sex, diagnosis, and medical center. Demographic options were defined by the investigators and participants chose their classifications. HC patients were patients without known or suspected IBD undergoing elective colonoscopy or sigmoidoscopy for various indications (e.g., colorectal cancer screening). Biopsy samples were categorized as coming from an area that was endoscopically non-inflamed (score=0), mildly inflamed (score=1), moderately inflamed (score=2), or severely inflamed (score=3). Samples were assigned unique identifiers before biobanking.

### Sample collection and storage

Colon biopsies were obtained with standard cold endoscopic biopsy forceps from two different regions of the colon (R=right/proximal and L=left/distal) with separate vials for different downstream applications. Two biopsies were collected in 10% formalin in 5 mL tubes for 24 h, then washed with PBS twice and stored in 70% ethanol for paraffin embedding; two biopsies were collected in 4 mL RNAlater in 5 mL tubes stored overnight at 4 °C for, then the solution was aspirated, and biopsies were stored at −70 °C until further analysis. Six biopsies were collected in a conical tube with Basal Media (Advanced DMEM/F12 with NEAA and Sodium Pyruvate, Thermo cat. No. 12634-010; 2 mM Glutamax, Thermo cat. No. 35050061; 10 mM HEPES (Corning); Penicillin-Streptomycin-Neomycin (PSN) Antibiotic Mixture, Thermo cat. No. 15640055; Normocin 100 μg/mL, Invivogen cat. No. ant-nr-2; 1 mM N-acetylcysteine, Sigma-Aldrich, A9165) with 10 μM Y-27632 (MedChem Express) at 4 °C. Samples were immediately placed on ice and transported to the laboratory for processing as previously described[51]. Biopsies were transferred into cryovials containing freezing media (90% (v/v) FCS, 10% (v/v) DMSO and 10 μM Y-27632) and immediately placed into a freezing container (Mr. Frosty or Coolcell) and stored at −70 °C for up to 4 weeks before transferring to liquid nitrogen cryostorage until further processing. A PAXgene RNA tube (Qiagen) for peripheral blood was collected, stored, and processed according to the manufacturer's instruction. For peripheral blood leukocyte (PBL) isolation, peripheral blood was collected into EDTA tubes (BD, 366643). 2 mL aliquots of peripheral blood were treated with 30 mL of RBC lysis buffer (Roche) for 5–8 min at room temperature with gentle mixing. Cells were re-suspended in CryoStor CS10 (at 4 °C) freezing medium, aliquoted in cryovials, transferred to a freezing container (Mr. Frosty or Coolcell), and stored at −70 °C for up to 4 weeks before transferring to liquid nitrogen cryostorage until further processing.

### Preparation of colon and peripheral blood single-cell suspensions

Colon biopsies were thawed for 2 min with gentle agitation in a 37 °C water bath, transferred to a gentleMACS C Tube (Miltenyi Biotec), washed twice with basal media containing 10 μM Y-27632, and then incubated in 5 mL digestion buffer (basal media), 10 μM Y-27632, 600 U/mL Collagenase IV (Worthington cat. No. LS004189), 0.1 mg/mL DNAse I (Sigma-Aldrich, D4513) and digested for 20 min at 37 °C in a shaking incubator set at 225 rpm. Subsequently, samples were placed in the gentleMACS Dissociator, processed with the gentleMACS program m_intestine_01, followed by 15 min incubation at 37 °C in a shaking incubator set at 225 rpm. The suspension was then strained through a 100 μm strainer (Miltenyi) and centrifuged at 450 g for 5 min at RT. Two additional washes were performed in Hanks' Balanced Salt Solution (HBSS) (Corning), containing 0.1 mg/mL DNAse I (Sigma-Aldrich, D4513). $1 \times 10^6$ total cells were set aside for CyTOF. For the remaining cells, dead cells were removed with the Dead Cell Removal Kit (Miltenyi) according to the manufacturer's instructions. Cell suspensions were counted using a TC20 Automated Cell Counter (Bio-Rad) with 0.4% Trypan Blue Solution (Thermo Fisher Scientific). Live-cell enriched colon single-cell suspensions were used for tissue scRNA-seq and CITE-seq, with a final pooled viability >75%. PBLs were thawed for 2 min with gentle agitation in a 37 °C water bath and then washed twice with complete DMEM (Thermo Fisher) supplemented with nonessential amino acids (Thermo Fisher Scientific), sodium pyruvate (Thermo Fisher Scientific), HEPES (10 mM; Corning), Glutamax (2 mM; Thermo Fisher Scientific), Normocin (100 μg/mL; Invivogen, ant-nr-2), penicillin-streptomycin (Thermo Fisher Scientific) and 10% Fetal Bovine Serum (VWR). Cells were incubated with ACK lysis buffer (Quality Biological) for 5 min at room-temperature, washed twice with complete DMEM, treated with HBSS (Corning) containing 0.1 mg/mL DNAse I (Sigma-Aldrich, D4513) for 5 min, and then strained through a 20um pre-separation filter (Miltenyi). Cells were counted using a TC20 Automated Cell Counter (Bio-Rad) with 0.4% Trypan Blue Solution (Thermo Fisher Scientific). Peripheral blood leukocytes (PBLs) from each donor were used for scRNA-seq and CyTOF, with a final pooled viability of >85%.

### Bulk RNA-seq sample and computational processing

RNA was extracted from blood or biopsies following manufacturer's protocol using the PAXgene kit and Qiagen Rneasy Mini kit (Qiagen), respectively. RNA quality and integrity were measured with the Agilent RNA 6000 Nano Kit on the Agilent 2100 Bioanalyzer, according to manufacturer's instructions. Ribosomal and hemoglobin depleted total RNA-sequencing libraries were created using FastSelect (Qiagen cat#: 335377) and Tecan Universal Plus mRNA-Seq (0520-A01) with adaptations for automation of a Beckmen BioMek FXp system. Libraries were subsequently normalized and pooled for Illumina sequencing using a Labcyte Echo 525 system available at the Center for Advanced Technology at UCSF. The pooled libraries were sequenced on an Illumina NovaSeq S4 flow cell lane with paired end 150 bp reads. Computation processing for genotyping was performed as previously described[23,52]. Briefly, sequencing reads were aligned to the human reference genome and Ensembl annotation (GRCh38 genome build, Ensembl annotation version 95) using STAR v2.7.5c (PMID: 23104886) with the following parameter− --outFilterType BySJout −outFilterMismatchNoverLmax 0.04 − outFilterMismatchNmax 999 −alignSJDBoverhangMin 1 −outFilterMultimapNmax 1 − alignIntronMin 20 −alignIntronMax 1000000 −alignMatesGapMax 1000000. Duplicate reads were removed and read groups were assigned by individual for variant calling using Picard Tools v2.23.3 (https://broadinstitute.github.io/picard/). Nucleotide variants were identified from the resulting bam files using the Genome Analysis Tool Kit (GATK, v4.0.11.0) following the best practices for RNA-seq variant calling[53]. This includes splitting spliced reads, calling variants with Haplotype-Caller (added parameters: --don't-use-soft-clipped-bases -standcall-conf 20.0), and filtering variants with VariantFiltration (added parameters: -window 35 − cluster 3 −filter-name FS -filter FS > 30.0 −filter-name QD -filter QD < 2.0). Variants were further filtered to include a list of high quality SNPs for identification of the subject of origin of individual cells by removing all novel variants, maintaining only biallelic variants with MAF greater than 5%, a mix missing of one individual with a missing variant call at a specific site and requiring a minimum depth of two (parameters: --max-missing 1.0 −min-alleles 2 −max-alleles 2 −removeindels −snps snp.list.txt −min-meanDP 2 −maf 0.05 −recode −recode-INFO-all −out).

## scRNA-seq and CITE-seq sample loading and sequencing

PBL or colon single-cell suspensions from each patient were pooled with equivalent number of live cells and resuspended at $1–2.5 \times 10^3$ cells/μl in 0.04%BSA/PBS, with the addition of 10 μM Y-27632 (Med-Chem Express) for colon samples. Samples from unique individuals were pooled, and samples from the proximal (Right,R) or distal (Left,L) colon of the same individual were placed in separate pools, so each sample could later be uniquely identified using demuxlet[21]. For the primary biopsy UC and VDZ analysis, the two pools were loaded into four wells each of a Chromium Single Cell 3′ v2 Reagent Kit (10X Genomics), with a total of 8 wells (Supplementary Table 2). $1 \times 10^6$ cells of both single-cell colon suspension pools were stained with a custom TotalSeq-A panel, (BioLegend) (Supplementary Table 3) according to the manufacturer's instructions and loaded into two wells. For all experiments, 60,000 cells were loaded per well and processed for single-cell encapsulation and cDNA library generation using the Chromium Single Cell 3′ v2 Reagent Kits (10X Genomics). TotalSeq-A library generation was performed according to manufacturer's instructions (BioLegend). Libraries were sequenced on an Illumina NovaSeq6000 to obtain 25,000 reads per cell for the gene expression libraries and 10,000 reads per cell for the TotalSeq libraries.

## scRNA-seq and CITE-seq data pre-processing, inter-sample doublet detection, and demuxlet

10x Genomics Chromium scRNA-seq data were processed as previously described[23,52]. Briefly, sequencer-obtained bcl files were demultiplexed into individual library fastq trios using the mkfastq program from the Cellranger 3.0.2 suite of tools (https://support.10xgenomics.com). Feature-barcode matrices were obtained for each sample by aligning the raw fastqs to GRCh38 reference genome (annotated with Ensembl v85) using the Cellranger count. Raw feature-barcode matrices were loaded into Seurat 3.1.5[54] and low-quality cells (with fewer than 100 features), and features in 3 or fewer cells were dropped from the dataset. The remaining events were assessed for inter-sample doublet detection (generated due to libraries containing samples pooled prior to loading) using Freemuxlet (https://github.com/statgen/popscle), the genotype-free version of Demuxlet[21]. Clusters of cells belonging to the same patient were identified via SNP concordance to a truth set generated by bulk RNASeq. Briefly, the aligned reads from Cellranger were filtered to retain reads overlapping a high-quality list of SNPs obtained from the 1000 Genomes Consortium (1KG)[55]. Freemuxlet was run on this filtered bam using the 1KG vcf file as a reference, the input number of samples/pool as a guideline for clustering groups of cells by SNP concordance, and all other default parameters. Cells are classified as singlets (arising from a single library), doublets (arising from two or more libraries), or as ambiguous (cells that cannot be accurately assigned to any existing cluster due to a lack of sufficient genetic information). Clusters of cells belonging to a unique sample were mapped to patients using their individual Freemuxlet-generated genotype, and ground truth genotypes per patient identified via bulk RNA-seq. The pairwise discordance between inferred and ground-truth genotypes was assessed using the bcftools gtcheck command[56]. The feature-barcode Matrices were further filtered to remove cells with greater than 50% percent mitochondrial content or ribosomal content, and cells assigned as doublets or ambiguous by Freemuxlet. Visual outliers in the feature-vs-UMIs plots were filtered uniformly across all libraries. The cell cycle state of each cell was assessed using a published set of genes associated with various stages of human mitosis[57].

## scRNA-seq and CITE-seq quality control, normalization, and intra-sample heterotypic doublet detection

The filtered count matrices were normalized, and variance stabilized using negative binomial regression via the scTransform method offered by Seurat[58]. The effects of mitochondrial content, ribosomal content, and cell cycle state were regressed out of the normalized data to prevent any confounding signal. The normalized matrices were reduced to a lower dimension using Principal Component Analyses (PCA) and the first 30 principal coordinates per sample were subjected to a non-linear dimensionality reduction using Uniform Manifold Approximation and Projection (UMAP). Clusters of cells sharing similar transcriptomic signals were initially identified using the Louvain algorithm, and clustering resolutions varied between 0. 6 and 1.2 based on the number and variety of cells obtained in the datasets. All libraries were further processed to identify intra-sample heterotypic doublets arising from the 10X sample loading. Processed and annotated Seurat objects were processed using the DoubletFinder package[59]. The prior doublet rate per library was approximated using the information provided in the 10x knowledgebase (https://kb.10xgenomics.com/hc/en-us/articles/360001378811) and this was corrected to account for inter-sample doublets identified by freemuxlet, and for homotypic doublets using the per-cluster numbers in each dataset. Heterotypic doublets were removed. The raw and log-normalized counts per library were then pruned to retain only genes shared by all libraries. Pruned counts matrices were merged into a single Seurat object and the batch (or library) of origin was stored in the metadata of the object. The log-normalized counts were reduced to a lower dimension using PCA and the individual libraries were aligned in the shared PCA space in a batch-aware manner (Each individual library was considered a batch) using the Harmony algorithm[60]. ADT counts were centered log-ratio (CLR) normalized. The resulting Harmony components were used to generate batch corrected UMAP visualizations and cell clustering.

## scRNA-seq and CITE-seq cell annotation and differential expression

For both blood and biopsy scRNA-seq and CITE-seq, we generated h5ad files with the UMAP, Louvain clusters, and metadata. We then refined the "coarse" and "fine" cell-type annotations in a semi-supervised manner using exploratory CZ CELLxGENE (ExCellxGene) (https://pypi.org/project/excellxgene/), a restructured version of CZ CELLxGENE[61,62]. Expression of cell-type specific markers were used to assign identities to "coarse" and "fine" clusters, guided by previously described gene sets[12,13,15,27,63]. Cells that were unable to be further categorized by fine annotations were labeled as the coarse parental population-not otherwise specified (NOS). The "Mito Hi immune" cluster, consisting of cells with high mitochondrial gene expression, were not considered in downstream analysis. scRNA-seq and CITE-seq data analysis and visualization were then performed in Jupyter note-books using Scanpy =1.9.1[64]. To compute differentially express (DE) genes between two conditions, we first subsetted our datasets with a pair of conditions (HC vs UC, UC vs VDZ, and HC vs VDZ). Then, we used the MAST R package v1.20 which implements a negative-binomial model using the zlm method and corrects for differences in sequencing depth across samples[65]. Briefly, for a subsetted dataset of two conditions, we analyzed all cells in aggregate, and then subsetted again for each cell-type (coarse annotation for biopsies, and fine annotation for blood) to identify DE genes for each cell-type between the two conditions. We also corrected for the number of detected genes as a potential confounding variable[65]. When comparing paired cryopreserved versus fresh samples, patients were treated as a random effect. $P$-values were corrected using the Bonferroni correction. Platelets and erythrocytes were excluded from blood DE gene analysis.

## Fresh versus cryopreserved biopsy scRNA-seq comparison

For the fresh versus cryopreserved comparison, twelve colon biopsies were divided into two vials, each containing six biopsies, for two donors. The fresh biopsies were stored in Basal Media on ice for 80 min, while the cryopreserved biopsies were transferred into cryovials containing freezing media (90% (v/v) FCS, 10% (v/v) DMSO and 10 μM Y-27632) and immediately placed into a freezing container (Mr

Frosty or Coolcell) and stored at −70 °C for 80 min. The cryopreserved biopsies were then thawed, and both the fresh and cryopreserved biopsies were digested as above. Half of the fresh biopsies from each donor were labeled with TotalSeq-A0251 barcode sequence GTCAACTCTTTAGCG (Biolegend 394601), and the other half of the fresh biopsy sample was labeled with MULTI-seq lipid-tag barcode sequence CCTTGGCACCCGAGAATTCCAGGAGAAGA[66]. Half of the cryopreserved biopsies from each donor were labeled with TotalSeq-A0252 barcode sequence TGATGGCCTATTGGG (Biolegend 394603), and the other half of the cryopreserved biopsy sample was labeled with MULTI-seq lipid-tag barcode sequence CCTTGGCACCCGAGAATTCCACCACAATG[66]. The MULTI-seq labeled samples were combined into one pool, and the TotalSeq Hasthag samples were combined in a second pool. The two pools were loaded separately into one well each of a Chromium Single Cell 3′ v2 Reagent Kit (10X Genomics). 60,000 cells were loaded per well and processed for single-cell encapsulation and cDNA library generation as described above. Representative UMAPs comparing cryopreserved and fresh biopsies and coarse cell subset annotations, as well as the scRNA-seq DE genes, were generated from the MULTI-seq labeled samples as described above. Quantitation of cell subset frequency was performed on both MULTI-seq and TotalSeq barcoded samples.

## Mass Cytometry (CyTOF) sample staining and acquisition

A 37-parameter CyTOF panel was designed (Supplementary Table 3). All mass cytometry antibodies were conjugated in-house to their corresponding metal isotope. Metals were conjugated according to the manufacturer's instructions (Fluidigm, South San Francisco, CA, USA). In brief, this process consisted of loading the metal to a polymer for 1 h at RT. The unconjugated antibody is transferred into a 50 kDA Amicon Ultra 500 V-bottom filter (Fisher Scientific, Hampton, NH, USA) and reduced for 30 min at 37 °C with 1:125 dilution of Tris (2-carboxyethyl) phosphine hydrochloride (TCEP) (ThermoFisher, Waltham, MA, USA). Subsequently, the column was washed twice with C-buffer (Fluidigm) and the metal-loaded polymer was suspended in 200 μL of C-buffer in the 3 kDA Amicon Ultra 500 mLV-bottom filter. The suspension was transferred to the 50 kDA filter containing the antibody and incubated for 1.5 h at 37 °C. After incubation, antibodies were washed three times with W-buffer (Fluidigm) and quantified for protein content using Nanodrop. Once the concentration was determined, the antibodies were resuspended at a concentration of 0.2 mg/mL with Antibody Stabilizer (Boca Scientific, Dedham, Ma, USA) and stored at 4 °C. Optimal concentrations for all antibodies were determined by different rounds of titrations. The staining protocol was optimized to use each antibody in aliquots of 6 million cells as previously described[67].

## CyTOF panel design, staining, and acquisition

CyTOF was performed on paired peripheral blood and colon biopsies obtained from the primary case control study ($n = 12$) and from the secondary remission case control study ($n = 17$). Single-cell suspensions of colon biopsies from the right and left colon were pooled to ensure sufficient cell counts per donor for the primary case control study ($n = 12$), or run in separate pools for the secondary remission case control study ($n = 17$). Dead cells were labeled with Cisplatin (Fluidigm) according to the manufacturer's instructions, washed in wash buffer (PBS, 0.5% BSA, 5 mM EDTA), fixed in 1.6% PFA for 10 min, washed in wash buffer and then resuspended in freezing medium (PBS, 0.5% BSA, 10% DMSO) and stored at −80 °C until staining. Samples were then thawed and washed with wash buffer. Prior to staining with the antibody panel (Supplementary Table 3), cells from each patient were barcoded using a unique set of metals, enabling sample identification as previously described[68,69]. The barcode staining was performed following the manufacturer's instructions (Fluidigm, South San Francisco, CA, USA). Briefly, each sample was incubated for 15 min at RT on a shaker (200 rpm) with a barcoding solution containing 10 μL of

barcode in 1x Perm Buffer solution (Fluidigm, Cat#201057) diluted in cell staining media (CSM, Fluidgim, Cat#201068). Samples were then washed, centrifuged, resuspended in CSM, and pooled. Subsequently, extracellular staining was performed for 30 min at 4 °C. After incubation, the samples were washed with CSM and centrifuged before resuspending in 1x Permeabilization Buffer (eBioscience™ Permeabilization Buffer Cat# 00-8333-56) for 10 min at 4 °C. The samples were then washed and incubated with Ir-intercalator (Biolegend CNS, San Diego, CA, USA) diluted 1:500 in 4% fresh PFA for 20 min at RT. After incubation, samples were washed and kept at 4 °C overnight in EQ™ bead solution (Fluidigm Cat#201078) diluted in MaxPar Water (Fluidigm Cat# 201069) at $1.2 \times 10^6$ cells/mL. Samples were analyzed on the CyTOF®2 instrument (Fluidigm). Commercial Fluidigm CYTOF software was used for CyTOF data acquisition.

## CyTOF data analysis

After acquisition, the.fcs files were concatenated, normalized to the EQ™ calibration beads, and de-barcoded using CyTOF software (Fluidigm). FlowJo v10 software was used for confirming the elimination of the EQ™ calibration beads, concatenating, and manually gating the files. Singlets were gated by Event Length and DNA. Live cells were identified as cisplatin-negative. The unsupervised analysis was performed on the case control study ($n = 12$) using an R-based Cytometry Clustering Optimization aNd Evaluation (Cyclone) pipeline developed by the UCSF Data Science CoLab (https://github.com/UCSF-DSCOLAB/cyclone)[38]. Specifically, the data were preprocessed, arcsinh transformed (cofactor 5) and then clustered using FlowSOM 2.6.0[70]. We used default values for FlowSOM parameters except for the grid size. A grid size of $3 \times 7$ was chosen based on a local minimum of Davies-Bouldin Index (DBI). The clustering was visualized using UMAP, which was calculated using uwot package in R. The pre-cluster median expression levels of each of the 37 antibodies panel for each cluster were used to annotate clusters. This was plotted as a heatmap. In parallel, supervised analysis was performed on the same dataset and on the remission case control study ($n = 17$) defining cell subsets based on canonical markers and followed the scheme illustrated in Supplementary Fig. 10 and Supplementary Table 4. Additionally, the supervised analysis was used to further define cell subsets expressing $\alpha_4\beta_7$. Finally, specific populations and markers of the focused panel were manually gated to validate and extend the results from the unsupervised analysis.

## Histology FFPE tissue microarray (TMA) construction

Colon biopsies from patients were fixed in 10% neutral-buffered formalin (Millipore Sigma) for 16–24 h, washed in PBS three times, then placed in 70% ethanol and stored at RT until paraffin-embedding by the Biospecimen Resource Program (BIOS) at UCSF. Sectioning, hematoxylin and eosin (H&E), and high-resolution (40X) scanning were performed according to standard protocols. H&E histologic severity quantification was performed using the Geboes scoring system[71]. Tissue microarrays were constructed from prospectively collected FFPE blocks by UCSF BIOS or retrospectively retrieved, clinical archived FFPE blocks by Pantomics, with 1.1–2 mm cores. The recipient TMA block was sectioned with a clearance angle of 10° and a thickness of 4 μm or 5 μm along the width of the block and used for H&E staining and multiplexed RNA-ISH. The TMA block was stored at −20 °C in order to reduce sample degradation and preserve RNA detection in the transcriptomic assays[72]. Freshly cut TMA sections were prepared before each experiment, according to the type of spatial assay performed.

## CODEX multiplexed tissue staining, imaging, and data analysis

Colon biopsies from the FFPE TMA block were mounted on coverslips provided by Akoya Biosciences and prepared according to the CODEX® User Manual. Briefly, FFPE TMAs were sectioned onto poly-L-lysine-coated coverslips with a thickness of 5 μm. TMA coverslips were stored individually at 4 °C to avoid tissue damage until the experiment

run. Conjugation of DNA oligonucleotides to purified antibodies, antibody validation and titration, and CODEX multicycle reactions were optimized and performed by Akoya Biosciences according to the protocol previously described[73]. CODEX samples were stained and imaged by Akoya. Briefly, TMA coverslips were pretreated on a hot plate at 55 °C with tissue facing up for 20–25 min, then positioned on a rack at RT for 5 min to allow the tissues to cool down. Subsequently, deparaffinization and hydration steps were performed. Next, the TMA coverslips were submerged in a beaker containing antigen retrieval solution (1x Tris/EDTA buffer, pH 9) and placed in a pressure cooker for 20 min. After incubation, samples were cooled at RT for 10 min, washed in distilled water for 2 min (2x), incubated in a hydration buffer for 2 min (2x), and then equilibrated in a staining buffer for 20-30 min before antibody staining. A customized CODEX panel of 28 antibodies was used and each CODEX®-tagged antibody had a barcode that was complementary to a specific reporter; all related details are summarized in Supplementary Table 3. A single staining step with the antibody cocktail solution containing blocking buffer and the panel of antibodies was performed on the tissues for 3 h at RT in a humidity chamber. After incubation, the TMA coverslips were washed with the staining buffer for 2 min (2x) then the coverslips were incubated with post staining fixing solution (2% PFA) for 10 min at RT. Subsequently, TMA coverslips were washed in 1x PBS (3x), placed in ice-cold methanol for 5 min and transferred rapidly in 1x PBS for three consecutive washes. After the washing steps, TMA coverslips were placed on a tray and tissues were incubated with a final fixative solution for 20 min in a humidity chamber. After incubation, TMA coverslips were rinsed with 1x PBS (3x) and were ready for the CODEX® multicycle experiment. Images were collected using a Keyence BZ-X800 Fluorescent microscope configured with fluorescent channels (ATTO550, Cy5, AF647, and AF750) and equipped with a cooled CCD camera. The resolution was 0.25 μm per pixel. Images for the 28 DNA-conjugated antibodies were acquired over 13 cycles of staining. In each cycle, the reporters revealed up to three markers of interest (and DAPI) simultaneously. After each cycle of imaging, the reporters were removed from the tissue by a gentle isothermal wash. Each core was imaged with a 20x objective. Images were processed, stitched with background subtraction and deconvolution, and aligned into a 28-color overlay figure for the selected markers for analysis at UCSF. Cell segmentation was performed using HALO 3.4 image analysis software (Indica Labs), DAPI was used as the reference nuclear dye. Nuclear contrast threshold, minimum nuclear intensity, and nuclear size were determined using the pre-trained Nuclear Segmentation algorithm (AI default) and kept constant for the entire analysis. Membrane segmentation was done considering a cytoplasm radius of 6 μm based on average cell sizes determined using reference H&E images. After cell segmentation, comma-separated value (.csv) files containing signal intensity at single-cell level were generated from each core and used for unsupervised analysis. Data were imported into R and the unsupervised analysis was performed using the Cyclone pipeline as described above[38]. The number of clusters was defined using the FlowSOM clustering (grid size 4 × 6) and then visualized using UMAPs. The median level of expression of each of the 28 antibodies panel for each cluster was used to annotate the clusters. This was plotted as a Heatmap. Subsequently, based on the unsupervised clusters, a supervised analysis in HALO was performed defining 18 phenotypes based on positive and/or negative channel selection criteria using the HiPlex FL v 4.1.3 module. Cell frequency for each phenotype was exported as a percentage of positive cells over the total number of cells defined. To define cell interactions, spatial plots were generated for each core using the Spatial Analysis module in HALO. Nearest Neighbor Analysis was conducted on the object data from each suitable core to determine the average distance between two cells or object populations. For this experiment, a total of 20 cores were imaged and 15 of them were selected for further analysis based on the quality of the core. Low-quality cores were excluded if they were grossly damaged or detached during processing, or if they had poor nuclei staining that precluded cell segmentation. For Nearest Neighbor Analysis 12 cores were considered, all the cores containing less than 5 unique cells per phenotypes of interest were excluded from the analysis.

## MIBI staining, data acquisition, and analysis

Slide preparation, staining, antibody optimization, and imaging were performed by the MIBI Core at UCSF. Antibodies were conjugated to metal-loaded MIBItags (Ionpath) according to the manufacturer's instructions. Initial imaging QC was performed on the conjugated antibodies and compared to already established positive controls of FFPE spleen and tonsil tissue. The final 39-plex antibody panel for MIBI-TOF is reported in Supplementary Table 3. TMA slides containing 20 cores of colon tissues were sectioned (5 μm section thickness) from paraffin tissue blocks on gold and tantalum-sputtered microscope slides. Slides were baked at 70 °C overnight, followed by deparaffinization and rehydration with washes in xylene (3x), 100% ethanol (2x), 95% ethanol (2x), 80% ethanol (1x), 70% ethanol (1x) and distilled water (1x). Next, tissues underwent antigen retrieval by submerging the slides in 1X Target Retrieval Solution (pH 9, DAKO Agilent) and incubating at 97 °C for 40 min in a Lab Vision PT Module (Thermo Fisher Scientific). After cooling to RT for 30 min, slides were washed in 1X TBS-T pH7.6 (IONpath). Subsequently, all tissues underwent two rounds of blocking, the first to block endogenous biotin and avidin with an Avidin/Biotin Blocking Kit (BioLegend). Tissues were then washed with wash buffer and blocked for 11 h at RT with 1X TBS with 5% (v/v) normal donkey serum (Sigma-Aldrich). The first antibody cocktail was prepared in 1X TBS-T 5% (v/v) normal donkey serum (Sigma-Aldrich) and filtered through a 0.1μm centrifugal filter (Millipore) prior to incubation with tissue overnight at 4 °C in a humidity chamber. Following the overnight incubation, slides were washed twice for 5 min in 1X TBS-T pH7.6. The second day, an antibody cocktail was prepared as described and incubated with the tissues for 1 h at 4 °C in a humidity chamber. Following staining, slides were washed twice for 5 min in wash buffer and fixed in a solution of 2% glutaraldehyde (Electron Microscopy Sciences) solution in low-barium PBS for 5 min. Slides were washed in PBS (1x), 0.1 M Tris at pH 8.5 (3x) and distilled water (2x) and then dehydrated by washing in 70% ethanol (1x), 80% ethanol (1x), 95% ethanol (2x) and 100% ethanol (2x). Slides were dried under vacuum prior to imaging. Imaging was performed using a MIBIscope (IonPath) with a Hyperion ion source. Xe+ primary ions were used to sequentially sputter pixels for a given field of view (FOV). The following imaging parameters were used: acquisition setting, 80 kHz; field size, 400 μm x 400 μm at 1,024 ×1,024 pixels; dwell time, 24 ms; median gun current on tissue, 1.45 nA Xe + ; ion dose, and 3.75 nAmp h per mm2 (450 μm2 FOVs). Mass correction was done using MIBI/O (IONpath) with their standard JSON file. FOV images were analyzed using the HALO software and the HiPlex FL v 4.1.3 module was used. Cellular segmentation was performed using the Histone H3 nuclear DNA marker as the reference nuclear dye. Nuclear contrast threshold, minimum nuclear intensity, and nuclear size were determined using the traditional segmentation method and kept constant throughout the entire image analysis. Membrane segmentation was done using EpCAM as the reference dye, and a cytoplasm radius of 6 μm was set based on average cell sizes determined using reference H&E images. 20 phenotypes each corresponding to a major cell type were created based on positive and/or negative channel selection criteria. To define cell interactions, spatial plots were generated for each suitable FOV using the Spatial Analysis module. Nearest Neighbor Analysis was conducted on the object data to determine the average distance between two cells or object populations. A total of 37 FOV images were acquired and 32 of them were used for phenotype identification and quantification. FOVs that contained exclusively lymphoid aggregates, with no mucosal or submucosal cellular compartments, were excluded. Among the 32 FOVs

analyzed, 29 were selected for Nearest Neighbor Analysis based on the number of cells per phenotype. FOVs containing less than 5 unique cells per phenotype of interest were excluded from the analysis.

## Multiplexed (12-plex) RNA in situ hybridization (RNA-ISH)

FFPE TMA slides containing colon biopsies from our study cohort were used for multiplexed RNA-ISH. The assay was performed following the manufacturer's instructions (HiPlex RNAscope, ACDBio). Briefly, FFPE TMA slides were deparaffinized sequentially with xylene (2x, 5 min) and 100% ethanol (2x, 2 min). After deparaffinization, a retrieval step of 15 min at 99 °C was performed. Slides were permeabilized with Protease III for 30 min at 40 °C. The 12 probes were hybridized for 2 h at 40 °C and amplified (3 amplification steps, 30 min each at 40 °C) using the HybEZ hybridization system. Probes were chosen from high-abundance landmark genes from the scRNA-seq data and additional genes of interest. Probes are listed in Supplementary Table 3. After probe hybridization and amplification steps, slides were hybridized with the first round of fluorophores for 15 min at 40 °C then washed, counterstained with DAPI, and mounted with ProLong Gold Antifade Mountant. The first round of imaging was performed by scanning each full core using the automated Zeiss Axioscanner Z1 with a 20X objective on Orca Flash 4.0 v2 (Hamamatsu) camera in fluorescence channels (DAPI, FITC/GFP, Cy3/Atto550, and Cy5). The image scanner was controlled using ZEN 3.1 software. After each round of imaging, the coverslip was removed by soaking in 4x SSC buffer for 30 min and fluorophores were cleaved using manufacturer cleaving solution. Then the slides were washed and prepared for the second round of fluorophore hybridization and imaging, these last steps were repeated for a total of four rounds. All images were acquired at the UCSF HDFCCC Histology and Biomarker Core. A total of 21 core images were obtained and 19 of them were used for further analysis. Low-quality cores were excluded if they were grossly damaged or detached during processing. Images obtained from the four rounds of staining were then registered, fused, and analyzed using HALO 3.1 image analysis software (Indica Labs). For this analysis, we used the following modules: TMA and FISH v3.1.3. Cell detection was based on Nuclei staining (DAPI) with traditional nuclei segmentation type, nuclear contrast threshold was set at 0.5 with an intensity of 0.015 and aggressiveness of 0.6. The nuclear size was ranging from 5 to 150 ($\mu m^2$), the minimum roundness of segmentation was set at 0.05 and the cytoplasm radius was set at 1.5μm. For the analysis, the FISH score was set at 2+ minimum copies/cell, and phenotypes were defined by exclusive channels and according to the round of imaging. Data were expressed as the percentage of positive cells among the total number of cells detected.

## Highly multiplexed CosMx spatial transcriptomics tissue processing, staining, imaging, and analysis

FFPE TMA processing, staining, imaging, and cell segmentation were performed as previously described by Nanostring and data were analyzed at UCSF[39]. Briefly, 4–5 μm sections of an FFPE TMA block were sectioned on the back of a VWR Superfrost Plus Micro Slide (Cat# 48311-703) using a microtome, placed in a heated water bath, and adhered. Slides were then dried at 37 °C overnight, vacuum sealed, and stored at 4 °C until analysis. Manual FFPE tissue preparation, ISH hybridization, coverslip application, and cyclic RNA readout on the SMI were performed as previously described[39]. After all cycles were completed, additional visualization markers for morphology and cell segmentation were added including DAPI, pan-cytokeratin (PanCK), CD45, CD3, CD68, and/or CD298/B2M as indicated in AnnData objects (anndata =0.8.0). A 3D multichannel image stack was obtained at each FOV location. Registration, feature detection, localization, determination of the presence individual transcripts, and cell segmentation were performed as previously described[39]. The final segmentation mapped each transcript location in the registered image to the corresponding cell, as well as to the cell compartment (nuclei, cytoplasm, membrane), where the transcript is located. Other features/properties generated included shape (area, aspect ratio) and fluorescence intensity statistics (minimum, maximum, average) per cell. Single-cell spatial RNA-ISH results were analyzed using scanpy[64] and squidpy =1.2.3[74]. Low-quality cores were excluded if they were extensively damaged or detached during processing, or if they had poor nuclei staining that precluded cell segmentation. For the primary CosMx experiment, 22 FOV were selected, five were ultimately eliminated due to poor tissue adherence or low gene count per cell. For the secondary longitudinal experiment of archived FFPE specimens, 73 out of 81 FOV passed quality control. Cells with <10 unique genes per cell or <50 counts per cell were filtered out, and genes expressed in fewer than 1 cell (1st batch) or 10 cells (2nd batch) were excluded. The data were further normalized, log-transformed, and scaled as previously described[75]. For cell type identification and annotation of CosMx results alone, principal components were computed using scanpy's tl.pca() function with default settings. UMAP plots and leiden clustering were calculated using tl.umap() and tl.leiden(). To annotate the computed clusters, we examined the top differentially expressed genes in each cluster using the tl.rank_genes_groups() function and compared with known marker genes for the various cell types. We then refined the "coarse" and "fine" cell-type annotations in a semi-supervised manner using exploratory CZ CELLxGENE (ExCellxGene =2.9.2)[61,62]. Neighborhood enrichment analysis was performed using squidpy's sq.gr.nhood_enrichment() function. CosMx pseudobulk DE genes were analyzed using DESeq2 to compare non-responders to responders pre-treatment[76].

## Antibodies and reagents

Antibodies and reagents for all experiments are listed in Supplementary Table 3.

## Gene Set Enrichment Analysis (GSEA) on a validation cohort

The bulk transcriptomic study (GSE73661)[1] performed on colonic biopsies obtained from UC patients before and after VDZ treatment was downloaded from GEO database and used for GSEA analysis[41]. Samples were divided into pre- and post-VDZ treatment and responders (R, responding at week 52, $n = 9$) or non-responders (NR, not responding at week 52). Three comparisons were made: 1-pre- vs post-VDZ treatment for responders, 2-pre- vs post-VDZ treatment for non-responders, and 3-responders (at either week 6,12 or 52) vs non-responders (at either week 12 or 52) pre-VDZ treatment ($n = 11$ for R and $n = 9$ for NR). Data were normalized and expressed as Z-scores before reading into the GSEA program (version 4.3.2). Based on our gene expression data we defined 14 cell type gene signatures with a minimum of 10 genes (Supplementary Table 5). GSEA analysis was performed for each gene signature. The number of permutations was set at 1000, no collapse dataset, chip Affymetrix Human Gene 1.0 ST Array, t-test. For each analysis and gene set, a Normalized Enrichment Score (NES) was calculated and only NES with a $p$-value < 0.05 and adjusted q-value (FDR) < 0.1 were considered significant. A leading edge analysis was performed to display overlaps between all cell subsets and to elucidate key genes that contributed the most to the enrichment signal of specified gene sets. Based on our spatial transcriptomic data from the longitudinal experiment we defined pre-VDZ responder and non-responder signatures (Supplementary Table 5) and queried these genes in responders vs non-responders from pre-VDZ (R = 11 and NR = 9) and pre-IFX (R = 8 and NR = 15) dataset[1]. Additional leading edge analyses were performed on these GSEAs in order to define the genes that significantly contributed to the core enrichment for pre-VDZ and pre-IFX.

## Statistics

Non-parametric comparisons were performed using one-way ANOVA Kruskal-Wallis test for multiple groups or Mann-Whitney test for two groups, followed by a two-stage linear step-up procedure of

Benjamini, Krieger and Yekutieli to correct for multiple comparisons by controlling the false discovery rate (FDR). Parametric comparisons for multiple groups were performed using two-way ANOVA if comparing multiple variables per group or one-way ANOVA if comparing one variable per group, followed by FDR correction. The q-value is the FDR-adjusted $p$ value, and q < 0.1 was used as the threshold for discovery, unless otherwise indicated. For biopsy CITE-seq cell subset frequency analysis, an additional nested one-way ANOVA test was performed on log transformed values treating biopsies as replicates, with unadjusted $p < 0.05$ as an additional threshold for discovery. Categorical variables were analyzed by Chi-square test as indicated. For MAST DE gene analysis, $p$ values were corrected using Bonferroni correction, and corrected $p$ value of <0.05 was used as the threshold for statistical significance. For CosMx Deseq analysis, in addition to q < 0.1 after FDR-correction for all transcripts per FOV, DEseq comparisons were made treating FOVs or biopsies as replicates with unadjusted $p < 0.05$ as an additional threshold for discovery (Supplementary Table 7). For all other analyses, biopsy values were averaged per patient. Additional analyses were performed using GraphPad PRISM 9. Hierarchically clustered heatmaps for cell subset abundance (Euclidean distance, average linkage) and marker similarity matrices (Pearson correlation) were generated with the Morpheus software (https://software. broadinstitute.org/morpheus/). The ComplexHeatmap R package was used to generate expression z-score heatmaps for DE genes.

### Reporting summary

Further information on research design is available in the Nature Portfolio Reporting Summary linked to this article.

## Data availability

Processed data are deposited as a GEO Super Series under accession code GSE250498 and are publicly available. The raw sequencing data generated in this study have been deposited on the dbGaP database under accession code phs003502.v1.p1. The raw sequencing data are available under controlled access for privacy concerns; access can be requested through dbGaP. All additional data generated in this study are provided in the Supplementary Information/Source Data file and on Figshare. Processed and annotated objects are saved in AnnData (h5ad) format[77]. These AnnData objects are accessible in Figshare (fresh versus cryopreserved scRNA-seq 10.6084/m9.figshare. 21936240; blood scRNA-seq 10.6084/m9.figshare.21900948; biopsy scRNA-seq https://figshare.com/s/3ceec45e4a640a7bceb4; biopsy CITE-seq 10.6084/m9.figshare.21919356; PBL CyTOF 10.6084/m9. figshare.21977834; biopsy CyTOF 10.6084/m9.figshare.21977798; secondary CyTOF PBL and biopsy analysis 10.6084/m9.figshare. 23902065; CosMx 960-plex RNA-ISH 1st run 10.6084/m9.figshare. 21919338; CosMx 1000-plex RNA-ISH 2nd longitudinal analysis 10. 6084/m9.figshare.23896959). Publicly available microarray data (GSE73661) were downloaded from the NCBI gene expression omnibus. GRCh38 was used as the reference genome. Source data are provided with this paper.

## Code availability

All code used in this study including R markdowns, Jupyter notebooks, and conda environment.yaml files are available on the Ulcerative Colitis project GitHub repository (https://github.com/mkattah/UC_VDZ). R-based Cytometry Clustering Optimization aNd Evaluation (Cyclone) pipeline was developed by the UCSF Data Science CoLab (https://github.com/UCSF-DSCOLAB/cyclone).

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

## Acknowledgements

MGK was supported by a Career Development Award from the Crohn's and Colitis Foundation and NIH K08 DK123202. Michael George Kattah, M.D., Ph.D., holds a Career Award for Medical Scientists from the Burroughs Wellcome Fund. This work was also supported by funding from UCSF ImmunoX and the Kenneth Rainin Foundation. Schematics created with BioRender.com. This work was also supported by RRID:SCR_018206, DRC Center Grant NIH P30 DK063720, NIH S10 1S10OD018040-01 (CyTOF), and NIH S10 1S10OD025187-01 (MIBI). We would like to acknowledge Nanostring CosMx™ SMI Technology Access Program, Akoya Biosciences Spatial Tissue Exploration Program (STEP), ACDBio, Pantomics, and the Indica Labs HALO team. We thank James Lord for helpful discussion. We also acknowledge the Research Core of the UCSF Division of Hospital Medicine. We thank the study participants for contributing to this research.

## Author contributions

Conceptualization: MGK; patient recruitment and sample collection: G.L., S.Li., J.L.B., F.B.V., K.B., N.E., S.Lewin, D.R.S., J.P.T., U.M., M.G.K.; sample processing (scRNA-seq/CITE-seq, CyTOF): E.M., G.L., I.R., J.T., A.S., W.E., D.Y.O., A.J.C., M.G.K.; scRNA-seq/CITE-seq analysis: Y.J.K., C.A., A.R., D.Y.O., G.K.F., A.P., M.G.K.; CyTOF analysis: E.M., G.L., I.R., R.K.P., J.T., A.S., G.K.F., A.J.C., M.G.K.; MIBI and CODEX processing and analysis: E.M., G.L., I.R., R.K.P., J.L.B., S.T., M.R., M.N., S.J.C., G.K.F.; 12-plex RNA-ISH: E.M., I.R., J.L.B., M.N., S.J.C.; CosMx: E.M., Y.J.K., G.L., I.R., L.C.D., J.L.B., M.L.L., E.F., G.K.F., A.P., A.J.C., M.G.K.; data curation: E.M., Y.J.K., M.L.L., L.C.D., M.G.K.; G.S.E.A.: E.M.; supervision: G.K.F., A.P., A.J.C., M.G.K.; funding acquisition: M.G.K.; all authors contributed to manuscript preparation.

## Competing interests

S. Lewin has received research support from Takeda. N. El-Nachef is a consultant for Ferring, Federation Bio Grant, and receives funding from Finch Therapeutics, Seres, Freenome, and Assembly Biosciences. U. Mahadevan serves as a consultant for Abbvie, BMS, Boeringher Ingelheim, Gilead, Janssen, Lilly, Pfizer, Prometheus biosciences, Protagonist, Rani Therapeutics, Surrozen, and Takeda. D. Oh has received research support from Merck, PACT Pharma, the Parker Institute for Cancer Immunotherapy, Poseida Therapeutics, TCR2 Therapeutics, Roche/Genentech, and Nutcracker Therapeutics, and travel/accommodations from Roche/Genentech. The Combes lab has received research support from Eli Lilly and Genentech and A. Combes consults for Foundery Innovations. The Kattah lab receives research support from Eli Lilly. M. Kattah has consulted for Sonoma Biotherapeutics and Morphic Therapeutic. The remaining authors declare no competing interests.
