## [Peer Review File · Nature Communications]

Single-cell and spatial multi-omics highlights effects of anti-integrin therapy across cellular compartments in ulcerative colitisReviewers' comments:

Reviewer #1 (Remarks to the Author):

In this manuscript, Mennillo et al. investigate the mechanisms of vedolizumab in patients with ulcerative colitis. To this end, they use a variety of cutting-edge techniques to characterize the immune cell composition in the peripheral blood and intestinal mucosa. They claim that their data show primary effects of vedolizumab on mononuclear phagocytes and rather modest effects on lymphocytes.

Overall, this manuscript addresses an important topic that has potential relevance for the therapy of patients with IBD.

Although performed with a small number of samples, the comprehensive characterization of the samples with complementary techniques provides important insights.

However, I have a few concerns, when reading the manuscript:

Most importantly, the authors should be very careful in their wording and conclusions throughout their manuscript. The data show abundance of cell populations and expression of molecules by these cells in healthy controls and patients with UC treated with VDZ or not. However, it is essential to acknowledge that observing differences in the abundance of cell populations or in the expression of molecules by these cells between patients treated with VDZ or not does not prove a direct mechanism of UC on these cells. Such effects can be secondary to other mechanisms (which the authors automatically assume for endothelial and epithelial cells) and conclusions on the mechanisms can only be drawn, when correlated to functional investigations.

Indeed, functional effects of vedolizumab on a broad variety of immune cell subsets have previously been shown. Therefore, the findings of the paper are not entirely new (although of unprecedented depth), but claiming a primary mechanism on one of these subsets is not supported by the data. More precisely, the authors show a more prominent effect (whether downstream or upstream) on certain populations.

What comes closest to a functional read-out is the analysis presented in Fig. 4o, but this does actually not support the main claim of the paper, since I can overall not observe a superior effect on MNP compared to lymphocytes in this panel.

It is important to appropriately describe and discuss these aspects in all its facets and/or to add additional data to further support the current claims.

Reviewer #2 (Remarks to the Author):

Review on NatComm

This is a study evaluating the potential mechanism of action of the anti-integrin antibody vedolizumab by exposing a very small clinical cohort (HC, UC, UC + VDZ, each n=4) to a wide array of technologies (scRNA seq, CITE-Seq, CyTOF, CODEX). In essence their key findings, as displayed in the main figures, are:

- scRNA from peripheral blood only reveals subtle differences in leukocyte frequencies
- CITE-Seq from mucosal biopsies shows statistically significant reduction in mDC and increase in intestinal epithelial cell in VDZ-treated patients.
- CyTOF analysis show increased abundance of circulating a4b7 DCs in the peripheral blood of VDZ-treated UC patients and broad expansion of IEC in UC-VDZ, compared to UC.

- Decrease in stromal fibroblast in UC-VDZ, as shown in CODEX.
- Decreased spatial proximity between MNP and fibroblast in UC-VDZ, as shown by spatial transcriptomics.

The amount of accumulated data is impressive, and I do like the concept of functional proximity. But I do have two major conceptual issue that raise my criticism:

- 1) There is a major flaw and structure of the clinical cohort as UC and UC-VDZ are significantly different in their endoscopic Mayo score and it is known that the degree of mucosal inflammation as the biggest effect on cellular composition, spatial proximity and transcriptomic signatures. In addition, HS12 (UC-VDZ, eMAYO 1) was on VDZ for only 2 month, meaning that the full manifestation of VDZ treatment is not yet established and underlies a much more dynamic fluctuation of mucosa inflammation as a patient on VDZ for e.g. 64 month. Bearing this heterogenous clinical phenotype in mind I seriously doubt that any observation and drawn conclusion can be attributed to VDZ effect.
- 2) There is no stringent line in which the data of each individual technical approach are tried consolidating into a bigger mechanistic picture on the MOA of VDZ and how this knowledge might be used to improve current treatment strategies with VDZ. With the exception of one example, a publicly available data set on VDZ response, which is not cited in the manuscript (see page 12, line 18), there is no external validation. Bearing in mind that the authors deal with n=3-4 samples/group this is surprising. In the particular example the author aim to validate their own identified gene signature in a cohort of VDZ response but their own dataset is not designed to test effects on VDZ response.

Minor:

Figure 6f/h: The authors show a representative picture of decreased spatial proximity between activated fibroblast and MNP in UC-VDZ patients. However, the shown IHC slide is counterintuitive as the overall number of activated fibroblast is dramatically increased.

Reviewer #3 (Remarks to the Author):

Mennillo et al compare single cell transcriptomic, proteomic and spatial profiling of blood and colonic biopsies in UC patients treated with aminosalicylates or Vedolizumab, a biologic which is thought to act through blocking leukocyte trafficking to the intestine. The treatment resulted in cell type changes indicating mucosal healing including increase in epithelial cell abundance and reduction in activated fibroblasts and inflammatory macrophages in the colon samples. The key finding is that while in general circulating leukocyte frequencies were stable in blood and colon, VDZ treatment blocks trafficking of integrin expressing MNP cells from blood to the colon. Authors propose an interaction of activated fibroblasts and MNPs upon VDZ treatment and observe similar populations in VDZ non-responders, but the mechanism could be interrogated in more detail. The main advantage is that the data presented is of high quality and presents an interesting dataset to interrogate the immune cell trafficking in IBD. The main weakness is the lack of more mechanistic insight into the cellular changes. We raise the following questions for the authors to comment on and suggestions for further analysis.

Major comments:

What was the time period between VDZ treatment and blood and mucosa sample collection in patients? Would it be possible that immediate changes in lymphocyte trafficking were not captured in the VDZ treated patients and that what the authors observe is mucosal healing as a consequence of balancing the influx of lymphocytes?

We recognise that patient recruitment is a complicated process and that perfect controls do not exist. However, could the authors comment in the manuscript what are the effects of 2-AZA treatment and why were they selected as untreated patient control? How do these patients compare with other UC patients on the single cell level such from published studies (e.g. Smillie et al., Cell, 2019)?

Through the manuscript, the comparisons are made between HC vs UC and UC vs VDZ-UC. Could the authors show the comparison between HC vs VDZ-UC to define the difference between these groups (e.i. if the treated patients show signs of returning to a healthy cellular state)?

Only one VDZ-UC patient has a high inflammation score, this should be made clear in text when discussing the conclusions between cell changes due to the treatment vs disease severity.

In Fig 1e, while this plot with log transformed scale is informative for fresh vs cryopreserved comparison, it may be misleading, as the sample is actually dominated by plasma and T cells, but this is not reflected in the plot. This applies to other bar plots throughout the manuscript.

Related to the previous point, to support the enrichment result, the authors can consider using milo tool (Dann et al., Nat. Biotechnology, 2022) for a statistical approach to investigate the differential abundance across conditions in single cell data.

In blood CyTOF data (or Figure 4p), there is an increase in retention of a few myeloid subsets (cDC1, cDC2, cDC2b, pDC), but the only population with significant reduction in colon samples (or Figure 4q) is cDC1. Can authors comment on their interpretation why only cDC1 shows change in the biopsy samples? Was this trend also observed using other profiling technologies (e.i. number of CD103 expressing cells in scRNAseq data)?

Related to the previous point, can the authors elaborate on how MNPs might contribute to disease based on the current state of knowledge of the role of MNPs in inflammatory diseases?

Could the authors also investigate/elaborate more on the MADCAM1 expression in your single cell and CyTOF datasets as this is the other part of the interaction via $\alpha 4\beta 7$ integrin? Is there a change in MADCAM1 expression or MADCAM1+ cell type abundance before and after VDZ treatment?

To make the MNP - activated fibroblast observation stronger, the authors should consider doing cell-cell interaction analysis in their single cell blood and colon data using CellphoneDB, NicheNet or similar ligand-receptor analysis method. This would allow the authors to interrogate the interactions between these cell subsets and propose a mechanism of how these interactions lead to epithelial cell recovery in VDZ-UC treated patients or are

dysregulated in non-responder patients.

In addition to responders vs non-responders to VDZ GSEA analysis, the authors could use bulk data deconvolution methods (MuSiC or SCDC) to infer the proportion of cells in the responders vs non-responders based on their single cell data.

Minor comments:

Fig 1d “03B CD4 T naïve and Treg” changed to “03B- CD4 T naïve and Treg” for consistency.

What is the reason for very low epithelial cell numbers from both fresh and cryopreserved samples? Typically epithelial cells are the majority of retrieved cells from mucosal biopsies. Please comment on this in the manuscript.

For Fig 1f, did authors compare changes in specific populations: for example mast cell clustering in Fig 1c seems to be affected more than the others.

“HS12 had an expanded circulating cytotoxic lymphocyte 3 population, but this was not observed in the other UC-VDZ patients (Fig. 2b,c).” This is not clear from the umap plots. There are clear changes in the circulating lymphocyte 6 populations. Could the authors show this in a more quantitative manner (e.g. barplot)?

In Fig 2f, to show that there is a deregulation of the UC associated genes in VDZ-UC patients, could the authors plot the expression of the UC upregulated genes in one plot with HC, UC and VDZ-UC patients, similar to Figure 4o.

“The relative increase in goblet cells we observed was likely due to low goblet cell counts, and a relative reduction in absorptive colonocytes and intestinal stem cells (ISCs) (Extended Data Fig. 6a,b).” The authors could cite other IBD single cell papers that report Goblet cell changes, including Elmentaite et al., *Developmental Cell*, 2019 (<https://doi.org/10.1016/j.devcel.2020.11.010>) & Kanke et al., *cmgh*, 2022 (<https://doi.org/10.1016/j.jcmgh.2022.02.005>)

What were the raw numbers for the comparisons made in Figure 4g-h, I-n (how many cells per condition)? This would be useful for interpretation, given that Cluster 18 in Fig 4b is <10% of all captured cells.

“Despite limited cell counts with MIBI, tissue MNPs and fibroblasts were associated with a trend toward reduction in UC-VDZ compared to UC (Fig. 5d; Extended Data Fig. 11c).” Fig 5d only shows fibroblasts, but not MNPs, could you highlight the MNP subsets in Extended Fig 11c or show these subsets in Fig 5d.

In Figure 6f, the field of view or section type for VDZ-UC patients is different than for HC or UC representative image. This could affect the differences found. In addition, there seems to be an increased number of activated fibroblasts in the VDZ-UC patient image, therefore smaller neighbourhood numbers could be driven by the few MNP neighbourhoods inferred. How do authors account for that?

Page 12 line 22 “Net Enrichment Scores (NES)” and the figure 6i legend says “Normalised Enrichment Scores (NES)”. Please use one.

Point-by-point response:

Reviewer 1:

“Reviewer #1 (Remarks to the Author):

In this manuscript, Mennillo et al. investigate the mechanisms of vedolizumab in patients with ulcerative colitis. To this end, they use a variety of cutting-edge techniques to characterize the immune cell composition in the peripheral blood and intestinal mucosa. They claim that their data show primary effects of vedolizumab on mononuclear phagocytes and rather modest effects on lymphocytes.

Overall, this manuscript addresses an important topic that has potential relevance for the therapy of patients with IBD.

Although performed with a small number of samples, the comprehensive characterization of the samples with complementary techniques provides important insights.”

Response:

We thank the reviewer for commenting on the study's importance, its “*comprehensive characterization*”, and the “*important insights*” provided. To our knowledge, this represents the first description of a cell surface proteome cell atlas in IBD, validated using CITE-seq, CyTOF, MIBI, and CODEX on identical patient samples. Further, this is the first study in IBD employing *demuxlet* and *freemuxlet* to deconvolute scRNA-seq from pooled patient samples. Multiplexing in this manner improves data quality, facilitates inter-sample comparison, reduces batch effects, and reduces sequencing costs over 4-fold. We also show that MIBI, CODEX, and CosMx are all compatible with archived FFPE mucosal biopsies, ensuring the preservation of spatial relationships and in situ cell frequencies. We thank the reviewer for recognizing the significance of this in-depth single-cell and spatial multi-omics analysis of UC and anti-integrin therapy.

Reviewer 1:

“However, I have a few concerns, when reading the manuscript:

Most importantly, the authors should be very careful in their wording and conclusions throughout their manuscript. The data show abundance of cell populations and expression of molecules by these cells in healthy controls and patients with UC treated with VDZ or not. However, it is essential to acknowledge that observing differences in the abundance of cell populations or in the expression of molecules by these cells between patients treated with VDZ or not does not prove a direct mechanism of UC on these cells.”

Response:

We value the insights provided by the reviewer concerning mechanism-of-action and agree. We have removed all claims regarding direct mechanism-of-action. Addressing the observations by Reviewer 1, we have extensively edited the revised manuscript to clarify that we are comprehensively characterizing differences in cell subset abundance, gene expression, and protein expression, stratifying patients by colitis and VDZ therapy. We are also now emphasizing that this is the first combined description of a cell atlas in IBD with single cell and spatial resolution using transcriptomics and proteomics (CITE-seq, CyTOF, MIBI/CODEX, and CosMx) on the same patient samples.

Of note, one piece of previously published functional data that argues that the high VDZ binding of mDCs is associated with functional impacts on trafficking is that mDCs are significantly reduced in the intestines of MAdCAM-1-deficient and $\beta 7$ integrin-deficient mice (Clahsen T et al. *Clin. Immunol.* 2015). Our human studies corroborate the murine findings that mDCs require MAdCAM-1 and $\beta 7$ for intestinal migration.

In summary, we are not asserting new mechanisms of action for VDZ, although we are extensively characterizing the effects of VDZ in the tissue and in the periphery with unprecedented detail and surprising results. Our manuscript now better accentuates these strengths.

Reviewer 1:

“Such effects can be secondary to other mechanisms (which the authors automatically assume for endothelial and epithelial cells) and conclusions on the mechanisms can only be drawn, when correlated to functional investigations.”

Response:

We appreciate this insight from Reviewer 1. Indeed, functional investigations are the gold standard for elucidating mechanisms, and we have emphasized this in our revised manuscript. Nonetheless, we would like to clarify our conclusions regarding the impact of VDZ on specific cell subsets. Given that the target of VDZ is known, and we have a clear understanding of which cells express $\alpha_4\beta_7$, it is reasonable to conclude that cells lacking $\alpha_4\beta_7$ expression are not directly affected by VDZ. With that rationale, we favor indirect effects of VDZ on endothelial cells and epithelial cells, as they do not express significant amounts of $\alpha_4\beta_7$ in our dataset. The effects on these cell subsets would therefore inherently be indirect. In contrast, circulating immune subsets express high levels of $\alpha_4\beta_7$, and it is logical that VDZ acts directly on those cells.

Reviewer 1:

“Indeed, functional effects of vedolizumab on a broad variety of immune cell subsets have previously been show. Therefore, the findings of the paper are not entirely new (although of unprecedented depth), but claiming a primary mechanism on one of these subsets is not supported by the data. More precisely, the authors show a more prominent effect (whether downstream or upstream) on certain populations.”

Response:

We thank Reviewer 1 for acknowledging that our study is “*of unprecedented depth*”. Regarding novelty, we acknowledge that VDZ has been reported to affect multiple immune subsets and emphasize that this is the most comprehensive characterization of the effects of VDZ on the peripheral and mucosal immune system.

Addressing the point from Reviewer 1 that “*functional effects of vedolizumab on a broad variety of immune cell subsets have previously been show*”, the effects of VDZ on a broad range of immune cell subsets have not been as extensively addressed as one might assume. For instance, a search of PubMed using the terms “vedolizumab” AND “lymphocytes” yields 179 results, while “vedolizumab” AND “dendritic cells” yields 10 results, showing a clear bias of the literature on the effect of VDZ on lymphocytes.

Very few studies examine the effect of VDZ on circulating innate immune populations. A frequently cited study investigating VDZ binding to human leukocytes did not include dendritic cells in their analysis (Soler D. *J Pharmacol Exp Ther.* 2009). That study analyzed VDZ binding to CD4 and CD8 naïve and memory subsets, B cells, NK cells, neutrophils, eosinophils, basophils, and monocytes, but overlooked dendritic cells. Two recent mechanistic studies looked at the effects of VDZ on effector/memory and regulatory T cell subsets (Abreu M et al. *Inflamm Bowel Dis.* 2022; Becker E et al. *Gut.* 2022). Another recent study focused on the effects of VDZ on adaptive immune subsets, including plasma cells and memory T cells (Canales-Herrerias P et al. *bioRxiv.* 2023). These studies provide valuable insights into the role of VDZ on adaptive immune populations. In contrast, few studies explore the effects of VDZ on innate immune populations. One recent study demonstrated that VDZ did not consistently alter the phenotype, activation, or repertoire of lamina propria T cells by flow cytometry and TCR sequencing, but bulk transcriptomic data was consistent with a shift in MNP gene signatures (Zeissig, S. et al. *Gut.* 2019), aligning with our findings. In applying an unbiased, comprehensive approach, we can contextualize the effects of VDZ on both innate and adaptive immune subsets in a way that has not been previously described.

Further evidence of the literature's bias towards VDZ's role in lymphocyte trafficking is evident from recent discussions in top journals:

1. “Vedolizumab blocks the interaction of $\alpha_4\beta_7$ integrin with the mucosal addressin cell adhesion molecule 1, thereby inhibiting the migration of gut-homing T lymphocytes across the intestinal vascular endothelium and consequently reducing intestinal inflammation” (Travis S et al. *NEJM.* 2023).
2. “It is perceived that by interfering with gut homing, vedolizumab reduces the number of immune cells recruited to the intestine and consistently attenuates inflammation. In particular, T cells are considered an important target of vedolizumab. Intriguingly, vedolizumab blocks $\alpha_4\beta_7$ -mediated gut homing of pro-inflammatory effector T (T_{Eff}) as well as anti-inflammatory regulatory T (T_{Reg}) cells” (Becker E et al. *Gut.* 2022).

3. “lymphocyte trafficking blockade (anti- $\alpha 4\beta 7$ agents such as vedolizumab)” (Digby-Bell JL et al. *Nat Rev Gastroenterol Hepatol.* 2020).
4. “Since Vedolizumab binds to peripheral memory CD4⁺ T lymphocytes with high specificity but also to memory CD8⁺ T lymphocytes, gut-selective homing of these cell types is negatively affected” (Knauss A, et al. *Cells.* 2022).
5. “The class of anti-integrin monoclonal antibodies, which includes vedolizumab (VDZ), selectively binds the $\alpha 4\beta 7$ integrin on the surface of circulating lymphocytes, preventing their interaction with the adhesion MAdCAM-1 receptor on the endothelial cells of intestinal vasculature” (Sablich R et al. *Sci Rep.* 2023).
6. “VDZ is a humanized monoclonal antibody directed toward $\alpha 4\beta 7$ integrin. $\alpha 4\beta 7$ integrin is expressed on the surface of lymphocytes, and it interacts with mucosal addressin cell adhesion molecule-1 (MAdCAM-1), which leads to the migration of lymphocytes to the intestine” (Miyoshi J et al. *Sci Rep.* 2021).
7. “The integrin $\alpha 4\beta 7$ -expressing T lymphocytes bind to MAdCAM-1 controlling adhesion to the endothelium of postcapillary venules (also called high endothelial venules [HEVs]) in the intestine, thereby enabling these effector T lymphocytes to access the gut tissue... Targeting the homing of T lymphocytes to the gut could lead to reduction of inflammatory infiltration. Vedolizumab serves as an anti-homing integrin by blocking the binding of $\alpha 4\beta 7$ to MAdCAM-1.” (Roosenboom B et al. *Inflamm Bowel Dis.* 2023).
8. “Vedolizumab, a gut-selective anti-lymphocyte trafficking humanized monoclonal antibody that specifically binds to the $\alpha 4\beta 7$ integrin” (Kopylov U et al. *Inflamm Bowel Dis.* 2023).
9. “Vedolizumab is a humanized monoclonal antibody targeting the $\alpha 4\beta 7$ integrin heterodimer expressed on the surface of lymphocytes that mediates migration into intestinal Peyer’s patches and lamina propria via interaction with the mucosal vascular adhesion MAdCAM-1.” (Hsu P et al. *Inflamm Bowel Dis.* 2023).
10. “Vedolizumab (VDZ), a monoclonal antibody against $\alpha 4\beta 7$ integrin, inhibits lymphocyte extravasation into intestinal mucosae and is effective in ulcerative colitis (UC) and Crohn’s disease (CD).” (Abreu M et al. *Inflamm Bowel Dis.* 2022).

To summarize, we believe that our approach and findings are novel and add to the scientific literature.

Reviewer 1:

“What comes closest to a functional read-out is the analysis presented in Fig. 4o, but this does actually not support the main claim of the paper, since I can overall not observe a superior effect on MNP compared to lymphocytes in this panel.

It is important to appropriately describe and discuss these aspects in all its facets and/or to add additional data to further support the current claims.”

Response:

We are grateful to Reviewer 1 for drawing attention to our heatmap display (formerly presented as **Fig. 4o**, now **Fig. 4j** in the revised manuscript, also shown below). We included this heatmap to show that VDZ indeed affects the frequency of $\alpha 4\beta 7^+$ cells in the biopsies versus the blood across many immune subsets. There are several reasons we emphasize the effect on mDCs. The first is that mDCs exhibit the largest delta in circulating $\alpha 4\beta 7^+$ cells in VDZ-treated patients in our dataset (**Fig. 4k** in the revised manuscript, also shown below). This provides evidence for “a superior effect on MNP compared to lymphocytes in this panel”, although we agree that VDZ also targets other subsets.

Another reason we emphasize the effect on mDCs is because both the CITE-seq and CyTOF data from the initial cohort suggested reductions in total mDCs in the biopsies of patients on VDZ (**Fig. 3h, Fig. 4i, Extended Data Fig 11b**), whereas the effects on lymphocyte subsets were less consistently observed.

Reviewer 1 suggested, adding “*additional data to further support the current claims.*” Taking into account this constructive feedback, we expanded our analysis to include 31 additional unique patients, increasing the total number of patients from 12 to 43. The revised manuscript now includes new CyTOF data on an additional 7.2 million cells. These additional patients confirmed the mDCs exhibited the largest delta in the $\alpha_4\beta_7^+$ fraction circulating in VDZ-treated patients. In addition to increasing the number of patients and cells analyzed, we also ensured that these additional cases and controls address the effects of endoscopic severity, treatment duration, and treatment comparisons. The additional CyTOF are detailed in responses to Reviewers 2 & 3 below, and in the revised manuscript.

In response to Reviewer 1’s valuable suggestions, we have also incorporated substantial new data to investigate pathways associated with treatment response or non-response. These data highlight insights for MNP, stromal, and IEC subsets. We performed a retrospective, longitudinal spatial transcriptomics analysis using 1000-plex CosMx on archived FFPE biopsies, before and after treatment, in VDZ responders and non-responders. This new analysis adds spatial transcriptomics data on an additional 126,368 cells from 20 patients. In that longitudinal spatial transcriptomic dataset, we observed an increase in activated MNPs in UC patients compared to controls, with decreases in responders and increases in non-responders post-treatment. Additionally, activated MNPs and activated fibroblasts expressed transcripts that favored refractory disease. These data collectively provide compelling evidence that myeloid cell subsets are related to UC disease activity and may play a role in resistance to VDZ. The additional longitudinal CosMx spatial transcriptomics data are detailed in responses to Reviewers 2 & 3 below, and in the revised manuscript.

We thank Reviewer 1 for the insightful comments and suggestions, and we believe the revised version is significantly improved as a result.

Reviewer #2 (Remarks to the Author):

“Review on NatComm

This is a study evaluating the potential mechanism of action of the anti-integrin antibody vedolizumab by exposing a very small clinical cohort (HC, UC, UC + VDZ, each n=4) to a wide array of technologies (scRNA seq, CITE-Seq, CyTOF, CODEX). In essence their key findings, as displayed in the main figures, are:

- scRNA from peripheral blood only reveals subtle differences in leukocyte frequencies
- CITE-Seq from mucosal biopsies shows statistically significant reduction in mDC and increase in intestinal epithelial cell in VDZ-treated patients.
- CyTOF analysis show increased abundance of circulating a4b7 DCs in the peripheral blood of VDZ-treated UC patients and broad expansion of IEC in UC-VDZ, compared to UC.
- Decrease in stromal fibroblast in UC-VDZ, as shown in CODEX.
- Decreased spatial proximity between MNP and fibroblast in UC-VDZ, as shown by spatial transcriptomics.

The amount of accumulated data is impressive, and I do like the concept of functional proximity.”

Response:

We thank Reviewer 2 for commenting “ *The amount of accumulated data is impressive, and I do like the concept of functional proximity*”. As mentioned above, this represents the first description of a cell surface proteome cell atlas in IBD, validated using CITE-seq, CyTOF, MIBI, and CODEX. We thank the reviewer for recognizing the depth of single-cell and spatial multi-omics data provided for UC and anti-integrin therapy.

Reviewer 2:

“But I do have two major conceptual issue that raise my criticism:

1) There is a major flaw and structure of the clinical cohort as UC and UC-VDZ are significantly different in their endoscopic Mayo score and it is known that the degree of mucosal inflammation as the biggest effect on cellular composition, spatial proximity and transcriptomic signatures. In addition, HS12 (UC-VDZ, eMAYO 1) was on VDZ for only 2 month, meaning that the full manifestation of VDZ treatment is not yet established and underlies a much more dynamic fluctuation of mucosa inflammation as a patient on VDZ for e.g. 64 month. Bearing this heterogenous clinical phenotype in mind I seriously doubt that any observation and drawn conclusion can be attributed to VDZ effect.”

Response:

Reviewer 2 raises valid concerns regarding imperfectly matched endoscopic severity and treatment duration. Our original study included data from 12 patients. Considering the constructive feedback, we expanded our analysis to include 31 additional unique patients, increasing the total number of patients to 43. In addition to increasing the number of patients, we also ensured that these additional cases and controls address the effects of endoscopic severity and treatment duration. The additional study subjects are included in revised **Supplementary Table 1**, and they are included on the next page for reference:

Patient ID	Age	Sex	Race	Ethnicity	Disease status	UC Medication	Disease duration (y)	Duration VDZ (mo)	Prior anti-TNF exposure	Montreal classification	Mayo Endoscopic subscore	Responder to VDZ
HS1	55	M	White	Non-hispanic	HC	None	n/a	n/a	No	n/a	0	n/a
HS2	49	M	White	Non-hispanic	HC	None	n/a	n/a	No	n/a	0	n/a
HS3	28	F	White	Non-hispanic	HC	None	n/a	n/a	No	n/a	0	n/a
HS4	51	F	White	Non-hispanic	HC	None	n/a	n/a	No	n/a	0	n/a
Median (IQ1-IQ3)	50 (33-54)											
HS5	25	F	White	Non-hispanic	UC	5-ASA	15	n/a	No	E2	2	n/a
HS6	40	M	White	Non-hispanic	UC	5-ASA	25	n/a	Yes	E3	2	n/a
HS7	69	F	Black	Non-hispanic	UC	5-ASA	7	n/a	No	E2	1	n/a
HS8	51	F	White	Non-hispanic	UC	5-ASA	5	n/a	No	E1	1	n/a
Median (IQ1-IQ3)	46 (29-65)					Median	11 (7-18)					
HS9	54	M	White	Non-hispanic	UC	VDZ	25	8	Yes	E2	1	Yes
HS10	32	M	White	Non-hispanic	UC	VDZ	8	7	No	E2	1	Yes
HS11	24	F	White	Non-hispanic	UC	VDZ	8	64	Yes	E2	3	No
HS12	55	M	White	Non-hispanic	UC	VDZ	10	2	Yes	E3	1	Yes
Median (IQ1-IQ3)	43 (26-55)						9 (8-14)					
p-value	ns	ns	ns	n/a	p=0.0025	p<0.0001	ns	n/a	ns	ns	p=0.01	
HS13	29	M	Other	Hispanic	HC	None	n/a	n/a	No	n/a	0	n/a
HS14	39	M	White	Non-hispanic	HC	None	n/a	n/a	No	n/a	0	n/a
HS15	23	F	White	Non-hispanic	HC	None	n/a	n/a	No	n/a	0	n/a
HS16	60	F	White	Non-hispanic	HC	None	n/a	n/a	No	n/a	0	n/a
HS17	52	F	White	Non-hispanic	HC	None	n/a	n/a	No	n/a	0	n/a
HS18	47	F	Other	Hispanic	HC	None	n/a	n/a	No	n/a	0	n/a
Median (IQ1-IQ3)	43 (28-54)											
HS19	47	M	White	Non-hispanic	UC	ADA	27	n/a	Yes	E2	0	n/a
HS20*	31	F	White	Non-hispanic	UC	ADA	20	n/a	Yes	E2	1-2	n/a
HS21	57	M	White	Non-hispanic	UC	IFX/MTX	6	n/a	Yes	E3	0	n/a
HS22	30	F	White	Non-hispanic	UC	5-ASA	6	n/a	No	E3	0	n/a
HS23	46	M	White	Non-hispanic	UC	IFX/AZA	10	n/a	Yes	E3	1	n/a
HS30	45	F	White	Non-hispanic	UC	ADA	2	n/a	Yes	E3	0	n/a
Median (IQ1-IQ3)	46 (31-50)						8 (5-22)					
HS24	56	M	White	Non-hispanic	UC	VDZ	5.5	45	Yes	E2	0	Yes
HS25	64	M	White	Non-hispanic	UC	VDZ	37	5	No	E3	0	Yes
HS26	37	F	White	Non-hispanic	UC	VDZ	21	55	Yes	E3	0	Yes
HS27	32	F	White	Non-hispanic	UC	VDZ	1	6	No	E1	0	Yes
HS28	41	M	Asian	Non-hispanic	UC	VDZ	8	45	Yes	E1	0	Yes
Median (IQ1-IQ3)	41 (35-60)						8 (3-29)					
p-value	ns	ns	ns	n/a	p=0.0002	p<0.0001	ns	n/a	p=0.0329	ns	ns	
HS31 (HS16)	60	F	White	Non-hispanic	HC	None	n/a	n/a	No	n/a	0	n/a
HS33 (HS14)	39	M	White	Non-hispanic	HC	None	n/a	n/a	No	n/a	0	n/a
HS35 (HS15)	23	F	White	Non-hispanic	HC	None	n/a	n/a	No	n/a	0	n/a
HS37 (HS13)	29	M	Other	Hispanic	HC	None	n/a	n/a	No	n/a	0	n/a
HS39	29	F	White	Non-hispanic	HC	None	n/a	n/a	No	n/a	0	n/a
HS40	60	M	White	Non-hispanic	HC	None	n/a	n/a	No	n/a	0	n/a
HS41	55	M	White	Non-hispanic	HC	None	n/a	n/a	No	n/a	0	n/a
HS42	41	M	Other	Non-hispanic	HC	None	n/a	n/a	No	n/a	0	n/a
HS43	54	M	White	Non-hispanic	HC	None	n/a	n/a	No	n/a	0	n/a
Median (IQ1-IQ3)	41 (29-58)											
HS32	25	F	White	Non-hispanic	UC	VDZ	2	4	No	E2	pre 2 ; post 0	Yes
HS34 (HS27)	32	F	White	Non-hispanic	UC	VDZ	1	6	No	E1	pre 2; post 0	Yes
HS36	48	M	Asian	Non-hispanic	UC	VDZ	3	2	Yes	E2	pre 3; post 0	Yes
HS38 (HS9)	54	M	White	Non-hispanic	UC	VDZ	25	7	Yes	E3	pre 2; post 0	Yes
HS44	30	F	White	Non-hispanic	UC	VDZ	2	2	No	E1	pre 1	Yes
HS45	41	M	White	Non-hispanic	UC	VDZ	14	19	Yes	E2	pre 1; post 3	No
HS46	41	n/a	White	Non-hispanic	UC	VDZ	21	2	Yes	E2	pre 3; post 3	No
HS47	30	M	White	Non-hispanic	UC	VDZ	11	5	Yes	E2	post 3	No
HS48	78	M	White	Non-hispanic	UC	VDZ	8	6	Yes	E3	pre 2; post 2	No
HS49	37	F	White	Non-hispanic	UC	VDZ	2	3	Yes	E2	pre 2; post 3	No
HS50	23	M	White	Non-hispanic	UC	VDZ	0	5	Yes	E2	pre 2; post 2	No
Median (IQ1-IQ3)	37 (30-78)						3 (2-14)					
p-value	ns	ns	ns	n/a	p<0.0001	p<0.0001	n/a	n/a	p=0.0010	ns	pre, ns; post, p=0.0016	

Supplementary Table 1. Baseline demographic and clinical data for study participants. Categorical variables were analyzed by Chi-square test and continuous variables were compared using one-way ANOVA with FDR correction or Mann-Whitney test where appropriate. ns, not significant; n/a, not applicable; pre, pre-VDZ treatment; post, post-VDZ treatment. *HS20 had a short segment of moderate proctitis, otherwise in endoscopic remission with no change in therapy. HS31, HS33, HS35, HS37, HS34, HS38 were additional samples from HS16, HS14, HS15, HS13, HS27, and HS9, respectively.

Patients HS13-HS28 were all in a stable maintenance phase of therapy for a minimum of 5 months. This group therefore addresses concerns about the influence of timepoint on immunophenotype across groups. Moreover, 10 out of 11 patients in this second group were in endoscopic remission (Mayo 0-1), and the other patient was in near endoscopic remission, with only a very short segment of moderate rectal inflammation remaining that did not prompt a change in therapy. By ensuring that these patients were in near endoscopic remission, we have sought to allay concerns related to matching endoscopic severity. Furthermore, in addressing concerns about treatment comparisons, it is notable that 5 of the 6 patients in our comparator group were under anti-TNF therapy, providing a second biologic as a comparison group.

For this additional case-control study, primarily focusing on patients largely in remission and on maintenance therapy, we analyzed 1,407,739 live cells from colonic biopsies and 5,781,249 live cells from peripheral blood using our CyTOF panel. Our findings consistently show that the cell subset with the highest circulating percentage of $\alpha_4\beta_7^+$ cells are myeloid dendritic cells (mDCs) in VDZ-treated UC patients (**Extended Data Fig 12, and shown below**). Patients on VDZ exhibited increases in the circulating $\alpha_4\beta_7^+$ fractions of multiple cell subsets, but up to 80% of circulating mDCs were $\alpha_4\beta_7^+$. This is true whether patients have endoscopically active or inactive disease, and whether they have been on therapy for over 5 months. These results confirm that higher levels of circulating $\alpha_4\beta_7^+$ mDCs observed in VDZ-treated patients persist during stable maintenance therapy, in periods of remission, and when compared against anti-TNF agents.

Here is the comparison to the initial case-control group (Fig. 4k).

We emphasize our findings in relation to mDCs because it is the most prominent among circulating immune cell subsets. While much of the existing literature focuses prominently on VDZ's impact on CD4+ T cell subsets or Tregs, the influence of VDZ on mDCs and other innate immune subsets remains relatively unexplored. Our unbiased, comprehensive CyTOF panel highlights this shift. We acknowledge and agree that VDZ has significant effects on other cell subsets, such as CD8 T cells, NKT cells, gd T cells, NK cells, and plasma cells, which we do not intend to minimize. Rather, our focus on mDCs is due to the large delta in the $\alpha_4\beta_7^+$ fraction circulating in VDZ-treated patients, and because very few studies examine the effect of VDZ on circulating innate immune populations. One frequently cited study that investigated VDZ binding to human leukocytes did not include dendritic cells in their analysis (Soler D. *J Pharmacol Exp Ther.* 2009). That study analyzed VDZ

binding to CD4 and CD8 naïve and memory subsets, B cells, NK cells, neutrophils, eosinophils, basophils, and monocytes, but not dendritic cells.

When we examine the shift of $\alpha_4\beta_7^+$ cells from the tissue to the blood in VDZ-treated UC patients in this additional cohort, we again observe effects across multiple cell subsets. Through unsupervised clustering, we can readily discern patients that are on VDZ (**Extended Data Fig. 12e**, and shown below). mDCs are significantly affected in these additional patients (**Extended Data Fig. 12e,f**). However, the remission cohort did not exhibit a significant decrease in overall mDC cell frequency in the tissue of VDZ-treated patients (**Extended Data Fig. 12d** and shown below). We hypothesize that this could be attributed to the reduced frequency of these cells in patients in remission.

As noted above, Reviewer 2 also raised the concern “In addition, HS12 (UC-VDZ, eMAYO 1) was on VDZ for only 2 month, meaning that the full manifestation of VDZ treatment is not yet established and underlies a much more dynamic fluctuation of mucosa inflammation as a patient on VDZ for e.g. 64 month”. We appreciate this observation. It is valuable to place this concern within the context of landmark clinical trials of VDZ. As Reviewer 2 might recall, the phase 3 clinical trial that led to FDA-approval for Vedolizumab in UC had a primary endpoint of induction at week 6 (Feagan BG et al. *NEJM*. 2013). In that trial, response rates at week 6 were 47.1%, compared to 25.5% for placebo. Similarly, the rate of mucosal healing at week 6 was 40.9% compared to 24.8% for placebo. Week 6 is when the 3rd IV induction dose is administered. It is also important to highlight that by week 52, mucosal healing rates were 51.6%, compared to 19.8% for placebo. Meaning that many patients with mucosal healing on VDZ at week 52 already had shown such improvement by week 6. In our study, patient HS12 analyzed at week 8 had completed all 3 induction doses. Considering the clinical efficacy and mucosal healing associated with VDZ by week 8 in many patients, we propose that this represents an appropriate timepoint for measuring VDZ’s effects. While we acknowledge the possible benefits of extended therapy, the demonstrated clinical significance at week 8 supports including this timepoint. Additionally, we now include patients who were in stable maintenance therapy, as described above.

In summary, our additional CyTOF dataset includes patients who are in remission and on stable maintenance therapy, with comparisons to both anti-TNF and 5-ASA therapies. These studies confirm a significant increase in circulating $\alpha_4\beta_7^+$ mDCs in patients on VDZ. These additional participants effectively address the initial concerns raised by Reviewer 2 regarding the timepoints and endoscopic severity. Importantly, the newly incorporated CyTOF data incorporate an analysis of 7.2 million total cells, greatly enriching our single-cell proteomic data.

In addition to these CyTOF data, we also outline below a new spatial transcriptomics CosMx analysis of archived FFPE specimens, pre- and post-treatment, in VDZ responders and non-responders. Those additional data include patients with balanced endoscopic disease activity prior to VDZ therapy, further controlling for endoscopic severity (**Supplementary Table 1**). Furthermore, by examining samples before and after therapy across both responder and non-responder groups, we account for the potential effects of varying timepoints.

Reviewer 2:

"2) There is no stringent line in which the data of each individual technical approach are tried consolidating into a bigger mechanistic picture on the MOA of VDZ and how this knowledge might be used to improve current treatment strategies with VDZ. With the exception of one example, a publicly available data set on VDZ response, which is not cited in the manuscript (see page 12, line 18), there is no external validation. Bearing in mind that the authors deal with n=3-4 samples/group this is surprising. In the particular example the author aim to validate their own identified gene signature in a cohort of VDZ response but their own dataset is not designed to test effects on VDZ response."

Response:

We value the insights provided by Reviewer 2 concerning mechanism-of-action, VDZ response, sample size, and external validation. We have addressed these concerns in our extensively revised manuscript.

Addressing the observations by Reviewer 2, we have extensively edited the revised manuscript to clarify that we are comprehensively characterizing differences in cell subset abundance, gene expression, and protein expression, stratifying patients by colitis and VDZ therapy. We are also now emphasizing that this is the first combined description of a cell atlas in IBD with single cell and spatial resolution using transcriptomics and proteomics (CITE-seq, CyTOF, MIBI/CODEX, and CosMx) on the same patient samples. We are not asserting new mechanisms of action for VDZ, although we are extensively characterizing the effects of VDZ in the tissue and in the periphery with unprecedented detail and surprising results.

In response to Reviewer 2's valuable suggestions, we have incorporated substantial new data to investigate pathways associated with treatment response or non-response. These data highlight insights for MNP, stromal, and IEC subsets. To consolidate our understanding of the peripheral and tissue effects of VDZ in patients with UC *"into a bigger mechanistic picture on the MOA of VDZ and how this knowledge might be used to improve current treatment strategies with VDZ"*, we performed a new, retrospective, longitudinal spatial transcriptomics analysis using 1000-plex CosMx on archived FFPE biopsies, before and after treatment, in VDZ responders and non-responders. Our goal was to discern potential pathways that either facilitate or antagonize VDZ efficacy. The purpose of this new analysis was to determine if there are any differences in the pre-treatment biopsies of patients that respond to VDZ versus those that are refractory, and whether we would observe alterations in cell subset abundance longitudinally (**Fig. 7a** and schematic shown above). Whereas our initial submission focused solely on post-treatment insights, we now broaden our scope in the revised manuscript to include a retrospective analysis of pre-treatment samples from both VDZ responders and non-responders.

The data quality using retrospectively identified, clinical archived FFPE samples was slightly inferior to that of prospectively collected FFPE samples, likely reflecting sample age and storage. After filtering and quality control, approximately 80% of the cells were annotated, and 20% of cells were left unassigned. This was due to diminished expression of landmark genes and some ambiguity in cell identification for those 20% of cells. Importantly, landmark genes for the myeloid, stromal, and epithelial compartments were expressed at high levels, and we could confidently annotate those subsets. (**Extended Data Fig 14a** and shown below). Building on our initial spatial transcriptomics dataset of 48,783 cells, we now add 126,368 cells, for a total of 175,151 cells. These patients were also balanced for disease severity prior to treatment (**Supplementary Table 1**).

We observed an increase in activated MNPs in UC patients compared to controls, with a decrease in responders and an increase in non-responders post-treatment (**Fig 7b** and shown below). IECs expressing high levels of MHCII were similarly elevated in active colitis compared to controls, with an apparent reduction in responders post-treatment (**Fig 7c** and shown below). Neighborhood enrichment analysis revealed trends toward increased proximity of activated fibroblast and activated MNP subsets in active colitis, and apparent reduction post-treatment (**Fig 7d** and shown below). However, these observations were not statistically significant, and not clearly associated with response or non-response to VDZ.

Pre-treatment differences are the most relevant for developing precision medicine algorithms. Therefore, we performed pseudobulk DE gene analysis in pre-treatment FFPE biopsies from non-responders versus responders. Stromal and MNP genes including *MMP1*, *MMP2*, and *THBS1* were among the top differentially expressed genes in VDZ non-responders, while genes associated with the IEC crypt base including *REG1A*,

OLFM4, *AGR2*, *SPINK1*, and *LYZ* were associated with response to VDZ (Fig 7e and shown below). IgA plasma cell associated genes were also associated with response to VDZ (Fig 7e and shown below).

Spatial scatter plots of subsets of these cells and transcripts suggested that the abundance and activation of fibroblasts and MNPs were higher in non-responders, while a robust IEC crypt base was associated with response to VDZ (Fig 7f-i and shown below).

As the reviewer is aware, colonoscopy with biopsy is standard-of-care in the management of UC, so all patients have FFPE specimens archived in Pathology departments. One barrier to validating and implementing multi-omic biomarkers is the need for specialized prospective sample collection such as cryopreserving biopsies or collecting blood or tissue in RNAlater. Identifying spatial signatures in routine, archived clinical FFPE tissue sections could potentially allow for more rapid biomarker validation and dissemination.

We believe these experiments directly address appropriate concerns raised by Reviewer 2, “*Bearing in mind that the authors deal with n=3-4 samples/group this is surprising. In the particular example the author aim to validate their own identified gene signature in a cohort of VDZ response but their own dataset is not designed to test effects on VDZ response.*” Our revised manuscript now includes significantly more patients and assesses response and non-response to VDZ. To further emphasis this point, we also include the following heatmap displaying expression z-scores for the most differentially regulated genes in responders and non-responders (**Extended Data 14b** and shown below).

These experiments further address the important concerns from Reviewer 2: “2) *There is no stringent line in which the data of each individual technical approach are tried consolidating into a bigger mechanistic picture on the MOA of VDZ and how this knowledge might be used to improve current treatment strategies with VDZ.*” Here we show that the tissue MNP and stromal subsets that we identified in our initial dataset express genes associated with non-response to VDZ, while IEC-specific genes are associated with VDZ response. Furthermore, the identification of these gene signatures in archived FFPE tissue underscores the potential clinical applicability of our findings. Spatial transcriptomic panels built around these pathways could conceivably “*improve current treatment strategies with VDZ*”. It is also clear that an IEC crypt base signature of elevated *OLFM4* and *AGR2* could reflect higher Wnt and Notch activity in responders. This observation provides potential pathways of augmenting mucosal healing.

We then turned our attention to important comments from Reviewer 2 regarding external validation. To further validate the association of MNP, stromal, and IEC gene signatures with VDZ response and non-response, we performed gene set enrichment analysis (GSEA), as described in our original manuscript, referencing a longitudinal, publicly available, bulk transcriptomic dataset of patients pre- and post-treatment with VDZ (Arijs, I. *et al. Gut.* 2018). We included landmark genes from our multi-omics analysis (**Supplementary Table 5**). As we presented in our original manuscript, VDZ responders (n=9) exhibited broad reductions in immune and activated stromal Normalized Enrichment Scores (NES), with epithelial gene set enrichment post-treatment, consistent with reduced inflammation and mucosal healing (**Fig. 8a** and shown below). In contrast, VDZ non-responders (n=5) exhibited a marked pre-treatment cytotoxic lymphocyte signature, and persistent activated and S2 fibroblast gene signatures post-treatment. There was a notable absence of epithelial enrichment post-treatment, suggesting that high initial cytotoxic lymphocyte injury and persistent stromal tissue inflammation prevents mucosal healing (**Fig. 8a** and shown below). Interestingly, the reduction in immune subsets in VDZ non-responders was smaller and not statistically significant compared to VDZ responders. VDZ non-responders (n=9) were differentiated from responders (n=11) by pre-treatment enrichment for endothelial, activated fibroblast, neutrophil, macrophage, and monocyte signatures (**Fig. 8b** and shown below). Gene signatures investigated were clearly distinguished by leading edge analysis (**Fig. 8c** shown below). These data are shown here to highlight the initial GSEA validation analysis using an external dataset.

Building on that initial validation analysis, our revised manuscript now includes GSEA using VDZ response and non-response signatures selected from our longitudinal spatial transcriptomics analysis of FFPE biopsies (Fig. 8d,e, and shown below). These FFPE gene signatures were validated in the external, publicly available bulk transcriptomic dataset.

Interestingly, the VDZ non-response signature was also significantly enriched in Infliximab non-responders prior to treatment (pre-IFX). This categorizes these non-response genes as markers of non-response to both treatments (Fig. 8f,g and shown below). This is further validation of our approach, as all VDZ-treated patients in that study had previously been exposed to anti-TNF therapy, similar to our VDZ non-responders (Supplementary Table 1). In contrast, the pre-VDZ-response signature was specific to VDZ, and not associated with response to IFX (Fig. 8f,g).

Mechanistically, these data suggest that VDZ non-responders have higher pre-treatment tissue innate immune and activated stromal subset inflammation, and that these cell subsets may drive inflammatory cell trafficking via $\alpha_4\beta_7$ -independent pathways. Conversely, a robust IEC crypt base signature pre-treatment is associated with response to VDZ and mucosal healing. The crypt base genes also suggest that both Wnt and Notch signaling pathways are active in pre-treatment responders, which may explain their better treatment response.

Per Reviewer 2's concerns "With the exception of one example, a publicly available data set on VDZ response, which is not cited in the manuscript (see page 12, line 18), there is no external validation." we have ensured that this dataset is now cited at every mention throughout the revised manuscript. Further, our revised manuscript addresses both internal and external validation with additional data. By expanding our patient cohort and augmenting our findings with additional CyTOF and spatial transcriptomics (CosMx) experiments, we have significantly enhanced the depth and rigor of our study.

We also wish to clarify our rationale for choosing the Arijs, I. *et al. Gut*. 2018 study for external validation by GSEA. There are limited publicly available, longitudinal transcriptomic data for vedolizumab treated patients (table below). Our selection of Arijs et al., 2018 was driven by the following considerations:

1. It has been validated in several IBD studies using VDZ (Soendrgaard et al., *BMJ Open Gastroenterol* (2018); Verstockt et al., *Clin Gastroenterol Hepatol* (2020); Friedrich et al., *Nat Med* (2021); Gubatan et al., *J Crohns Colitis* (2021); Singh et al., *Int J Colorectal Dis* (2022).
2. It is one of the few longitudinal study available where we can analyze both responders and non-responders, pre- and post-therapy.

Reference	Availability	GSE ID	Disease	Longitudinal	specimen	Type of data	External Validation
Rath et al., 2018	Yes	SRP151738	UC/CD	Yes	Biopsy	RNA-seq data	-
Verstockt et al., 2020	Yes	E-MTAB-7845	UC/CD	No; collection only pre-treatment	Biopsy	RNA-seq data	Arijs et al., 2018
Gubatan et al., 2021	No	-	UC/CD	No; collection VDZ naïve	Blood/Biopsy	CyTOF data	Arijs et al., 2018
Lee et al., 2021	Yes	PRJNA685168	UC/CD	No; collection only pre-treatment	Blood/Stool	Metagenome and metabolomic sequencing	

In Rath et al. *Frontiers in Immunology*. 2018, only 2 remitters and non-remitters were analyzed by bulk RNA-seq, limiting its utility for validation. Nonetheless, VDZ non-responders in the Rath et al study exhibited persistently high levels of *MMP1*, *MMP2*, *SPP1*, and *ITGA5*. In contrast, responders demonstrated elevated pre-treatment levels of *REG1A*, as well as other genes expressed in the IEC crypt base including *REG1B*, *URAD*, *SAA2*. Those data are in line with our current observations.

Reviewer 2:

"Minor:

Figure 6f/h: The authors show a representative picture of decreased spatial proximity between activated fibroblast and MNP in UC-VDZ patients. However, the shown IHC slide is counterintuitive as the overall number of activated fibroblast is dramatically increased."

Response:

We thank Reviewer 2 for this observation regarding **Fig 6f/h**. We now include a different UC image to compare to the UC-VDZ patients to address this concern. Our new extended spatial transcriptomics analysis in the revised manuscript underscores the variability in neighborhood enrichment z-scores. The proximity analysis is highly dependent on chosen radius and cell frequency. Consequently, we are now reporting a trend toward proximity of activated fibroblasts and activated MNPs in active colitis. In our revised manuscript, we have switched from a parametric to a non-parametric analysis of z-scores, as the enrichment scores are not normally distributed. Given the small sample size and nonparametric analysis, the observed differences do not reach statistical significance. We show the updated scatter plots below.

We thank Reviewer 2 for the helpful comments and suggestions, and we believe the revised version has significantly improved as a result.

Reviewer #3 (Remarks to the Author):

“Mennillo et al compare single cell transcriptomic, proteomic and spatial profiling of blood and colonic biopsies in UC patients treated with aminosalicylates or Vedolizumab, a biologic which is thought to act through blocking leukocyte trafficking to the intestine. The treatment resulted in cell type changes indicating mucosal healing including increase in epithelial cell abundance and reduction in activated fibroblasts and inflammatory macrophages in the colon samples. The key finding is that while in general circulating leukocyte frequencies were stable in blood and colon, VDZ treatment blocks trafficking of integrin expressing MNP cells from blood to the colon. Authors propose an interaction of activated fibroblasts and MNPs upon VDZ treatment and observe similar populations in VDZ non-responders, but the mechanism could be interrogated in more detail. The main advantage is that the data presented is of high quality and presents an interesting dataset to interrogate the immune cell trafficking in IBD. The main weakness is the lack of more mechanistic insight into the cellular changes. We raise the following questions for the authors to comment on and suggestions for further analysis.”

Response:

We thank Reviewer 3 for commenting "*The main advantage is that the data presented is of high quality and presents an interesting dataset to interrogate the immune cell trafficking in IBD.*" As mentioned above, this represents the first description of a cell surface proteome cell atlas in IBD, validated using CITE-seq, CyTOF, MIBI, and CODEX on identical patient samples and investigating a specific therapy. We are also adding a new longitudinal spatial transcriptomics analysis in the revised manuscript.

Reviewer 3:

“Major comments:

What was the time period between VDZ treatment and blood and mucosa sample collection in patients? Would it be possible that immediate changes in lymphocyte trafficking were not captured in the VDZ treated patients and that what the authors observe is mucosal healing as a consequence of balancing the influx of lymphocytes?

We recognise that patient recruitment is a complicated process and that perfect controls do not exist. However, could the authors comment in the manuscript what are the effects of 2-AZA treatment and why were they selected as untreated patient control?”

Response:

Reviewer 3 raises important points regarding imperfectly matched treatment duration and treatment comparisons. As mentioned above, we expanded our analysis to include 31 additional unique patients, increasing the total number of patients to 43. In addition to increasing the number of patients, we also ensured that these additional cases and controls address the effects of treatment duration and treatment comparisons. The revised manuscript now includes new CyTOF data on an additional 7.2 million cells. The additional study subjects are included in revised **Supplementary Table 1**, and they are included on the next page for reference:

Patient ID	Age	Sex	Race	Ethnicity	Disease status	UC Medication	Disease duration (y)	Duration VDZ (mo)	Prior anti-TNF exposure	Montreal classification	Mayo Endoscopic subscore	Responder to VDZ
HS1	55	M	White	Non-hispanic	HC	None	n/a	n/a	No	n/a	0	n/a
HS2	49	M	White	Non-hispanic	HC	None	n/a	n/a	No	n/a	0	n/a
HS3	28	F	White	Non-hispanic	HC	None	n/a	n/a	No	n/a	0	n/a
HS4	51	F	White	Non-hispanic	HC	None	n/a	n/a	No	n/a	0	n/a
Median (IQ1-IQ3)	50 (33-54)											
HS5	25	F	White	Non-hispanic	UC	5-ASA	15	n/a	No	E2	2	n/a
HS6	40	M	White	Non-hispanic	UC	5-ASA	25	n/a	Yes	E3	2	n/a
HS7	69	F	Black	Non-hispanic	UC	5-ASA	7	n/a	No	E2	1	n/a
HS8	51	F	White	Non-hispanic	UC	5-ASA	5	n/a	No	E1	1	n/a
Median (IQ1-IQ3)	46 (29-65)					Median	11 (7-18)					
HS9	54	M	White	Non-hispanic	UC	VDZ	25	8	Yes	E2	1	Yes
HS10	32	M	White	Non-hispanic	UC	VDZ	8	7	No	E2	1	Yes
HS11	24	F	White	Non-hispanic	UC	VDZ	8	64	Yes	E2	3	No
HS12	55	M	White	Non-hispanic	UC	VDZ	10	2	Yes	E3	1	Yes
Median (IQ1-IQ3)	43 (26-55)						9 (8-14)					
p-value	ns	ns	ns	n/a	p=0.0025	p<0.0001	ns	n/a	ns	ns	p=0.01	
HS13	29	M	Other	Hispanic	HC	None	n/a	n/a	No	n/a	0	n/a
HS14	39	M	White	Non-hispanic	HC	None	n/a	n/a	No	n/a	0	n/a
HS15	23	F	White	Non-hispanic	HC	None	n/a	n/a	No	n/a	0	n/a
HS16	60	F	White	Non-hispanic	HC	None	n/a	n/a	No	n/a	0	n/a
HS17	52	F	White	Non-hispanic	HC	None	n/a	n/a	No	n/a	0	n/a
HS18	47	F	Other	Hispanic	HC	None	n/a	n/a	No	n/a	0	n/a
Median (IQ1-IQ3)	43 (28-54)											
HS19	47	M	White	Non-hispanic	UC	ADA	27	n/a	Yes	E2	0	n/a
HS20*	31	F	White	Non-hispanic	UC	ADA	20	n/a	Yes	E2	1-2	n/a
HS21	57	M	White	Non-hispanic	UC	IFX/MTX	6	n/a	Yes	E3	0	n/a
HS22	30	F	White	Non-hispanic	UC	5-ASA	6	n/a	No	E3	0	n/a
HS23	46	M	White	Non-hispanic	UC	IFX/AZA	10	n/a	Yes	E3	1	n/a
HS30	45	F	White	Non-hispanic	UC	ADA	2	n/a	Yes	E3	0	n/a
Median (IQ1-IQ3)	46 (31-50)						8 (5-22)					
HS24	56	M	White	Non-hispanic	UC	VDZ	5.5	45	Yes	E2	0	Yes
HS25	64	M	White	Non-hispanic	UC	VDZ	37	5	No	E3	0	Yes
HS26	37	F	White	Non-hispanic	UC	VDZ	21	55	Yes	E3	0	Yes
HS27	32	F	White	Non-hispanic	UC	VDZ	1	6	No	E1	0	Yes
HS28	41	M	Asian	Non-hispanic	UC	VDZ	8	45	Yes	E1	0	Yes
Median (IQ1-IQ3)	41 (35-60)						8 (3-29)					
p-value	ns	ns	ns	n/a	p=0.0002	p<0.0001	ns	n/a	p=0.0329	ns	ns	
HS31 (HS16)	60	F	White	Non-hispanic	HC	None	n/a	n/a	No	n/a	0	n/a
HS33 (HS14)	39	M	White	Non-hispanic	HC	None	n/a	n/a	No	n/a	0	n/a
HS35 (HS15)	23	F	White	Non-hispanic	HC	None	n/a	n/a	No	n/a	0	n/a
HS37 (HS13)	29	M	Other	Hispanic	HC	None	n/a	n/a	No	n/a	0	n/a
HS39	29	F	White	Non-hispanic	HC	None	n/a	n/a	No	n/a	0	n/a
HS40	60	M	White	Non-hispanic	HC	None	n/a	n/a	No	n/a	0	n/a
HS41	55	M	White	Non-hispanic	HC	None	n/a	n/a	No	n/a	0	n/a
HS42	41	M	Other	Non-hispanic	HC	None	n/a	n/a	No	n/a	0	n/a
HS43	54	M	White	Non-hispanic	HC	None	n/a	n/a	No	n/a	0	n/a
Median (IQ1-IQ3)	41 (29-58)											
HS32	25	F	White	Non-hispanic	UC	VDZ	2	4	No	E2	pre 2 ; post 0	Yes
HS34 (HS27)	32	F	White	Non-hispanic	UC	VDZ	1	6	No	E1	pre 2; post 0	Yes
HS36	48	M	Asian	Non-hispanic	UC	VDZ	3	2	Yes	E2	pre 3; post 0	Yes
HS38 (HS9)	54	M	White	Non-hispanic	UC	VDZ	25	7	Yes	E3	pre 2; post 0	Yes
HS44	30	F	White	Non-hispanic	UC	VDZ	2	2	No	E1	pre 1	Yes
HS45	41	M	White	Non-hispanic	UC	VDZ	14	19	Yes	E2	pre 1; post 3	No
HS46	41	n/a	White	Non-hispanic	UC	VDZ	21	2	Yes	E2	pre 3; post 3	No
HS47	30	M	White	Non-hispanic	UC	VDZ	11	5	Yes	E2	post 3	No
HS48	78	M	White	Non-hispanic	UC	VDZ	8	6	Yes	E3	pre 2; post 2	No
HS49	37	F	White	Non-hispanic	UC	VDZ	2	3	Yes	E2	pre 2; post 3	No
HS50	23	M	White	Non-hispanic	UC	VDZ	0	5	Yes	E2	pre 2; post 2	No
Median (IQ1-IQ3)	37 (30-78)						3 (2-14)					
p-value	ns	ns	ns	n/a	p<0.0001	p<0.0001	n/a	n/a	p=0.0010	ns	pre, ns; post, p=0.0016	

Supplementary Table 1. Baseline demographic and clinical data for study participants. Categorical variables were analyzed by Chi-square test and continuous variables were compared using one-way ANOVA with FDR correction or Mann-Whitney test where appropriate. ns, not significant; n/a, not applicable; pre, pre-VDZ treatment; post, post-VDZ treatment. *HS20 had a short segment of moderate proctitis, otherwise in endoscopic remission with no change in therapy. HS31, HS33, HS35, HS37, HS34, HS38 were additional samples from HS16, HS14, HS15, HS13, HS27, and HS9, respectively.

Reviewer 3 stated “We recognise that patient recruitment is a complicated process and that perfect controls do not exist”. Despite the challenge of recruiting additional patients for these types of studies, we agreed that controlling for endoscopic severity and treatment were essential.

Patients HS13-HS28 were all in a stable maintenance phase of therapy for a minimum of 5 months. This group therefore addresses concerns about the influence of timepoint on immunophenotype across groups. Moreover, 10 out of 11 patients in this second group were in endoscopic remission (Mayo 0-1), and the other patient was in near endoscopic remission, with only a very short segment of moderate rectal inflammation remaining that

did not prompt a change in therapy. By ensuring that these patients were in a stable endoscopic remission (Mayo 0-1), we have sought to allay concerns related to matching endoscopic severity. Furthermore, in addressing concerns about treatment comparisons, it is notable that 5 of the 6 patients in our comparator group were under anti-TNF therapy, providing a second biologic as a comparison group.

For this additional case-control study, primarily focusing on patients largely in remission and on maintenance therapy, we analyzed 1,407,739 live cells from colonic biopsies and 5,781,249 live cells from peripheral blood using our CyTOF panel. Our findings consistently show that the cell subset with the highest circulating percentage of $\alpha_4\beta_7^+$ cells are myeloid dendritic cells (mDCs) in VDZ-treated UC patients (**Extended Data Fig 12, and shown below**). Patients on VDZ exhibited increases in the circulating $\alpha_4\beta_7^+$ fractions of multiple cell subsets, but up to 80% of circulating mDCs were $\alpha_4\beta_7^+$. This is true whether patients have endoscopically active or inactive disease, and whether they have been on therapy for over 5 months. These results confirm that higher levels of circulating $\alpha_4\beta_7^+$ mDCs observed in VDZ-treated patients persist during stable maintenance therapy, in periods of remission, and when compared against anti-TNF agents.

Here again is the comparison to the initial case-control group (**Fig. 4k**).

We emphasize our findings in relation to mDCs because it is the most prominent among circulating immune cell subsets. While much of the existing literature focuses prominently on VDZ's impact on CD4+ T cell subsets or Tregs, the influence of VDZ on mDCs and other innate immune subsets remains relatively unexplored. Our unbiased, comprehensive CyTOF panel highlights this shift. We acknowledge and agree that VDZ has

significant effects on other cell subsets, such as CD8 T cells, NKT cells, gd T cells, NK cells, and plasma cells, which we do not intend to minimize. We have also revised the title and abstract to avoid over-emphasizing effects on MNP subsets. Rather, our focus on mDCs is due to the large delta in the $\alpha_4\beta_7^+$ fraction circulating in VDZ-treated patients, and because very few studies examine the effect of VDZ on circulating innate immune populations.

When we examine the shift of $\alpha_4\beta_7^+$ cells from the tissue to the blood in VDZ-treated UC patients in this additional cohort, we again observe effects across multiple cell subsets. Through unsupervised clustering, we can readily discern patients that are on VDZ (**Extended Data Fig. 12e**, and shown below). mDCs are significantly affected in these additional patients (**Extended Data Fig. 12e,f**). However, the remission cohort did not exhibit a significant decrease in overall mDC cell frequency in the tissue of VDZ-treated patients (**Extended Data Fig. 12d** and shown below). We hypothesize that this could be attributed to the reduced frequency of these cells in patients in remission.

We cannot exclude an immediate early effect on other subsets. Nevertheless, our data indicate that circulating mDCs and other tissue MNPs are significantly affected whether looking immediately post-induction or in stable maintenance phases of treatment.

Specifically with regard to the comment, “*However, could the authors comment in the manuscript what are the effects of 2-AZA treatment and why were they selected as untreated patient control?*”, we believe Reviewer 3 is referring to 5-ASA therapy (5-aminosalicylates). The 4 UC patients on VDZ and 5-ASA were chosen because they all had at least some endoscopic disease activity at the time of collection, even if mild-to-moderate. 5-ASA is first-line for mild-to-moderate UC, while VDZ is used for moderate-to-severe UC. This creates some inherent imbalance, but we think it is insufficient to explain such a pronounced effect on mDCs in the circulation. Importantly, our new CyTOF data includes patients who were on anti-TNF as a comparison group, and we observed similar results in the circulating $\alpha_4\beta_7^+$ immune subsets.

In response to Reviewer 3's valuable suggestions, we also have performed a longitudinal spatial transcriptomics analysis of patients before and after VDZ treatment, more closely matched for endoscopic disease activity at baseline (**Fig. 7a** and shown right). The patients were categorized as VDZ responders and non-responders. This additional analysis adds spatial transcriptomics data on an additional 126,368 cells from 20 patients.

As mentioned above in response to Reviewer 2, the data quality using retrospectively identified, clinical archived FFPE samples was slightly inferior to that of prospectively collected FFPE samples, likely reflecting sample age and storage. After filtering and quality control, approximately 80% of the cells were annotated, and 20% of cells were left unassigned. This was due to diminished expression of landmark genes and some ambiguity in cell identification for those 20% of cells. Importantly, landmark genes for the myeloid, stromal, and epithelial compartments were expressed at high levels, and we could confidently annotate those subsets. (**Extended Data Fig 14a** and shown below). Building on our initial spatial transcriptomics dataset of 48,783 cells, we now add 126,368 cells, for a total of 175,151 cells. These patients were also balanced for disease severity prior to treatment (**Supplementary Table 1**).

We observed an increase in activated MNP in UC patients compared to controls, with a decrease in responders and an increase in non-responders post-treatment (**Fig 7b** and shown below). IECs expressing high levels of MHCII were similarly elevated in active colitis compared to controls, with an apparent reduction in responders post-treatment (**Fig 7c** and shown below). Neighborhood enrichment analysis revealed trends toward increased proximity of activated fibroblast and activated MNP subsets in active colitis, and apparent reduction post-treatment (**Fig 7d** and shown below). However, these observations were not statistically significant, and not clearly associated with response or non-response to VDZ

Pre-treatment differences are the most relevant for developing precision medicine algorithms. Therefore, we performed pseudobulk DE gene analysis in pre-treatment FFPE biopsies from non-responders versus responders. Stromal and MNP genes including *MMP1*, *MMP2*, and *THBS1* were among the top differentially expressed genes in VDZ non-responders, while genes associated with the IEC crypt base including *REG1A*, *OLFM4*, *AGR2*, *SPINK1*, and *LYZ* were associated with response to VDZ (Fig 7e and shown below). IgA plasma cell associated genes were also associated with response to VDZ (Fig 7e and shown below).

Spatial scatter plots of subsets of these cells and transcripts suggested that the activation of fibroblasts and MNPs were higher in non-responders, while a robust IEC crypt base was associated with response to VDZ (Fig 7f-I and shown below).

As the reviewer is aware, colonoscopy with biopsy is standard-of-care in the management of UC, so all patients have FFPE specimens archived in Pathology departments. One barrier to validating and implementing multi-omic biomarkers is the need for specialized prospective sample collection such as cryopreserving biopsies or collecting blood or tissue in RNAlater. Identifying spatial signatures in routine, archived clinical FFPE tissue sections could potentially allow for more rapid biomarker validation and dissemination.

In summary, our additional CyTOF and spatial transcriptomics data help address concerns regarding treatment duration and relevant controls.

Reviewer 3:

“How do these patients compare with other UC patients on the single cell level such from published studies (e.g. Smillie et al., Cell, 2019)?”

Response:

Thank you for highlighting this important study. Smillie et al did not analyze patients on Vedolizumab, and so a detailed comparison would be tangential to our study. Nevertheless, we observe similar cell subsets in our mucosal biopsies, and we discuss these parallels in our CITE-seq results section. Furthermore, we would like to clarify that the Smillie et al. study employed a serial single-plex scRNA-seq approach, which introduced significant batch effects. The batch effects are evident in the publicly available data on CELLxGENE, as depicted in the left panel below. In contrast, our approach aimed to minimize batch effects experimentally. We achieved this by pooling patient samples together and distributing the pool across multiple wells without additional barcoding. As a result, we have minimal batch effects and can more easily compare across patient samples, as shown below in the right panel.

Our multiplexed approach also significantly reduces the cost per cell. We can better discriminate doublets using *demuxlet* and *freemuxlet*, so we can load 3x as many cells per well and an average of 4 patient samples per well, yielding >4x cost-savings. Please see the schematic below for our processing workflow.

Reviewer 3:

“Through the manuscript, the comparisons are made between HC vs UC and UC vs VDZ-UC. Could the authors show the comparison between HC vs VDZ-UC to define the difference between these groups (e.i. if the treated patients show signs of returning to a healthy cellular state)?”

Response:

Thank you for raising this point. Throughout the manuscript, we have conducted comparisons among HC, UC, and UC-VDZ patients for both abundance and proximity analyses using ANOVA. The notable exception was the MAST analysis where a direct comparison between HC and UC-VDZ patients was performed but not explicitly shown. However, we have performed this comparison, and the results can be found below.

Blood and biopsy scRNA-seq DE genes in coarse cell subsets comparing HC and VDZ-treated UC patients. **a**, scRNA-seq pseudobulk DE genes in PBLs with $\log_2\text{fc} > 1$ or < -1 HC vs UC-VDZ. **b**, PBL scRNA-seq DE genes in coarse cell annotation subsets identified by MAST with $\log_2\text{fc} > 2$ or < -2 HC vs UC-VDZ. **c**, scRNA-seq pseudobulk DE genes in biopsies with $\log_2\text{fc} > 1$ or < -1 HC vs UC-VDZ. **d**, Biopsy scRNA-seq DE genes in coarse cell annotation subsets identified by MAST with $\log_2\text{fc} > 2$ or < -2 HC vs UC-VDZ. All displayed p-values are Bonferroni corrected; ribosomal and mitochondrial genes are not displayed.

Reviewer 3:

“Only one VDZ-UC patient has a high inflammation score, this should be made clear in text when discussing the conclusions between cell changes due to the treatment vs disease severity.”

Response:

We appreciate this concern. Our revised manuscript validates significant effects on mDCs in patients during remission, indicating that the changes we observed are linked to VDZ treatment rather than solely stemming from differences in endoscopic disease severity. Furthermore, our original study revealed decreases in mDCs in biopsies from VDZ-treated patients even when compared to healthy controls. This suggests that the observed reduction in mDCs were unlikely to be solely due to the heightened endoscopic activity in the UC patients.

In addition to these CyTOF data, we also outline a new spatial transcriptomics CosMx analysis of archived FFPE specimens, pre- and post-treatment, in VDZ responders and non-responders. Those data include patients with balanced endoscopic disease activity prior to VDZ therapy, further controlling for endoscopic severity (**Supplementary Table 1**). Furthermore, by examining samples before and after therapy across both responder and non-responder groups, we account for the potential effects of varying timepoints. These points are discussed in greater detail above.

Reviewer 3:

“In Fig 1e, while this plot with log transformed scale is informative for fresh vs cryopreserved comparison, it may be misleading, as the sample is actually dominated by plasma and T cells, but this is not reflected in the plot. This applies to other bar plots throughout the manuscript.”

Response:

We appreciate the observation and understand the concern. Cell frequencies are often displayed on a log scaled to highlight important differences among less abundant cell subsets. Less abundant cell subsets can still exert potent effects on homeostasis. Displaying abundance on a log transformed scale allows visualization of cell subsets across a wide range of frequencies. To address the concern, we present the data both in its original and log-transformed scales side-by-side below. There is a noticeable decrease in Mast cells in the cryopreserved biopsies, which is somewhat anticipated. This was not statistically significant, and Mast cells were not the focus of this study.

We want to emphasize that we are not asserting that there are no differences between fresh and cryopreserved biopsies. Rather, we are highlighting that most cell subsets of interest are retained post-cryopreservation. Neutrophils were not present in either fresh or cryopreserved biopsies, but this is known to be a result of degranulation and poor neutrophil encapsulation by the 10X platform. Overall, the degree of concordance between freshly processed and cryopreserved intestinal biopsies, combined with the logistic, financial, and batch processing benefits of cryopreserved biopsies, favored cryopreservation for this study.

Regarding the concern “this applies to other bar plots throughout the manuscript.”, our choice between log-transformed and linear representations is made to optimize data visualization for each context. For instance, the CITE-seq data display each cell subset separately, on a linear scale.

However, when we represent multiple subsets in a single graph, we employ a log-transformed scale for clarity, as shown in the figure below.

As Reviewer 3 is aware, the displayed scale does not affect the downstream statistical testing. Our choice between linear and log scales is driven by the range of the data set at hand. Cell subset abundance can vary 100-fold for some cell subsets, so it is appropriate to use a log scale to display this dynamic range.

Reviewer 3:

“Related to the previous point, to support the enrichment result, the authors can consider using milo tool (Dann et al., Nat. Biotechnology, 2022) for a statistical approach to investigate the differential abundance across conditions in single cell data.”

Response:

We thank Reviewer 3 for recommending that we consider the milo tool for examining differential abundance in single-cell data. We are not employing milo here for several reasons. Primarily, we used a multiplexed and pooled scRNA-seq strategy, combining cells into a large pool prior to loading. This approach dramatically reduces batch effects and simplifies differential abundance analysis (Fig 3d). Our coarse and fine annotations were conservative, and we are not analyzing continuous covariates that would benefit from a package like milo.

We would also like to emphasize that our multi-omics approach employed CyTOF on the same exact samples as the CITE-seq analysis. We analyzed the CyTOF data with both a supervised gating strategy and an unsupervised pipeline. These additional layers of experimental and technical rigor allow us to identify cell subsets with greater confidence, without relying exclusively on computational methods to deal with variability in cell clusters.

Reviewer 3:

“In blood CyTOF data (or Figure 4p), there is an increase in retention of a few myeloid subsets (cDC1, cDC2,

cDC2b, pDC), but the only population with significant reduction in colon samples (or Figure 4q) is cDC1. Can authors comment on their interpretation why only cDC1 shows change in the biopsy samples? Was this trend also observed using other profiling technologies (e.i. number of CD103 expressing cells in scRNAseq data)?”

Response:

Thank you for raising this insightful question. First, we highlight that in the revised manuscript, we now aggregate multiple cDC subsets into a single tissue mDC subset, because the low frequency of individual cDC subsets in the tissue could be unreliable. The overall $\alpha_4\beta_7^+$ mDCs in the tissue by CyTOF were not significantly reduced by VDZ, although the trend remains. This could be explained by multiple factors. A crucial point to bear in mind is that the mucosal biopsies specifically target the colonic mucosa, which is the primary site affected by ulcerative colitis. This strategy was deliberate. However, with this approach, we are not assessing the submucosa, muscle layers, Peyer’s patches, mesenteric lymph nodes, and the entirety of the small intestine. It is conceivable that the peripheral $\alpha_4\beta_7^+$ rise is a cumulative result of shifts occurring in these non-sampled sites, with only a subtle change observable in the mucosal layer of standard colonic biopsies. Another possibility is that monocytes enter the colon and differentiate into monocyte-derived DCs, filling the DC niche.

We did analyze CD103+ DCs using the CD103 ADT marker from CITE-seq. CD103+ DCs constituted 30% of the DCs in the biopsies. There is an apparent reduction in UC-VDZ patients, but this may reflect the broad reduction of mDCs in UC-VDZ patients in the CITE-seq data. We can say that the CD103+ DCs follow the same overall trend as mDCs in the CITE-seq data.

These observations align with murine data from Clahsen et al. *Clinical Immunology*. 2015.

Reviewer 3:

“Related to the previous point, can the authors elaborate on how MNPs might contribute to disease based on the current state of knowledge of the role of MNPs in inflammatory diseases?”

Response:

MNPs play a critical role in maintaining intestinal homeostasis by controlling immune tolerance towards dietary and commensal antigens, while also triggering immune responses against insults (Kim et al., 2018. *Immunity*; Mowat et al., 2014. *Immun Rev*; Hadis et al., 2011. *Immunity*; Cerovic et al., 2014. *Trends Immunol*). Understanding MNP function is challenging due to their diverse subsets, which vary in function. Additionally intestinal MNP activity is influenced by other cell types such as other immune, epithelial, and stromal cells as well as microbial components (Garrido-Trigo et al., 2023 *Nat Commun*; Friedrich et al., 2021 *Nat Med*; Martin et al., 2019. *Cell*; Smillie et al. 2019. *Cell*; Kinchen et al., 2018. *Cell*; Schirmer et al., 2019. *Nat Rev Microbiol*; Russell et al., 2019. *Nat Rev Immunol*). Our study supports an important role for MNP subsets in UC, and their potential interaction with other cellular compartments. This observation aligns with studies indicating a heightened presence of various myeloid cells in IBD-afflicted colons (Bain et al., 2014. *Immunol Rev*; Steinbach et al., 2014. *Inflamm Bowel Dis*; Liu et al., 2019. *BMC Immunol*). In our spatial transcriptomic analysis, we localized activated MNPs in VDZ non-responders pre-treatment, corroborated by external validation, and supported by another study conducted *in silico* (Liu et al., 2019. *BMC Immunol*). Indeed, in the Liu et al study, the authors also reported enrichment of MNPs in VDZ non-responders as well as in murine models of chronic colitis (Liu et al., 2019. *BMC Immunol*). Our work, builds on these insights, highlighting an amplified abundance and activation of tissue MNP subsets in UC compared to healthy controls. Notably, these MNP and stromal subsets express genes indicative of VDZ non-response before treatment.

Reviewer 3:

“Could the authors also investigate/elaborate more on the MADCAM1 expression in your single cell and

CytoTOF datasets as this is the other part of the interaction via $\alpha4\beta7$ integrin? Is there a change in MADCAM1 expression or MADCAM1+ cell type abundance before and after VDZ treatment?”

Response:

Thank you for this important query regarding MADCAM1 expression. Anti-MAdCAM-1 was not included in our CyTOF panel or CITE-seq panels, and *MADCAM1* probes were not in the CosMx panels (**Supplementary Table 5**). However, we analyzed *MADCAM1* by RNA-ISH (**Extended Data Fig 13b** and shown below). There was no significant difference in *MADCAM1* by RNA-ISH across groups. There was also no significant difference in *MADCAM1* at the transcript level by scRNA-seq. Overall, mRNA levels for *MADCAM1* are low, and may not correlate well with protein levels. There are some prior studies that are relevant for this topic. Battat R et al. *Inflamm Bowel Dis*. 2019 reported that s-MAdCAM-1 declined more rapidly in responders compared to non-responders, but we are not analyzing soluble, secreted analytes in this study. Also, sMAdCAM-1 was not associated with outcomes in that study. Arijs I et al. *Gut*. 2018 showed that UC patients expressed higher levels of *MADCAM1* compared to healthy controls pre-treatment by qPCR and IHC, and expression decreased in UC responders at 52 weeks. Importantly, the levels did not appear to differ at baseline among responders and non-responders, and intermediate timepoints did not show a significant difference in *MADCAM1* expression. Recently, Roosenboom B et al. *Inflamm Bowel Dis*. 2023 reported that a higher proportion of MAdCAM-1+ venules, measured semi-quantitatively, may be associated with response to VDZ. Overall, expression of *MADCAM1* has unclear utility as a predictive biomarker for VDZ therapy thus far.

Reviewer 3:

“To make the MNP - activated fibroblast observation stronger, the authors should consider doing cell-cell interaction analysis in their single cell blood and colon data using CellphoneDB, NicheNet or similar ligand-receptor analysis method. This would allow the authors to interrogate the interactions between these cell subsets and propose a mechanism of how these interactions lead to epithelial cell recovery in VDZ-UC treated patients or are dysregulated in non-responder patients.”

Response:

We appreciate the suggestion provided by the reviewer to explore cell-cell communication. In response, we utilized the Cellchat R package (Jin et al., 2021), a tool comparable to those highlighted by Reviewer 3. Applying this analysis to our colonic scRNA-seq data, we identified 32 ligand-receptor pairs within the coarsely annotated cell types. The accompanying aggregate circle plots detail the number and intensity of cell-cell interactions (**A**).

Of note, while the CD8 T cells appear to interact with multiple cell types, this is largely due to MHC I expression on other cells. Similarly, collagen production by fibroblasts is largely responsible for their high of number ligand-receptor interactions. Our primary interest was the interaction between MNPs and fibroblasts. In examining these subsets (**B**), we noted that MNPs had limited interactions with other subsets. The most pronounced were interactions with CD8 T and NK cells due to MHC I expression (**B**). Conversely, fibroblasts interacted with all subsets, mainly through collagen interactions (**B**).

To delve deeper, we applied the analysis to our finely annotated data, focusing on the interaction of various MNP and fibroblast populations. Of the 84 pathways identified, we concentrated on a select few highlighting MNP-fibroblast interactions. Notably, the THY pathway indicated cell-cell interactions between different stromal subsets, monocytes, mDCs, and pDCs (**C,D**). THY-1, or CD90, is a marker for mesenchymal stromal cells. Consistent with prior research, our Cellchat analysis revealed that integrins like ITGB2 act as THY-1 receptors (Herrera-Molina et al., *Int. Rev. Cell. Mol. Biol.* 2013; Saalbach et al., *Oncogene*. 2005; Leyton et al., *Curr. Biol.* 2001). As depicted in plot C, the two THY-1 ligands, ITGAX and ITGB2, are predominantly expressed in myeloid cells, especially macrophages, monocytes, and mDCs, signifying their interaction via this ligand-receptor pair.

C. THY1. Violin plots of ligand:receptor gene expression patterns in the different cell types.

D. THY1. Circle plot showing the cell-cell interactions for the indicated ligand:receptor pairs.

Additionally, the OSM pathway emerged as an exclusive mediator of myeloid-stromal-endothelial interactions (E). OSM, a member of the IL-6 family, plays a role as an inflammatory mediator in various contexts.

E. OSM. (Top) violin plots of the ligand:receptor pair gene expression patterns in the different cell types, and (bottom) circle plot showing the cell-cell interactions for the indicated ligand:receptor pairs.

CellChat analysis also highlighted the NRG pathway, which displayed interactions within specific cell subsets of interest (stromal, MNP and epithelial subsets).

In response to the reviewer's insights, our further analysis with CellChat illuminated the role of the NRG pathway. This pathway exhibited interactions among specific cell subsets of interest, including stromal, MNP, and epithelial subsets. We observed that NRG1 is expressed by activated fibroblasts, stromal S2, and monocytes, while the ERBB3 and ERBB2 receptors are found in epithelial subsets (F). Previous research has identified NRG1 as a primary fibroblast-derived EGF ligand that promotes epithelial proliferation during homeostasis and regeneration (Jarde et al., *Cell Stem Cell*. 2020) and fosters epithelial maturation and cellular differentiation (Yu et al., *Cell*. 2021; Holloway et al., *Cell Stem Cell*. 2021). Recent studies have further probed NRG1's influence on the reprogramming and protection of the intestinal epithelium in IBD (Lemetyinen et al., *Dis Model Mech* 2023; Garrido-Trigo et al., *Nat Commun* 2023). Specifically, one study underscored the overexpression of NRG1 in S2 fibroblasts and a subset of macrophages in UC inflamed colon (Garrido-Trigo et al., 2023). Thus, NRG1 holds significant importance as stem cell niche factor and is closely associated with NOTCH signaling.

Next, we examined ligand-receptor pathways directly relevant to VDZ in UC (G). As anticipated, *MADCAM1* was prominently expressed on venous endothelial cells. High levels of *ITGA4* expression were detected in Tregs, monocytes, macrophages, and mDCs within biopsies. *mDCs* were among the top *ITGB7*-expressing subsets in intestinal biopsies, along with gd T cells, cycling T cells, and cycling plasmablasts. The elevated expression of *ITGB7* on mDCs, relative to most other subsets, could explain why mDCs are significantly excluded by VDZ. This observation further suggests a mechanism by which mDCs could be selectively targeted by anti- $\alpha 4\beta 7$. The binding of $\alpha 4\beta 1$ to VCAM1 is an alternative pathway for intestinal trafficking. Interestingly, *ITGB1* was expressed at high levels by NK, ILC, MNP subsets. It is worth noting that integrin mRNA is generally expressed at low levels, and this might not directly align with surface expression.

Given the importance of $\alpha 4\beta 7$ -independent pathways, we also analyzed chemokine pathways (H). Various chemokines were expressed by MNP and stromal subsets, notably CXCL12 by stromal subsets, and CXCR4 across multiple immune subsets, including mDCs and pDCs. Intriguingly, CXCR4 and CXCL12 were differentially expressed across HC, UC, and VDZ patients (Extended Data Fig.8).

H. Chemokine and chemokine receptor pathways relevant to MNP and fibroblast interactions. Violin plots of ligand:receptor pair gene expression patterns in the different cell types.

We are grateful for the recommendation to employ CellChat analysis. The colon mucosal microenvironment certainly presents numerous potential ligand-receptor interactions involving MNP, stromal, and epithelial cells. Those subsets were of particular interest to our study, although there are complex interactions to consider across all mucosal cell subsets. We have opted not to incorporate these analyses in the revised manuscript due to considerations about the direct correlation of gene and protein expression, and due to extensive extended data already in the revised manuscript. This analysis will be accessible online within the public peer review file.

Reviewer 3:

“In addition to responders vs non-responders to VDZ GSEA analysis, the authors could use bulk data deconvolution methods (MuSiC or SCDC) to infer the proportion of cells in the responders vs non-responders based on their single cell data.”

Response:

We appreciate this suggestion. Our approach prioritized direct measurement of intestinal cell subsets by leveraging expansive new datasets, specifically CyTOF and CosMx. We believe that these direct measurements offer a more robust validation of cell subset frequencies compared to inference methods like MuSiC or SCDC, even though those are innovative tools in the field. Furthermore, there are limited publicly-available, longitudinal, bulk transcriptomic datasets available for VDZ, constraining the application of bulk deconvolution methods.

GSEA is an extensively validated and well-characterized tool for analyzing cell subset-specific gene sets, with thousands of applications to existing bulk datasets. Our leading-edge analysis in Fig. 8c and shown below, exhibits minimal overlap among the cell subset-specific gene signatures. This suggests that our GSEA-based

approach is yielding an accurate and independent assessment of the relative abundance of each cell type. Importantly, the GSEA results agree with the other data we generated in our study.

Reviewer 3:

“Minor comments:

Fig 1d “03B CD4 T naïve and Treg” changed to “03B- CD4 T naïve and Treg” for consistency.”

Response:

Thank you for pointing this out. The labeling was fixed and now it is consistent with the others.

Reviewer 3:

“What is the reason for very low epithelial cell numbers from both fresh and cryopreserved samples? Typically epithelial cells are the majority of retrieved cells from mucosal biopsies. Please comment on this in the manuscript.”

Response:

We thank the reviewer for this question. We highlight this point in **Extended Data Fig. 14d**, shown here:

There are several reasons for this observation. While epithelial cells may be dominant in mucosal biopsies, this ratio is disrupted during subsequent processing steps, including cryopreservation, digestion, dead cell depletion, and encapsulation. Anoikis, a form of dissociation-induced apoptosis characteristic of IECs, contributes significantly to the decreased abundance of IECs during processing. We try to counteract this phenomenon by introducing the Y-27632 ROCK inhibitor during all processing stages. However, even with this

intervention, a substantial fraction of IECs will initiate apoptosis. Initiation of apoptosis increases surface membrane Annexin V. For CyTOF, we do not deplete dead or dying cells, because these cells can be easily gated out with Cisplatin staining. For CITE-seq, it is important to enrich for live cells by depleting Annexin V-expressing cells prior to loading. This process, though crucial for quality, further reduces the number of IECs. Additionally, 10X chips have a bias towards encapsulating smaller cells, making the larger IECs less likely to be captured. Importantly, reduced IEC frequency by CITE-seq as compared to CyTOF is true whether samples are processed fresh or cryopreserved. Finally, our IEC frequency by CITE-seq is similar to Garrido-Trigo A, et al. *Nature Communications*. 2023, which performed 10X scRNA-seq of freshly processed biopsies.

We agree with Reviewer 3 on the importance of this observation, which is why we are showing the advantages of CosMx, MIBI, CODEX, or any spatial multi-omics method for more accurately representing the *in vivo* frequency of IECs in the tissue (**Extended Data Fig. 14d**, also shown above).

Reviewer 3:

“For Fig 1f, did authors compare changes in specific populations: for example mast cell clustering in Fig 1c seems to be affected more than the others.”

Response:

Thank you for bringing to our attention **Fig 1c,f** and the seemingly evident changes in mast cells. Even though at a glance the mast cells seem to be affected, when we perform Mann-Whitney tests with FDR correction, the mast cells had a q-value of 0.7, surpassing our threshold of $q < 0.1$. It is likely that granulocytes, including mast cells, are under-represented in cryopreserved biopsies, so we would not choose cryopreservation if those were our primary cell subsets of interest.

Reviewer 3:

“HS12 had an expanded circulating cytotoxic lymphocyte 3 population, but this was not observed in the other UC-VDZ patients (Fig. 2b,c).” This is not clear from the umap plots. There are clear changes in the circulating lymphocyte 6 populations. Could the authors show this in a more quantitative manner (e.g. barplot)?”

Response:

Reviewer 3 makes an excellent observation. We were also intrigued by this finding, though we felt this cell population was unique to HS12, rather than a generalizable phenomenon. In the scRNA-seq of PBLs, when we stratified by condition and by patients, we observed VDZ-enriched subsets of CD8 T effector memory (Tem) and NK cell cells. As correctly observed by Reviewer 3, these were dominated by a single patient, HS12, with high levels of *GNLY*, *GZMH*, and *ZNF683* expression (**Fig. 2b-d**, and shown below).

This is an intriguing cell subset, but unique to HS12. When we compared the cell frequency for each fine leukocyte cell subset across conditions, we did not observe any statistically significant differences among HC, UC, and UC-VDZ patients. The HS12 cytotoxic population was also observed in CyTOF where these VDZ-enriched regions were dominated by HS12, confirming the distinct cytotoxic lymphocyte phenotype identified for this patient by scRNA-seq (see **Extended Data Fig 9b-d**, also shown below, clusters 7,10,11,14). This emphasizes why having orthogonal multi-omics techniques applied to the same samples can be useful for dissecting patient heterogeneity and increasing internal validity of cell subset assessments.

Here are the bar plots from **Fig. 2e**.

Here are stacked bar plots for the various cell subsets per patient by scRNA-seq, as requested by Reviewer 3:

ZNF683 has been linked to a T cell subset by scRNA-seq of patients with Crohn's disease (Jaeger N et al. *Nature Communications*. 2021). Its significance in this one patient with UC is unclear. Nevertheless, having identified this subset in one patient with CITE-seq and CyTOF demonstrates the power of using multiple methods to immunophenotype individual patients.

Reviewer 3:

"In Fig 2f, to show that there is a deregulation of the UC associated genes in VDZ-UC patients, could the authors plot the expression of the UC upregulated genes in one plot with HC, UC and VDZ-UC patients, similar to Figure 4o."

Response:

All DE genes have been plotted as heatmaps, which better reflects patient heterogeneity. Thank you for this suggestion. Regarding differential regulation, we feel this is best conveyed in the reciprocal expression analysis plots (**Fig 2h** and **Extended Data Fig 8b**).

Reviewer 3:

"The relative increase in goblet cells we observed was likely due to low goblet cell counts, and a relative

reduction in absorptive colonocytes and intestinal stem cells (ISCs) (Extended Data Fig. 6a,b).” The authors could cite other IBD single cell papers that report Goblet cell changes, including Elmentaite et al., *Developmental Cell*, 2019 (<https://doi.org/10.1016/j.devcel.2020.11.010>) & Kanke et al., *cmgh*, 2022 (<https://doi.org/10.1016/j.jcmgh.2022.02.005>)”

Response:

Thank you for these suggestions, these citations have been added.

Reviewer 3:

“What were the raw numbers for the comparisons made in Figure 4g-h, I-n (how many cells per condition)? This would be useful for interpretation, given that Cluster 18 in Fig 4b is <10% of all captured cells.”

Response:

Here are the raw cell counts (cells per condition) for mononuclear phagocytes in colonic biopsies by CyTOF, including both the unsupervised and supervised analyses.

Cell subsets	HC	UC	UC-VDZ
18_MNP_unsuperv (HS1-HS12)	4365	5041	1845
MNP_supervised (HS1-HS12)	4797	4569	1884
MNP_supervised (HS13-HS30)	4098	5153	2536

Reviewer 3:

“Despite limited cell counts with MIBI, tissue MNPs and fibroblasts were associated with a trend toward reduction in UC-VDZ compared to UC (Fig. 5d; Extended Data Fig. 11c).” Fig 5d only shows fibroblasts, but not MNPs, could you highlight the MNP subsets in Extended Fig 11c or show these subsets in Fig 5d.”

Response:

Thank you for pointing out this oversight. As rightly noted by Reviewer 3, **Fig. 5d** highlights the trend in fibroblasts between the two patient groups. We've made the necessary adjustment in the text to ensure the representation is accurate. This has been edited in the revised manuscript to “Despite limited cell counts with MIBI, there was a trend indicating a reduction in fibroblasts in UC-VDZ compared to UC biopsies (**Fig. 5d; Extended Data Fig. 11c**).” which is shown. Overall, the MIBI analysis was mostly useful for validating markers for immunophenotyping of FFPE biopsies, and for defining cell phenotypes that correlate with the other analyses. The FOVs were too small to draw substantial conclusions regarding differential abundance.

Reviewer 3:

“In Figure 6f, the field of view or section type for VDZ-UC patients is different than for HC or UC representative image. This could affect the differences found. In addition, there seems to be an increased number of activated fibroblasts in the VDZ-UC patient image, therefore smaller neighbourhood numbers could be driven by the few MNP neighbourhoods inferred. How do authors account for that?”

Response:

Thank you for pointing this out. As mentioned above, we now include a different UC image to compare to the UC-VDZ patients for clarity. With the additional spatial transcriptomics analysis in the revised manuscript, we agree there is variability in the measurement of neighborhood enrichment z-scores. The proximity analysis is highly dependent on chosen radius and cell frequency. We are now reporting trends toward proximity of activated fibroblasts and activated MNPs in active colitis, and describing how we analyzed these parameters.

We are also showing similar trends in our longitudinal dataset.

Reviewer 3:

“Page 12 line 22 “Net Enrichment Scores (NES)” and the figure 6i legend says “Normalised Enrichment Scores (NES)”. Please use one.”

Response:

We thank Reviewer 3 for this correction. The acronym is Normalized Enrichment Score, and it has been fixed in the text according to the figure legend.

We greatly appreciate the thorough comments and suggestions from Reviewer 3, and we believe the revised manuscript has improved significantly as a result.

REVIEWERS' COMMENTS

Reviewer #1 (Remarks to the Author):

The authors have substantially revised the manuscript in response to the reviewers' comments and the paper has clearly improved.

However, there are still some wording issues, where the authors claim to have observed effects of VDZ on specific cell populations (e.g., p. 2, l. 8/9; p. 5, l. 23) rather than more correctly describing (potentially indirect) effects on the abundance of these cell populations or the expression of genes.

I also miss an explicit acknowledgement of the limitation that no functional investigations have been performed and that it can therefore not be concluded whether the observed alterations are directly or indirectly related to VDZ.

I further feel that it is needed to soften the claim that "VDZ significantly impacts intestinal MNP trafficking" (p. 14, l. 7/8), since the data presented do not provide any direct evidence for altered trafficking.

Reviewer #2 (Remarks to the Author):

The authors present an extensively revised version and added substantial amount of newly generated data that sufficiently addressed my criticism.

Reviewer #3 (Remarks to the Author):

The authors address our concerns, provide additional analysis based on our suggestions as well as new data that further increases the value of the study as a resource. I don't have any additional comments for the authors.

Point-by-point response:

Reviewer 1:

“Reviewer #1 (Remarks to the Author):

The authors have substantially revised the manuscript in response to the reviewers' comments and the paper has clearly improved.

However, there are still some wording issues, where the authors claim to have observed effects of VDZ on specific cell populations (e.g., p. 2, l. 8/9; p. 5, l. 23) rather than more correctly describing (potentially indirect) effects on the abundance of these cell populations or the expression of genes.

I also miss an explicit acknowledgement of the limitation that no functional investigations have been performed and that it can therefore not be concluded whether the observed alterations are directly or indirectly related to VDZ.

I further feel that it is needed to soften the claim that "VDZ significantly impacts intestinal MNP trafficking" (p. 14, l. 7/8), since the data presented do not provide any direct evidence for altered trafficking.”

Response:

We thank the reviewer for commenting on the substantial revisions and improvements in the manuscript. We appreciate the concerns regarding wording and agree with all the points raised.

The lines indicated above have all been edited in the revised manuscript. The corresponding page and line numbers for the above sentences are now page 2 lines 7-8, page 5 lines 23-24, and page 14 lines 9-10, respectively, in the revised manuscript. We have also highlighted the absence of functional studies as a significant limitation in the discussion section (page 15 lines 13-15). We hope these revisions address your concerns.

Reviewer #2 (Remarks to the Author):

The authors present an extensively revised version and added substantial amount of newly generated data that sufficiently addressed my criticism.

Response:

Thank you very much for reviewing our manuscript and for your insightful comments.

Reviewer #3 (Remarks to the Author):

The authors address our concerns, provide additional analysis based on our suggestions as well as new data that further increases the value of the study as a resource. I don't have any additional comments for the authors.

Response:

Thank you very much for reviewing our manuscript and for your insightful comments.